

# Reviews and synthesis: Weathering of silicate minerals in soils and watersheds: Parameterization of the weathering kinetics module in the PROFILE and ForSAFE models

Harald Ulrik Sverdrup[1*], Eric Oelkers[5], Martin Erlandsson Lampa[2],
Salim Belyazid[3], Daniel Kurz[4], Cecilia Akselsson[6],

1-Industrial Engineering, University of Iceland, Reykjavik, Iceland, 2-Institute of Hydrology, University of Uppsala, Uppsala, Sweden, 3-Physical Geography, Stockholm University, Stockholm, Sweden, 4-EKG Geoscience, Bern, Switzerland, 5-Earth Sciences, University College London, Gower Street, WC1E 6BT, London, UK, 6-Earth Sciences, University of Lund, Lund, Sweden. *corresponding author (hus@hi.is)

## Abstract

The PROFILE model, now incorporated in the ForSAFE model can accurately reproduce the chemical and mineralogical evolution of the soil unsaturated zone. However, in deeper soil layers and in groundwater systems, it appears to overestimate weathering rates. This overestimation has been corrected by improving the kinetic expression describing mineral dissolution by adding or upgrading 'breaking functions'. The base cation and aluminium brakes have been strengthened, and an additional silicate brake has been developed, improving the ability to describe mineral-water reactions in deeper soils. These brakes are developed from a molecular-level model of the dissolution mechanisms. Equations, parameters and constants describing mineral dissolution kinetics have now been obtained for 102 different minerals from 12 major structural groups, comprising all types of minerals encountered in most soils. The PROFILE and ForSAFE weathering sub-model was extended to cover two-dimensional catchments, both in the vertical and the horizontal direction, including the hydrology. Comparisons between this improved model and field observations is available in Erlandsson Lampa et al. (2019, This special issue). The results showed that the incorporation of a braking effect of silica concentrations was necessary and helps obtain more accurate descriptions of soil evolution rates at greater depths and within the saturated zone.

## 1. Introduction

Chemical weathering of silicate minerals, and notably the dissolution rates of these minerals are one of the most important factors shaping soil chemistry over longer time periods. The quality of the kinetic database in most cases determines the quality of the simulations. In the 1980's, the need arose to mitigate acid deposition, to set critical loads for acid deposition, and to set limits for sustainable forest growth and nitrogen critical loads. The critical loads depend directly on the ability of the soil to neutralize the incoming acid, thus the critical load depends on the weathering rate. It became apparent that the usual approach to soil geochemical modelling of using the weathering rate as the adjustable parameter to make the simulations fit the data, would be inadequate for estimating the critical loads. As a consequence, a quest for creating a weathering rate models that would accurately reproduce field observations and based on fundamental principles was started (Warfvinge and Sverdrup 1985, Sverdrup and Warfvinge 1987).

With funding from the Swedish Environmental Protection Agency, the Swedish Agricultural Research Council and the Swedish Ministry of the Environment, a major research effort was begun. This mission led to a re-evaluation of the weathering observations available in scientific publications and books (Sverdrup 1990, Sverdrup and Warfvinge 1992, 1993, 1995, Drever et al., 1994, Drever and Clow 1995, Ganor et al., 2005, Svoboda-Colberg and Drever 1993, Crundwell 2013). The mission and the funding allowed creation of an alternate path that led to a model that accurately reproduced weathering rates under field conditions. The first steps and the narrative of the development was reported by Sverdrup and Warfvinge (1988a,b, 1992, 1993, 1995) and Sverdrup (1990). In 1990, we had a set of models that described the rates 14 minerals (K-feldpar, albite, plagioclase, pyroxene, hornblende, garnet, epidote, chlorite, biotite, muscovite, vermiculite, apatite, kaolinite, and calcite). Later more silicate





minerals were added, minerals including illite-1, illite-2, illite-3, smectite, montmorillonite, sericite and rich volcanic glass and poor volcanic glass, and eventually 45 additional silicate minerals where we had full kinetic data. In addition, we had full kinetic data for 25 different carbonates[1] at the time.

At the start of this effort in the middle of the 1980's, it became clear that we did not have a standard procedure for building a weathering rate model based on molecular level mechanisms. There are many reasons for this, the most important ones were the lack of a mechanistically oriented models for guiding experimental studies at the time. The lack of an understanding of the mechanisms, resulted in important factors being overlooked. Many essential variables were missing in the older experimental studies, sample preparation was often inadequate or not done, and/or the material was inadequately characterized (Sverdrup et al., 1981, 1984, Sverdrup, 1990). Often the experimental design had significant flaws and many experiments ran for too short a time; see Sverdrup (1990) for a full description. As such there needed to be a sorting of the data, to avoid the confusion brought by misleading data. This effort lead to the creation of the original PROFILE mineral kinetic weathering model (Sverdrup, 1990), to estimate the rate at which mineral dissolution provided essential cations to soil waters. Although this model provides accurate estimates for shallow soils, it became less accurate for deeper soils (e.g. > 1.5 meter soil depth).

This study outlines our efforts to update these early mineral weathering kinetics models for watershed water chemistry and deeper groundwater. This effort is the result of preparations for, discussions at, and subsequent efforts after a workshop held at Ystad Saltsjöbad, Ystad, Sweden, April 11-14, 2016, in connection to the Swedish QWARTS research programme. Key literature to read to aid in following this text are the weathering book by Sverdrup (1990) and the articles Sverdrup and Warfvinge (1988a,b, 1992, 1995) and Warfvinge and Sverdrup (1993). There is an advisory chapter on how to operationally estimate weathering rates in soils on a regional scale in Europe in the United Nations Economic Commission for Europe, Long Range Transboundary Convention Mapping Manual for Critical loads (Sverdrup, 1996). The weathering rate mapping methodology was tested and used throughout 26 different European countries, and peer reviewed at annual workshops from 1988 to 2017. Weathering rates in forest soils and open terrestrial ecosystem have been mapped during the period 1990 to the present (2019). The UN/ECE-LRTAP Critical loads and levels Mapping Manual was updated biannually during the period.

The revision of the original weathering rate models was motivated by several observations:

1. The PROFILE model works satisfactorily in the unsaturated zone (0-1 meter), on thin soils, on rock surfaces, in low concentration systems (Sverdrup and Warfvinge 1988a,b, 1991, 1992, 1993, 1995, 1998, Sverdrup 1990, Sverdrup et al., 1998, Hettelingh et al., 1992, Alveteg et al., 1996, 1998, 2000, Alveteg and Sverdrup 2000). Test show that the weathering kinetics as of 2015 works very well for these situations.
2. However, it appears as the chemical weathering rate for minerals is overestimated by this model in deeper soils, at depths of more than 1.5 meter depth. The original PROFILE model was used down to this depth (Sverdrup et al., 1988a,b, 1992, 1996, Sverdrup 1990, Janicki et al., 1993, Holmqvist et al., 2003) for critical loads for streams (Sverdrup et al., 1996) and groundwater (Warfvinge et al., 1987), and may have possibly resulted in an overestimation of the critical load.
3. The weathering rate is overestimated in the deeper soils and in ground water (Sverdrup 1990, Warfvinge and Sverdrup 1987, 1992a,b,c, Sverdrup et al., 1996). The PROFILE model was designed for groundwater composition calculations, and has proven to provide inaccurate estimates in such systems.
4. It is evident that the new experiments published in the literature after 1995 is of far better quality and consistency, with better experimental designs, better characterized materials and more complete data than previous studies. For example, the reader is encouraged to read two studies published by Holmqvist et al., (2002, 2003) on the weathering rates of clay minerals under soil

---

[1] **Calcite** (The calcites are all slightly different; $CaCO_3$ with 0-3% $MgCO_3$ and 0.05%-0.5% apatite, from Sweden, Norway, Denmark,and the United States. In addition, kinetics on **aragonite** ($CaCO_3$), **slavsonite** ($SrCO_3$), **dolomite** ($CaMg(CO_3)_2$, **magnesite** ($MgCO_3$), **brucite** ($MgOH$), **siderite** ($FeCO_3$), **witherite** ($BaCO_3$), and **rhodochroisite** ($MnCO_3$) is available.





conditions and the concept of mineral alteration sequences (Holmqvist 2004, PhD thesis from Chemical Engineering, Lund University). The minerals used in the weathering rate experiments in those studies were minerals extracted and separated from in-situ soils at experimental field sites near Uppsala, Sweden. The consideration of these data allow for a significant improvement in the previously created databases.

## 2. Scope and objectives

The scope of this study is to describe the updated mineral kinetics database used in the PROFILE and ForSAFE models, and describe how the model has been improved during the past several years. Notably this update includes reaction product 'brakes' in the kinetic rate equations to better fit the observed data down to the groundwater table and below. This was necessitated when the ForSAFE model (thus also the PROFILE model) was reconfigured for a sloping catchment, expanding the model structure from a 1-dimensional model, with only the vertical soil profile and forest stand aspect, to a 2-dimensional model accounting for vertical and horizontal solute transport in a catchment, including the ecosystem. In total 102 minerals are considered in the updated and expanded kinetics parameter databases. An exhaustive description of the parameterization of the rate equations for all of the 102 minerals will require a text far beyond what is possible in this manuscript, so that only a summary and several examples are provided here. This study is focussed on updating the mineral weathering kinetics parameterizations and their adaptation to soil profiles, watershed water chemistry and deeper groundwater to be able to enable improved integrated forestry and environmental assessments.

The arrow shows a causality. A variable at the tail causes a change to the variable at the head of the arrow.

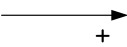

A plus sign near the arrowhead indicates that the variable at the tail of the arrow and the variable at the head of the arrow change in the *same* direction. If the tail *increases*, the head *increases*; if the tail *decreases*, the head *decreases*.

A minus sign near the arrowhead indicates that the variable at the tail of the arrow and the variable at the head of the arrow change in the *opposite* direction. If the tail *increases*, the head *decreases*; if the tail *decreases*, the head *increases*.

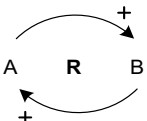

The letter **R** in the middle of a loop indicates that the loop is *reinforcing* a behavior in the same direction, causing either a systematic growth or decline. It is a behavior that is moving *away* from equilibrium point.

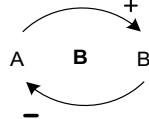

The letter **B** in the middle of a loop indicates that the loop is *balancing* and moves the system in the direction *towards* equilibrium or a fluctuation around equilibrium point.

*Figure 1. Weathering processes were mapped using systems analysis and by drawing causal loop diagrams (CLD) for the process and the whole system of the weathering process. This is a standard procedure in model building (Sverdrup and Stiernquist 2002, Sverdrup et al., 2018).*

## 3. Methodology

The methods used in this study have their basis in terrestrial ecosystems system analysis and ecosystems system dynamics as described by Sverdrup and Stiernquist (2002) and in general on system dynamics theory in Sverdrup et al., (2018). The main tools employed are the standard methods of system analysis and integrated system dynamics modelling (Forrester 1961, 1969, 1971, Meadows et al., 1972, 1974,





1992, 2005, Roberts et al., 1982, Senge 1990, Bossel 1998, Haraldsson and Sverdrup 2005, Haraldsson et al., 2006, Sverdrup and Stiernquist 20002, Sverdrup et al., 2018). The overall system is analysed using stock-and-flow charts and causal loop diagrams (Sverdrup et al., 2002). The learning loop was used as the adaptive learning procedure in past studies (Senge 1990, Kim 1992, Senge et al., 2008, Sverdrup et al., 2018). The conceptual model must be clearly defined and constructed before any computational work can be undertaken. It is fundamental to understand that the causal understanding is the model. Systems analysis produces a causal loop diagram (CLD) linking causes, effects, and feedbacks among the processes in terms of causalities and flows (Albin 1997, Sverdrup et al., 2018, Kim 1992). These CLD need to be internally consistent. A summary of this approach is provided in Figure 1. A causal loop diagram is thus a map of the underlying differential equations describing the evolution of the system. Mass- or energy flow charts and the causal loop diagram uniquely define the system. The ForSAFE model with its integrated weathering model used in this study is not calibrated on large amounts of system output data (Sverdrup and Warfvinge 1992, Sverdrup et al., 2018). Instead, the underlying system causal linkages and the mass balances, lead to characteristic equations that are parameterized using independent system properties, initial states and boundary conditions (Sverdrup et al., 2018).

## 4. Earlier development work and background

**4.1.** Critical to developing a database describing mineral dissolution rates is that it is coupled together into a comprehensive model that can account for the large number of processes that affect rates in the field. From the beginning, the weathering kinetics sub-model was developed and incorporated into the PROFILE model. This sub-model was parameterized using laboratory kinetics and applied to field conditions on a plot scale and on a regional scale for Sweden (Sverdrup 1990, Sverdrup and Wafvinge 1988a,b, 1992, 1995, Warfvinge and Sverdrup 1992, 1993). This sub-model was subsequently coupled into a biogeochemical ecosystem model, linking solute transport, soil chemistry, weathering, ion exchange, hydrology and biological interactions with microbiology and forest plants, called the SAFE model (Sverdrup et al, 1995). The steady-state model PROFILE and the dynamic variant SAFE, was further developed into ForSAFE and ForSAFE-VEG with full ecosystems subroutines, and full base cation nutrients, phosphorus, nitrogen and carbon cycles (Sverdrup and Warfvinge 1996, Sverdrup et al., 2005, 2007, 2008, 2012, 2014, Belyazid et al., 2005, 2007, 2008, 2010, 2011a,b, 2014, McDonnel et al., 2014, 2015, Bonten et al., 2014, Probst et al., 2014, Rizzetto et al., 2017). A description of the original weathering kinetics sub-model was published by Sverdrup (1990). However, much additional experimental data has been obtained since.

## 3.2. Weathering under field conditions

The dissolution of primary minerals at ambient temperature and pressure is irreversible with the exceptions of a few simple chloride and sulphate salts and a few carbonates (Sverdrup 1990). Such irreversible reactions do not attain equilibrium in near to ambient temperate systems. A formulation based on transition state theory for the formation of activated surface complexes that decay irreversibly was developed by (Sverdrup 1985, Sverdrup and Warfvinge 1987, 1988a,b, 1992, Sverdrup 1990) and has been the basis for the further developments. Removal of ions takes place through precipitation of amorphous secondary phases, solute transport and uptake to trees and ground vegetation. The modelling of weathering under field conditions can only be performed with an integrated ecosystems model where mineral reaction rates are coupled to solute transport, ion exchange, plant nutrient uptake, organic matter decomposition and nitrogen transformations have been included (Sverdrup and Warfvinge 1988a,b, Sverdrup 1990, Akselsson et al., 2006, 2005, 2004, Sverdrup et al., 1990, 1995, 2017). A comparison of calculated and observed weathering rates shown in Figure 2, demonstrates this approach can reproduce within ±5% of the observed rates across 4 orders of magnitude for the upper unsaturated parts of a soil (Sverdrup and Warfvinge 1992, Barkman et al., 1999, Jönsson et al., 1995, Belyazid 2005, Kurz et al., 1998a,b). Further comparisons of computed and calculated rates made with these models for field tests at Gårdsjön, Sweden and at various sites were published by Sverdrup et al. (1988a,b, 1993, 1995, 1996, 1998, 2010), Sverdrup (1990, 2009), Sverdrup and Alveteg (1998), Rietz (1995) and Warfvinge et al., (1996), and Holmqvist et al., (2003, 2002). In addition, several other authors tested this approach independently (In the United States; Kolka et al 1996, Phelan et al., 2014, in Scotland; Langan et al. 2006, in Germany; Becker 2002, in New Zealand: Zabowski et al., 2007. tests on controlled experiments



with granite slabs in the Swedish nuclear waste storage assessment research programme at Göteborg by Claesson-Nyström and Andersson 1996, in Swedish soil profiles; Lång 1998). Gunnar Jacks in KTH, Stockholm put these models to several blind test of the alteration of blank granite surfaces used for ancient rock carvings and controlled mini-catchments (Jacks, unpublished 1990). In each case a close correspondence was observed in calculated as compared to the field weathering rates.

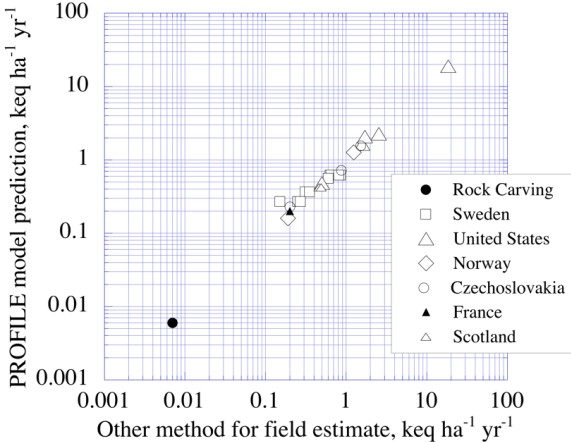

*Figure 2. Comparison of weathering rates calculated using the original PROFILE model with corresponding rates obtained from field observations of the upper undersaturated parts of soils. Rates shown were reported or compiled by Sverdrup and Warfvinge (1988a,b, 1991, 1992, 1993, 1995, 1998), Sverdrup (1990), Sverdrup et al. (1990, 1998), Hettelingh et al. (1992), Barkmann et al. (1999), Holmqvist et al. (2003).*

In 1988, these various models were used to map the weathering rates of the upper 0.5 meter of forest soils of Sweden, based on a regional grid sampling. The first weathering rate map was based on 28 sites where complete data were collected and extrapolated over the whole country using geological maps (Sverdrup and Warfvinge 1988a,b). This map was later enlarged to 1,306 sites and aligned in distinct geological provinces (Warfvinge and Sverdrup 1993, 1995). The database was subsequently extended to 1,884 forested sites, and finally this was expanded through a five-year sampling and analysis program within the Swedish Forest Inventory soil sampling program to approximately first 17,600 forest soil samples and finally to 27,500 forest soil sites across Sweden (Sverdrup and Warfvinge 1988a,b, 1992, Warfvinge et al., 1992, Warfvinge and Sverdrup 1995, Alveteg et al., 1996, 1998, 2000, Akselsson et al., 2004, 2005, 2006, 2007a,b,c, 2018, 2016, Lång 1995). These results were later complemented with about 3,000 additional sites across the agricultural soils. Later the weathering rates of other countries were mapped for the forest soils of Switzerland (Kurz et al., 1998a,b, 2001), France (Probst et al., 2015, Rizzetto et al., 2016a,b, Gaudio et al., 2015), China (Duan et al., 2002), Finland (Sverdrup et al., 1992) and Denmark (Sverdrup et al., 1992), Maryland (Sverdrup et al., 1996), North-western Russia and Far East Siberia (Semenov et al., 2000), Pennsylvania (Phelan et al., 2014, 2016), New York, Maine, Vermont (Sverdrup et al., 2014, Belyazid et al., 2015), New Hampshire (Sverdrup et al., 2012, Belyazid et al., 2015), Madrid Country (Ballesta et al., 1996), Scotland (Langan et al., 1996), Slovakia (Zavodski et al., 1995), and Poland (Malek et al., 2005). Further reports on regional use are available in the UN/EC CCE Annual Reports on mapping critical loads for the years 1995-2018. Further contributions to the developments of these models were made from scientists located at the Institute of Ecology and Lund University, in Bern, Switzerland, at the department of Soil Sciences, Swedish Agricultural University, and at the Physical Geography department of Stockholm University. The weathering rate map of the upper 0.5 meter of forest soils of Sweden is displayed in Figure 3. The grid size is 8.2 km$^2$ or approximately a 3x3km grid in the forested area (Akselsson et al., 2006, 2005, 2004, 2016, Sverdrup et al., 2017). Tests in many other parts of the world, suggests that the model is applicable to the unsaturated zone of any freely draining soil.





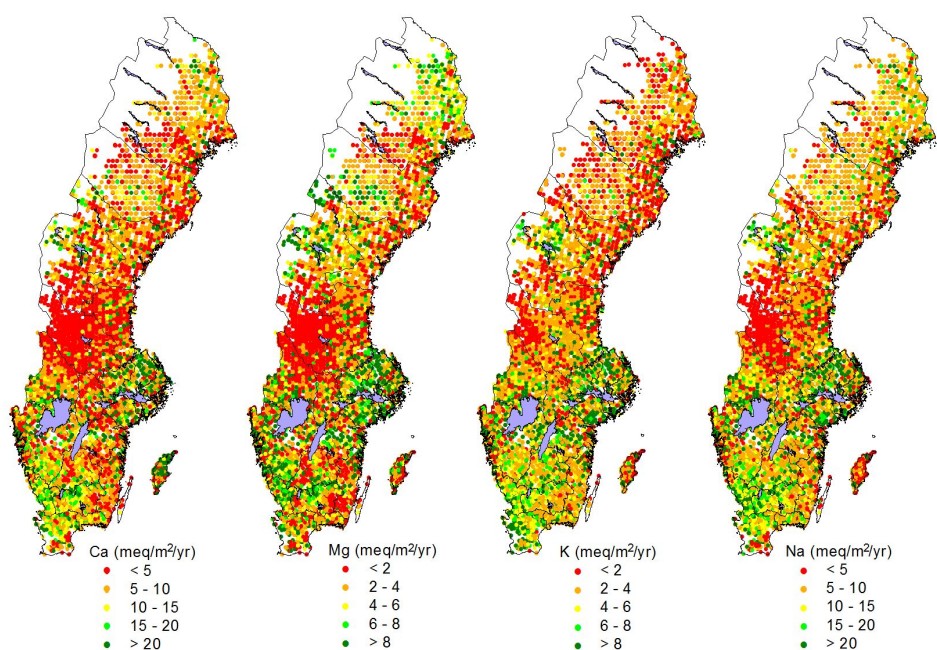

*Figure 3. Map of base cation release rates from chemical weathering of soil minerals in the upper 0.5 m of the soil in Sweden using the PROFILE model. The model accurately reproduces weathering rates in the upper soil layers, and provides useful estimates for soils of up to 1 meter in thickness. The map was created by Dr. Cecilia Akselsson at Lund University for Swedish forest sustainability assessments and critical loads for acid depositions (Akselsson et al., 2006, 2005, 2004, 2016, Sverdrup et al., 2017).*

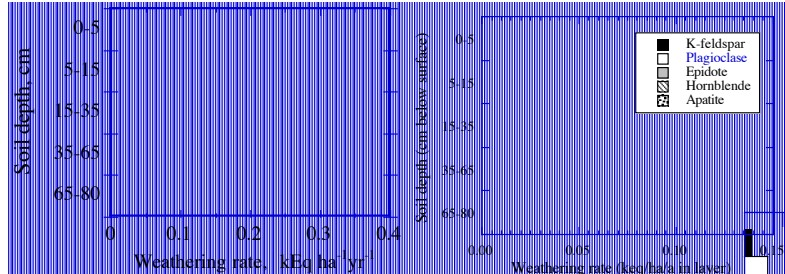

*Figure 4. The diagram shows the weathering rate distributed among minerals, the diagram to the right shows the total rate, plotted as the sum of base cations released to the aqueous phase as a function of depth down a soil profile. The diagram to the left shows how selected minerals contribute to this overall rate. The site is catchment F1 at the Gårdsjön Research site, Sweden (Adapted from Sverdrup and Warfvinge 1992, 1995).*

Figure 4 shows an example from earlier results for the Gårdsjön research site in Sweden (Sverdrup et al., 1992, 1993, 1998). The diagram shows the weathering rate distributed among minerals, and the total rate as a function of depth down a soil profile. The example shows the weathering rate at catchment F1 at the Gårdsjön Research site, Sweden (Sverdrup et al., 1992, 1993, 1996). The research site at Gårdsjön, near Göteborg, Sweden has played a key role in the development of our biogeochemical ecosystem models. The research site is one of Sweden's most important field research sites for soils, soil chemistry, material fluxes, geology, mineralogy, ecology, forestry and environmental pollution




research, with nearly all aspects excellently documented and recorded for the last 40 years (Hultberg et al., 2007). Here the models were tested, adapted and used for assessments. Differences in calculated and observed results became evident when calculating weathering rates for deeper layers. Notably the model overestimate the weathering rate at depths below 1-1.5 meters.

Figure 5 shows an example of a soil weathering simulation of the weathering rate at Niwot Ridge, Rocky Mountain National Park, Colorado for four different environmental pollution scenarios with background acid deposition, current policy, no pollution control and elevated temperature from climate change. The weathering rate is reduced under the climate change scenario. The weathering rate is somewhat increased by the increase in temperature, but more reduced by reduced rainfall leading to drier soils at the site. The ForSAFE model was used with a daily time step, estimating a weathering rate every day. The time-step is numerically determined by the stiffness of the differential equations in the system. The timestep is set automatically by the model numeric routine and thus is variable and is optimized during integration. Under conditions where short-term changes happen, the timestep may be on the scale of hours.

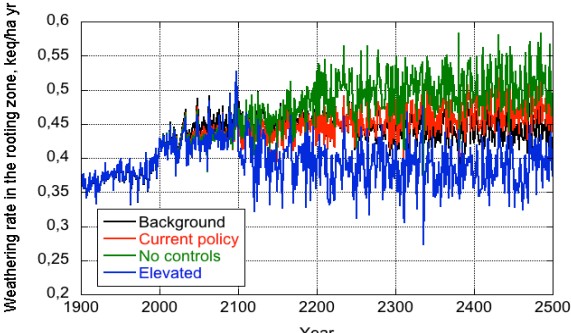

*Figure 5. Example of a soil weathering rate calculation for Niwot Ridge, Rock Mountain National Park, Colorado for four different environmental pollution scenarios and their effect on the ecosystems (trees and biodiversity): 1) background acid deposition from sulphur and nitrogen, 2) current policy, 3) no pollution control and 4) elevated temperature. The weathering rate was extracted from the simulations to assess the site for pollution control. In this case the ForSAFE model was used with a daily time step to estimate daily weathering rates (Sverdrup et al., 2014, McDonnel et al., 2017, Belyazid et al., 2019).*

### 3.3. Weathering Model overview

A number of computational weathering models based on this approach have been developed over the years. The PROFILE model was developed for critical load assessments, forestry sustainability assessments, and estimation of field weathering rates. The SAFE and later ForSAFE models are dynamic models for making dynamic terrestrial ecosystem assessments. The PROFILE model is the steady-state version of the SAFE model. Both models were first completed in 1987 (Sverdrup et al., 1987a,b, Sverdrup and Warfvinge 1988a,b). To clarify these models and their interconnections the following list is provided, which also lists the key scientists involved in their research and development:

1. **Steady-state weathering rate models**
   a. 1987-1995; Warfvinge P. and Sverdrup, H.; The single site version of the **PROFILE** model for the calculation and mapping of critical loads and rates of field chemical weathering was developed. It is a widely used soil model, validated and used operationally in more than 50 countries worldwide. It uses laboratory generated kinetic models and coefficients to predict field weathering rates. The interface software for PROFILE became outdated, thus, this version is no longer available.
   b. 1992-present; Sverdrup, H., Warfvinge, P., Alveteg, M., Walse, C., Kurz, P., Posch, M., Belyazid, S.; The code **RegionalPROFILE** was developed. This code is a regionalized version of the PROFILE model, used for creating weathering rate maps for soils and catchments across regions and countries, as well as to estimate critical loads for forest soils. Updated versions of the code are available upon request from Sverdrup, Akselsson or



Belyazid.

c. 2000; Sverdrup, H. and Alveteg, M., The **CLAY-PROFILE** code was developed. This model was made for volcanic and clayey agricultural soils. This code is no longer operable. Archived, the code is available upon written request from Sverdrup or Belyazid.

2. **Dynamic weathering models**

a. 1987-2008; Warfvinge P., Sverdrup, H., Alveteg, M., Walse, C., Martinsson, L.: The **SAFE** model and its helper routine **MakeDep** were created. **SAFE** is a generally applicable dynamic soil chemistry and acidification model. This tool is used worldwide for acidification research, forest sustainability assessments and for mapping critical loads.

b. 1995-1996; Rietz, F., Sverdrup, H., Warfvinge, P.; The **SkogsSAFE** model was developed. This long-term perspective dynamic model simulates soil genesis, mineralogy dynamics, soil chemistry and base cation release from chemical weathering in soils over time since the most recent glaciation (14,000 years ago to present) (Rietz 1995, Warfvinge et al., 1996). This code is written in FORTRAN. This code and its databases are available upon written request from Sverdrup.

c. 1996-2004; Sverdrup, H., Wallman P., Belyazid, S., Alveteg, M., Walse, C., Martinsson, L.: These scientists developed **ForSAFE,** an integrated biogechemical forest ecosystem model for growth, nitrogen and carbon cycling. This code is written in FORTRAN code, and the code is available upon written request from Sverdrup or Belyazid.

3. **Regional mineralogy estimation**

a. 1990; Sverdrup, H., Melkerud, P. A., Kurz, D.: The **UPPSALA** model was developed for the reconstruction of soil mineralogy from soil total analysis data. This model is run in a spreadsheet. It is available upon written request from Sverdrup.

b. 1998; Sverdrup, H. and Erdogan, B. The **Turkey** mineral depletion model (TMD) was developed. This model estimates soil mineralogy from bedrock geology and estimates of soil age. This code is written in STELLA®. It is archived and available upon written request from Sverdrup.

c. 2005-2010; Posch, M., Kurz, D., Alveteg, M., Akselsson, C., Eggenberger, U., Holmqvist, J; 2007 **A2M**, a model to quantify mineralogy from geochemical analyses was developed. This code is available on-line from doi:10.1016/j.cageo.2006.08.007, https://dl.acm.org/citation.cfm?id=1231715or from Kurz or Akselsson (Posch et al., 2006, 2007).

These models are not commercial products. They do not have ready-made handbooks (only the early single site PROFILE models had a good users interface and a user's manual). The models are available, but the best option to learn how to run these get training from the contact scientists in how to operate the models and how to set up the input data for a site or a region. The core code is written in FORTRAN.

**4. Theory**

The model described here originates from the kinetic weathering model first proposed by Sverdrup and Warfvinge (1987a,b, 1988a,b, 1992a, 1995) and Sverdrup (1990), but numerous features have been added since. Some of the updates have been described in later studies (Akselsson et al., 2005, 2005, 2006, 2007, Alveteg et al., 2000, Kurz et al., 1998a,b, Sverdrup et al., 1997, 2002, 2008), and the latest updates have been done specifically for this study. New weathering rate data published over the past 25 years have been regressed and new temperature dependencies and modifications of some rate coefficients has resulted (Sverdrup 2010, Sverdrup et al., 1998, Rizzetto et al., 2016, Holmqvist et al., 2002, 2003). The weathering sub-model in ForSAFE requires no calibration. It originates from the regression of laboratory based experiments. The mineralogy and surface area inputs are based on site measurements, and in general are not adjustable parameters.. Some of parameters can be challenging to measure, such as some primary minerals with low soil content (apatite, epidote, pyroxene, amphiboles, garnets accurate to 0.1%), or the determination of surface area estimates. However, getting accurate field estimates of the weathering rates is also challenging, as it requires making many assumptions, and has limitations on the accuracy of the obtained estimate. Thus, we are comparing uncertain model estimates with equally or more uncertain field estimates at the best (Sverdrup et al., 1998).

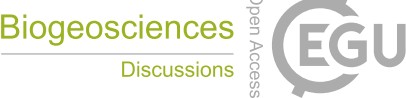



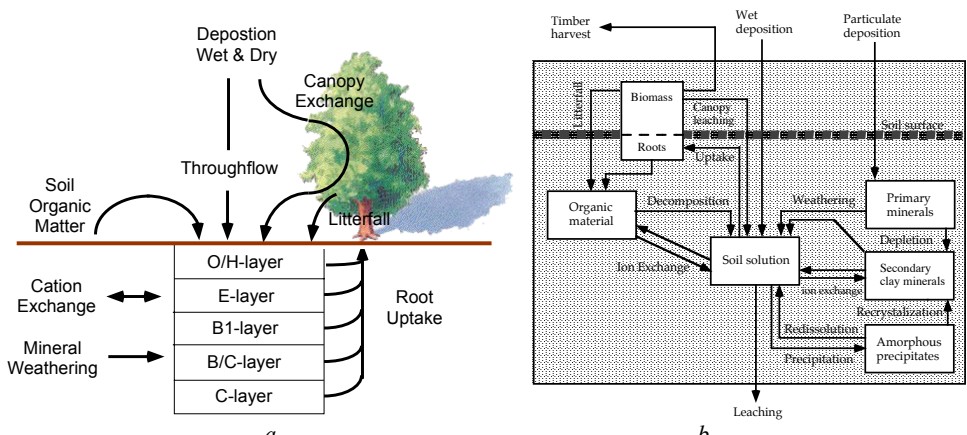

*Figure 6. Overview of the PROFILE model. The original PROFILE model operates with a number of layers, and a vertical percolation of water. A set of processes take place in every layer. (b) A look inside PROFILE, showing how weathering is connected with other ecosystem processes (Sverdrup and Warfvinge 1995).*

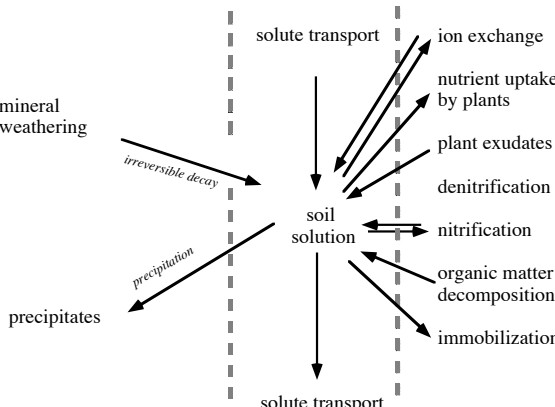

*Figure 7. Different soil processes communicate with the weathering processes via the soil solution. (Sverdrup et al., 2002).*

The main ForSAFE model is calibrated on two variables, 1) the initial base cation saturation in the fluid phase is adjusted to its an initial value at the starting simulation time to insure the cation concentrations are consistent with the observed base cation saturation, and 2) the initial stock of nitrogen in the soil is adjusted to match that currently observed in the system. Once this main model calibration is complete, the ForSAFE model can calculate weathering rates from its kinetics database (Sverdrup et al., 1996, 1998, 2007) and the soil inputs. The ForSAFE model must be provided site specific characteristics like mineralogy of the soil, soil layering, soil density, soil mineral surface areas, hydrological characteristics, site temperature, ecosystem characteristics (trees, plants), typical inputs of rain, chemistry of that rain and the amount of the  major deposited pollutants.

### 4.1. Defining chemical weathering

We have had a utilitarian view of the chemical weathering process.  Weathering is a provider of neutralization for acids (neutralizing all or part of acid rain) and as a provider of nutrients for vegetation ($Ca^{2+}$, $Mg^{2+}$, $K^+$, $PO_4$) (Sverdrup 1990, Sverdrup and Warfvinge 1995, Sverdrup et al., 2002). Thus weathering rates are defined as "base cation release rates from the chemical weathering of minerals", "plant nutrient base cation release from the chemical weathering of minerals"  or "the rate of acid neutralization by chemical weathering of soil minerals". Only secondarily were we interested in loss of





minerals and soil profile development (Rietz 1995, Warfvinge et al., 1996, Sverdrup et al., 1996, 2002). Thus, the weathering rates have been expressed as the sum of the release rates of base cations ($Ca^{2+}$, $Mg^{2+}$, $K^+$, $Na^+$) from the process. This is linked to the destruction of the mineral, though results are generally expressed in these terms.

### 4.2. Mineral weathering rates

The weathering rate of a mineral, r, defined here as its dissolution rate, is assumed to stem from the sum of 5 simultaneous chemical reactions, one involving the mineral surface and either aqueous $H^+$, $H_2O$, $OH^-$, organic acid ligands, or $CO_2$. Assuming that the reactions occur at distinct active mineral surface sites, they can be summed linearly in accord with (Sverdrup 1990, Sverdrup and Warfvinge 1995):

$$R_W = \sum_{j=1}^{\text{Minerals}} A_j * \sum_{i=1}^{\substack{\text{Dissolution}\\\text{reactions}}} r_i \qquad (1a)$$

where $R_W$ stands for the soil weathering rate in a single soil layer. $A_j$ refers to the soil mineral surface available for dissolution for each mineral j considered, $r_i$ designates the rate of the individual chemical reactions i. If some reactions occupy the same active mineral surface sites, the expression given above would change to a quadratic sum. Note that the results of the two equations are quite similar, so that the importance of knowing if several reactions operate of the same surface site is relatively small. For the whole soil profile, we get:

$$R_{Soil} = \sum_{s=1}^{\text{Layers}} R_{W,s} \qquad (2)$$

where $R_{Soil}$ denotes the weathering rate in the whole soil profile, and s represents the layer number. Evidence that the $H^+$, $H_2O$ and $OH^-$ reactions take place at distinct surface sites has been reviewed by Sverdrup (1990) and again by Holmqvist et al., (2003). The $H_2O$, the organic reaction and the $CO_2$ reactions may occur at the same sites, but considering the available data, we have assumed that they occur at distinct sites and thus favour a linear sum of rates. More on these assumptions have been reported by Sverdrup (1990), Sverdrup and Warfvinge (1995), and Holmqvist et al. (2002, 2003).

### 4.3. Field weathering rates

To estimate field weathering rates using laboratory determined kinetic coefficients, an ecosystem model is required to scale the process to field conditions. This ecosystem model includes effects of climate, soil morphology, plants, trees, microbiology in the soil and fungi (Lin et al., 2017, Smits and Wallander 2016, Smits et al., 2014). An ecosystem model is incorporated within PROFILE and ForSAFE (Sverdrup and Warfvinge 1988a,b, 1991, 1992, 1993, 1995, 1998, Sverdrup 1990, Sverdrup et al., 1998, Hettelingh et al., 1992, Barkmann et al., 1999, Holmqvist et al., 2003, Barkman et al., 1999). Figure 6 shows how the steady-state model PROFILE was configured (Sverdrup and Warfvinge 1988a,b, 1992, 1993, Sverdrup and Alveteg 1998). In the dynamic integrated terrestrial ecosystem assessment model ForSAFE-VEG, the system evolution over time takes account of interactions with a living biosphere, organic matter turnover and ion exchange. Further details of these models can be found in the literature (Sverdrup et al., 1987, 1995, 1996a,b, 1998, 2007, 2017, 2014, 2014, 2016, 2017, 2019, Wallman et al., 2002, 2003, Zancchi et al., 2014, 2016a,b, Belyazid et al., 2017, 2018).

To estimate field weathering rates, each reaction i for every mineral j is corrected for the field site temperature and for the partial wetting of the soil (Sverdrup 1990, Sverdrup and Warfvinge 1995, Sverdrup and Alveteg 1998) in accord with:

$$R_W = h(\theta) * \sum_{j=1}^{\text{Minerals}} A_j * \sum_{i=1}^{\substack{\text{Dissolution}\\\text{reactions}}} \left( r_i * g_{i,j}(T) \right) \qquad (3)$$



where θ stands for the fraction of the soil mineral surfaces wetted, $A_j$ designates the surface area of the mineral j, h(θ) refers to a wetting function for the mineral material and T signifies the soil temperature in centigrade. $g_{ij}(T)$ corresponds to the temperature adjustment function for reaction i of mineral j. $r_i$ denotes the reaction rate of dissolution reaction i. This adjustment is based on the Arrhenius equation and takes account of the difference in rates between the temperature of the field site and that of the parameter database, which was set at 8°C (Sverdrup 1990). Figure 9 shows the reaction causal loop diagram for silicate minerals in the soil (Sverdrup 1990, Sverdrup and Warfvinge 1995). This diagram shows how the mineral weathering process communicates with other biogeochemical processes in a terrestrial ecosystem. The causal loop diagram is a graphical display of the differential balances in the system. Together with the flow charts, they define the system. The process has several intermediate equilibrium steps, but pass an irreversible dissolution threshold (Figure 10) The irreversible step makes the whole process irreversible. The reaction products exert a negative effect on the amount of activated complex that can decay, thus they retard the dissolution reaction. But once the activated complex has formed, it has a constant decay rate, set by quantum mechanics (Sverdrup 1990, Sverdrup and Warfvinge 1995). The full derivation of the rate equations, starting from the elementary chemical reactions and the decay of the surface complexes in transitional state has been reviewed by (Sverdrup 1990, Sverdrup and Warfvinge 1995).

### 4.4 The chemical reaction kinetics

As stated above five reactions are assumed to contribute to the total chemical weathering rate of a silicate mineral in soils (Sverdrup 1990, 2009, Sverdrup and Warfvinge 1995):

1. The reaction between the mineral surface and the aqueous hydrogen ion
2. The reaction between the mineral surface and the water molecule
3. The reaction between the mineral surface and aqueous carbon dioxide
4. The reaction between the mineral surface and aqueous organic acid ligands
5. The reaction between the mineral surface and the aqueous hydroxy ion

Reactions 1-4 in the list above were included in earlier versions of the PROFILE and ForSAFE mineral dissolution rate models (Sverdrup 1990, Sverdrup and Warfvinge 1995). This original model has been enlarged to include reaction 5.

The reaction of the mineral surface with the aqueous $H^+$ ion, reaction 1, is considered part of the reaction with the $H^+$ reaction regardless of the source of $H^+$ (Figures 8 and 10). Both $CO_2$ and organic acid can change the fluid pH, and this is accounted for in the $H^+$ reaction. Figure 8 shows the reaction pathway through the $H^+$ reaction, adapted after Sverdrup (1990). The solid residuals rearrange to secondary minerals. Amorphous phases may also precipitate from solution. These can slowly recrystallize to secondary minerals. This has been generalized in Figure 9.

Reaction number 4 with organic acid ligands and the mineral surface contains at least two distinct contributions one from fast and one from slower reacting organic acid ligands (Sverdrup 1990). We have simplified this to one generic rate equation that could be parameterized for some minerals (feldspar, olivine, pyroxenes, hornblende, apatite; Sverdrup et al., 1990, later literature has extended the list somewhat). The importance of organic acids for weathering has been frequently over estimated in the literature, and several claims of strong effects of organic acids (For a review see Smits and Wallander 2016, Smits et al., 2014, Sverdrup 1990, 2009 but also Keegan and Laskow-Lehey 2014 on why these claims have been so persistent). The highest concentration of organic acids occur in the upper soil layers, where the mineral content is lower. As the mineral contents increase with depth, the concentrations of organic acids reach low levels with only marginal effect on the overall weathering rate (Sverdrup 2009).

Organic acids in soils are mostly sourced from soil organic matter decomposition. Trees, soil fungi and mycorrhiza do not have the ability to increase the weathering rate significantly (See Sverdrup 1990, 2009, Sverdrup and Warfvinge 1992, Warfvinge and Sverdrup 1993 for details, kinetic expressions and data underpinning this, see Smits and Wallander 2016 and Smits et al., 2014 on the subject concerning apatite). Trees and vegetation can indirectly affect the weathering rates when they take up Ca, Mg, K as nutrients, and thereby removing weathering rate products that can slow mineral dissolution. Decomposition of plant debris and soil organic matter produce organic acids that may react with the minerals. This effect is passive, and does not occur not by design of the plants (See Smits and





Wallander 2016 and Smits et al., 2014 for measurements, Keegan and Laskow-Lehey 2014 for some social aspects and Sverdrup 2009 for a further analysis from a systemic perspective).

Fluorides form soluble complexes in water with aluminium and silicates. The reaction of the mineral surface with fluoride anions forms a strong reactions, but this occurs very rarely as the fluoride concentrations are very low. The fluoride reaction has been ignored for most soils in natural terrestrial ecosystems, as this would cause an unnecessary complication of the aluminium and silicate chemistry.

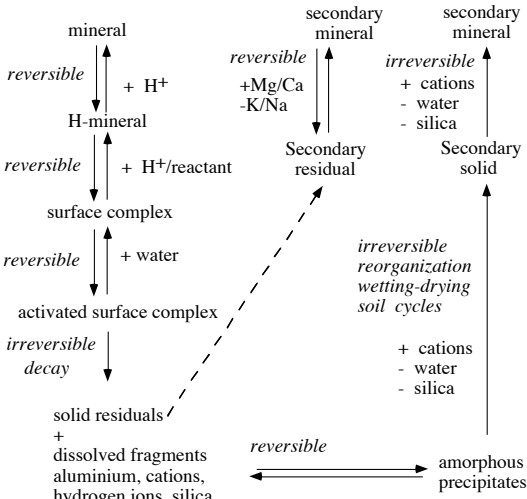

*Figure 8. The reaction pathway through the H⁺ reaction passes over several reversible steps that change the surface sites and create an unstable surface complex; the Transition State Surface Complex that will decay irreversibly. Note that the process is irreversible, and thus cannot go backwards. The mineral may dissolve completely, be altered to an alteration mineral or form precipitates that slowly recrystalize to secondary solid phases.*

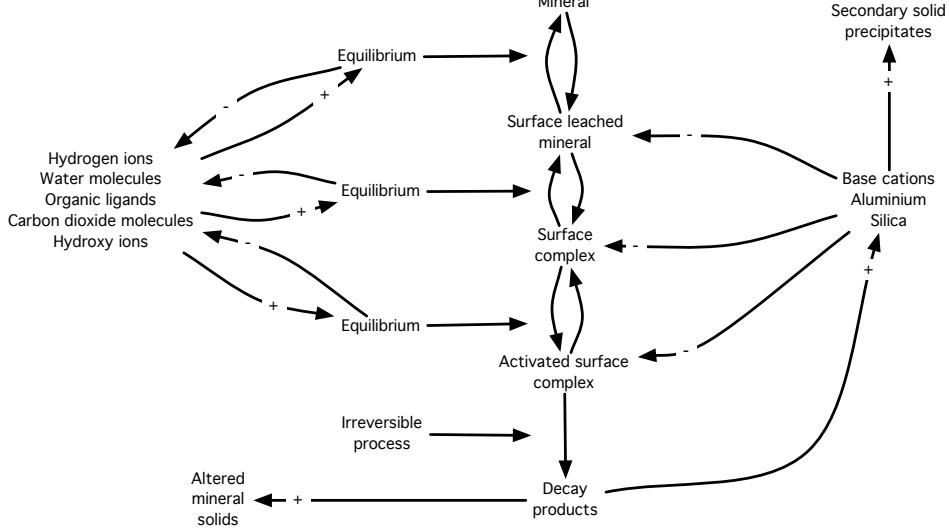

*Figure 9. Reaction pathway for silicate minerals in soils according to Transition State Theory as implemented by the authors (See Sverdrup 1990, Sverdrup and Warfvinge 1995 for a full explanation).*




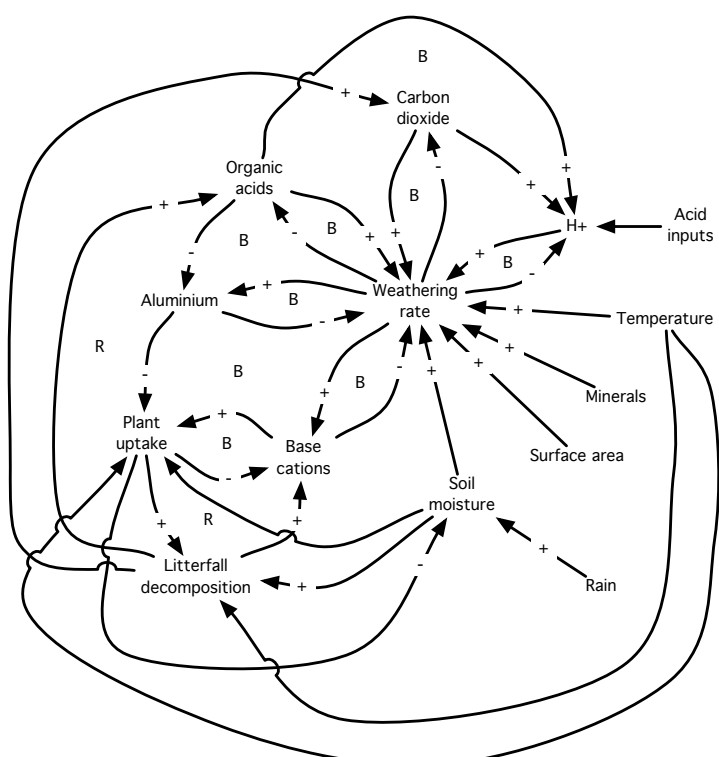

*Figure 10. The partial causal loop diagram for the weathering process in a soil. See Sverdrup et al., 2018 for a full explanation of causal loop diagrams and their use in modelling.*

The dissolution rate per surface area of a mineral is thus consistent with (Sverdrup and Warfvinge 1988, 1992):

$$r_{Total} = r_{H^+} + r_{H_2O} + r_{CO_2} + r_R \qquad (4)$$

The mineral dissolution kinetic equation for the 4 individual reactions applied in the original PROFILE model was the simplified version of the full kinetic expression based on the Transition State Theory applied to silicate chemical weathering (see Sverdrup 1990, Sverdrup and Warfvinge 1995):

$$r = k_H * \frac{[H^+]^{n_H}}{f_H} + \frac{k_{H_2O}}{f_{H_2O}} + k_{CO_2} * P_{CO_2}^{n_{CO_2}} * \frac{1}{f_{CO_2}} + k_R * \frac{[R]^{n_R}}{1 + K_{Org} * [R]^{n_R}} * \frac{1}{f_R} \quad (5)$$

where the different n designate reaction orders. The different $k_H$, $k_{H2O}$, $k_{CO2}$, $k_R$ stand for rate coefficients. The different $f_{H+}$, $f_{H2O}$, $f_{CO2}$, $f_R$, $f_{OH}$ signify retarding functions defined by (Sverdrup 1990, Sverdrup and Warfvinge 1992, Warfvinge and Sverdrup 1993, Sverdrup and Warfvinge 1995):

$$f_{H^+} = \left(1 + \frac{[BC]}{C_{BC,H}}\right)^{x_H} * \left(1 + \frac{[Al^{3+}]}{C_{Al,H}}\right)^{y_H} \qquad (6)$$

$$f_{H_2O} = \left(1 + \frac{[BC]}{C_{BC,H_2O}}\right)^{x_{H_2O}} * \left(1 + \frac{[Al^{3+}]}{C_{Al,H_2O}}\right)^{y_{H_2O}} \qquad (7)$$





$$f_{CO_2} = \left(1 + \frac{[BC]}{C_{BC,CO_2}}\right)^{x_{CO_2}} * \left(1 + \frac{[Al^{3+}]}{C_{Al,CO_2}}\right)^{y_{CO_2}} \qquad (8)$$

$$f_R = \left(1 + \frac{[BC]}{C_{BC,R}}\right)^{x_R} * \left(1 + \frac{[Al^{3+}]}{C_{Al,R}}\right)^{y_R} \qquad (9)$$

$$f_{OH^-} = \left(1 + \frac{[BC]}{C_{BC,OH}}\right)^{x_{OH}} * \left(1 + \frac{[Al^{3+}]}{C_{Al,OH}}\right)^{y_{OH}} \qquad (10)$$

Take note that the retardation functions represent molecular mechanisms that slow the reaction by forming less active surface complexes (Sverdrup 1990, Sverdrup and Warfvinge 1995), and that it is not a solution saturation term. Saturation with the liquid phase requires the assumption of reversibility and the dissolution of these silicate minerals is not reversible under normal soil conditions. Such an assumption is reasonable under high pressure and high temperature, but not valid under soil conditions, or at normal room temperature and pressure in a chemical laboratory. The process is irreversible, thus any equilibrium assumption is invalid (Denbigh 1971).

### 4.5. The updated kinetics equation

These original equations have been enlarged with all terms fully expressed, including the OH⁻-reaction and the brakes from silicate on all reactions in the present study. The complete equation adopted in this study for mineral dissolution rates per unit surface area is consistent with

$$r_{Total} = r_{H^+} + r_{H_2O} + r_{CO_2} + r_{R+} + r_{OH^-} \qquad (11)$$

The full kinetic equation for all 5 reactions is (Sverdrup 1990, Sverdrup and Warfvinge 1995):

$$r = k_H * \frac{[H^+]^{n_H}}{f_H} + \frac{k_{H_2O}}{f_{H_2O}} + k_{CO_2} * \frac{P_{CO_2}^{n_{CO_2}}}{1 + K_{CO_2} * P_{CO_2}^{n_{CO_2}}} * \frac{1}{f_{CO_2}}$$
$$+ k_R * \frac{[R]^{n_R}}{1 + K_{Org} * [R]^{n_R}} * \frac{1}{f_R} + k_{OH} * \frac{[OH^-]^{n_{OH}}}{f_{OH}} \qquad (12)$$

For most minerals, the effect of reaction products is the strongest for aluminium at pH < 7, followed by silica and base cations. At pH > 8, the retarding effect is strongest from silica and base cations, and less pronounced for aluminium (Sverdrup 1990). Before applying Equation (12) a number of new adaptions have been carried out as described below.

### 4.6. Retardation of mineral dissolution rates by organic ligands

The original formula for the effect of organic ligands on mineral dissolution rates was (Sverdrup 1990, Sverdrup and Warfvinge 1995):

$$r_{Org} = k_R * \frac{[R]^{n_R}}{1 + [R]^{n_R}} * \frac{1}{f_R} \qquad (13)$$

this has been reformulated to:

$$r_{Org} = k_R * \left(\frac{[R]}{1 + [R]}\right)^{n_R} * \frac{1}{f_R} \qquad (14)$$




The difference in these equations is that the latter contains one additional parameter $[R]_{Limit}$ in $f_R$ that has the effect to set a lower concentration, below which the organic acids have no effect. This equation has been parameterized and used in the final expression provided below. This limit was incorporated into the organic acid ligand retardation function $f_R$ (Smits and Wallander 2016, Smits et al., 2014, Sverdrup 1990, 2009).

**4.7. Retardation of mineral dissolution rates by aqueous $CO_2$**

The main effect of the presence of $CO_2$ is to change the pH of the solution. This effect is accounted for in the model by the chemical solution equilibria, and dealt with in the $H^+$ reaction. This term takes into account the effect of a reaction between the $CO_2$ and the surface. The effect of the presence of aqueous organic species decreases at higher concentrations of organic acids as the surface sites have become saturated with organic acid ligands. We hypothesize that $CO_2$ exhibits the same behaviour. Some data show that $CO_2$ also reacts with mineral surface sites as some type of carbonate ligand (a bicarbonate coordinated towards a cation in the lattice) adsorbed to the surface, setting up a transitional surface complex may decay. The mechanism by which $CO_2$ effects silicate dissolution rates appears to follow the sequence (Sverdrup 1990, Sverdrup and Warfvinge 1995, Brady and Carrol 1994, Golubev et al., 2005, Navarre-Sitchler and Thyne 2007, Berg and Banwart 2000):

1. The $CO_2$ molecule attaches to the mineral surface
2. The $CO_2$ molecule forms a bicarbonate-water-metal complex with the mineral surface on singly coordinated metal cations. Indications are that it may be the $CO_3^{2-}$ ligand that is forming a surface complex.
3. A cation is lifted into the complex (K, Na, Mg, Ca, Fe, etc..)
4. A small fraction of the surface complexes detaches from the surface and the mineral unit dissolves (Decay of the transitional surface complex)

Thus, potentially, there should be an upper concentration limit where additional aqueous $CO_2$ will have no further effect on mineral dissolution rates. This seems to occur between 10 and 50 atmospheres of $CO_2$ partial pressure for mica and chlorites (Drever et al., 1996, Mast and Drever 1987, Hausrath et al., 2009). Some other minerals have indications of a similar behaviour, but this limit remains elusive in terms of parameterization due to lack of data. In addition the dissolution rates of some minerals exhibit no detectable effect of the presence of aqueous $CO_2$, and some are only slightly inhibited by this species. Lagache (1965, 1976), Busenberg and Clemency (1976), Berg and Banwart (2000) and Golubev et al., (2005) reported experiments performed at different $CO_2$ partial pressures between 0 and 26.3 $CO_2$ atmospheres and temperatures between 0 °C and 200 °C. The original equation used by Sverdrup (1990) and Sverdrup and Warfvinge (1995) to describe these data was

$$r_{CO_2} = k_{CO_2} * \frac{P_{CO_2}^{n_{CO_2}}}{1 + K_{CO_2} * P_{CO_2}^{n_{CO_2}}} * \frac{1}{f_{CO_2}} \qquad (15)$$

In this study we use a variation of this equation of the form:

$$r_{CO_2} = k_{CO_2} * \left(\frac{P_{CO_2}}{1 + K_{CO_2} * P_{CO_2}}\right)^{n_{CO_2}} * \frac{1}{f_{CO_2}} \qquad (16)$$

Evidence suggests that the value of $P_{Limit\,CO2}$ is in the range of 5 to 10 atmospheres and $K_{CO2}=0.05$ and $n_{CO2}=0.6$ for albite (Sverdrup 1990). Navarre-Sitchler and Thyne (2007) suggests $n_{CO2}=0.45$, which is for practical purposes the same. Berg and Banwart (2000) suggested $n_{CO2}=0.25$ at low pressures of $CO_2$. As mentioned above, a similar behaviour was observed for mica, biotite and chlorites. Indications are that something similar takes place on the surface of montmorillonite, diaspore, gibbsite, goethite and lepicrocite. There almost no experimental data available that allow the retrieval of the parameters in Equation (14) for other minerals. The effect of increasing aqueous $CO_2$ has been overlooked in most experimental studies.



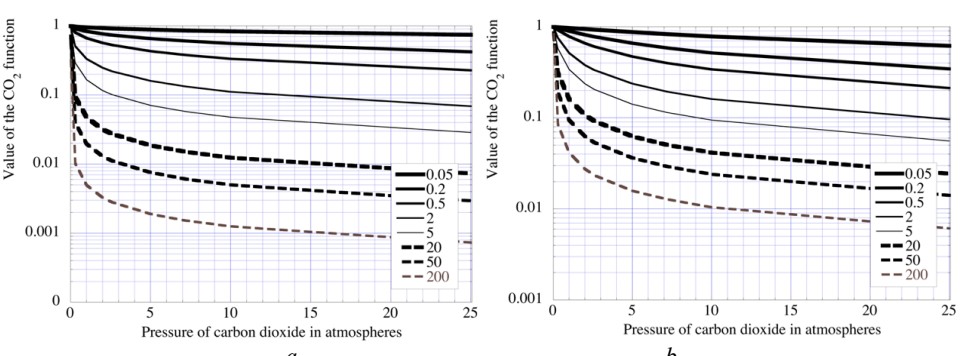

*Figure 11. The calculated effect of aqueous carbon dioxide on mineral dissolution reactions as calculated using Equation 15 in (a) and Equation 16 in (b). See Table 2 for values for different minerals.*

| Table 1. Selection table for parameterization of the parameter z in the silica brakes to the different weathering reactions. | | | | | |
|---|---|---|---|---|---|
| # | Silica brake response group | z-values suggested by the mineral reactions | | | |
| | | $H^+$ | $H_2O$ | $CO_2$ | Organic acids | $OH^-$ |
| 1 | K-Feldspar and sericite | 6 | 2 | 2 | 2 | 1 |
| | Muscovite group and illites | 7 | 3 | 3 | 3 | 2 |
| 2 | Albite | 8 | 4 | 4 | 4 | 3 |
| | Na-rich Plagioclase | 7 | 4 | 4 | 4 | 3 |
| | Ca-rich Plagioclase | 10 | 6 | 6 | 6 | 4 |
| 3 | Biotite group Chlorite group Serpentinite Aluminum-nesosilicates Aluminium pyroxenes Tourmaline group | 16 | 6 | 6 | 6 | 4 |
| 4 | Amphibole group Pyroxene group Epidote group Nesosilicate | 20 32 32 32 | 16 | 16 | 16 | 8 |
| 5 | All other silicates | 32 | 16 | 16 | 16 | 8 |
| 6 | Carbonates | n.a | n.a | n.a | n.a | n.a |

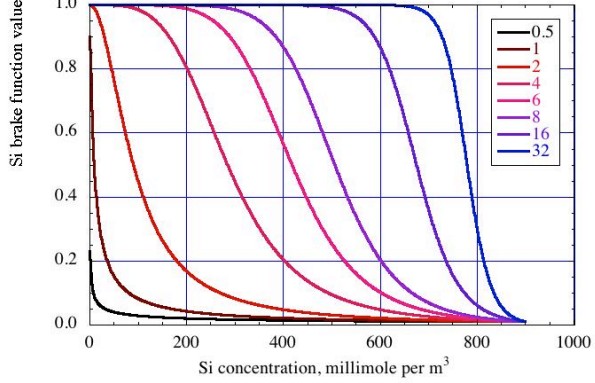

*Figure 12. Calculated effect of dissolved Si on silicate dissolution rates generated using Equation (17) together with $K_{Si}=100$, and the saturation concentration, $C_{Si}=900$ mmol per $m^3$ and the coefficients in listed in Table 1.*




Values calculated of the effect of aqueous $CO_2$ on silicate dissolution rates are illustrated in Figure 11. These calculations suggests that there is a significant saturation of the surface with $CO_2$ at approximately 5 to 10 atmospheres partial pressure of $CO_2$. Data regression suggests that $K_{CO2}$ has a value in the range of 2-20. See Table 1 for the values suggested for different minerals. Note that the values of this parameter are based on minimal supporting experimental data - the available experimental data are few and somewhat incomplete (See Golubev et al., 2005 for a limited but useful assessment). Overall, the effect of $CO_2$ at normal soil conditions is limited. Nevertheless, these results provide a range for model parameter adjustment. The effect of dissolved $CO_2$ on rates may become significant for deep aquifers, subsurface $CO_2$ storage and in industrial high-pressure situations (Sverdrup 1990).

**4.8 The silica retarding function**

An illustrative plot of the effect of aqueous silica on silicate mineral dissolution rates is provided in Figure 12. The equation proposed by the 2016 Ystad Workshop for the retardation effect of dissolved Si on rates was:

$$\frac{1}{f_{Si}} = \frac{1}{1 + K_{Si,i} * \left(\frac{[Si]}{C_{Si}}\right)^{z_{Si}}} \qquad (17)$$

The values $K_{Si,i} = 100$ was chosen to be used, which causes a gradual reduction in the dissolution rate of minerals down to a minimum of approximately 0.9% of the rate unaffected by silica at very high silica concentrations (see Table 1). Figure 13 shows values of the silica brake function as calculated using Equation 17, using the surface constant value, $K_{Si}=100$, and the saturation concentration $C_{Si}=900$ mmol per m$^3$ in Equation 17 together with the coefficients in Table 3. Exponents from $z_{Si} = 0.5$ to 32 in Equation (17) of the silica rate brake are shown in Figure 12.

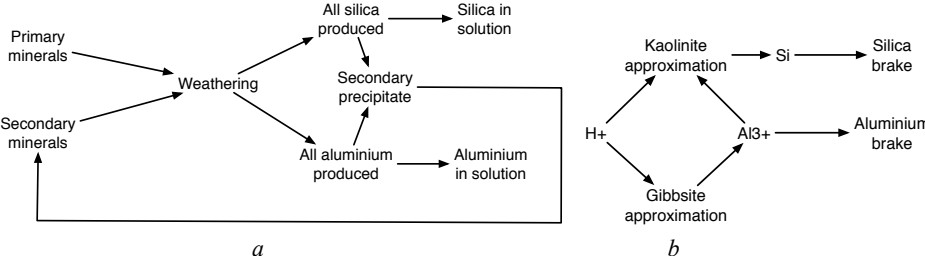

*Figure 13. a) Plot visualizing the fate of silica during the dissolution process. b) Diagram showing how the aluminium and silica concentrations are estimated in the model. The $H^+$ concentration is used with the equation called the "Gibbsite" equation (Eq. 19) to estimate the $Al^{3+}$ concentration in the soil solution. The $H^+$ concentration and the $Al^{3+}$ concentration is used in Equation 21 to estimate the silica concentration that is used in the silica brake on the mineral weathering reactions.*

Figure 13a shows a plot visualizing the fate of silica in the dissolution process. Only a small part of the aqueous aluminium and aqueous silica produced by the dissolution of minerals remain in solution. Most precipitates out as secondary phases. Figure 13b shows how the aluminium and silica concentrations are estimated in the model. We assume that aluminium precipitates out from the solution, controlled by something that appears to be gibbsite-like; it is likely something amorphous of unknown composition, see Alveteg et al. (1995). The "Gibbsite" reaction is:

$$Al^{3+} + 3\ OH^- = Al(OH)_3 \qquad (18)$$

Leading to the "Gibbsite" expression:

$$[Al^{3+}] = K_G * [H^+]^Y \qquad (19)$$





where the exponent Y has a value of 2.4-3. $K_G$ is the Gibbsite coefficient and defined in the critical loads mapping manual (Sverdrup et al., 1990). An expression analogous to the Gibbsite approximation is used to calculate the Si concentration (Equation 22b, below). We assume that the Si will be present as $H_4Si(OH)_4$ in the fluid phase, not upsetting any charge balance constraints. We assume that silica precipitates out, controlled by what that appears to be kaolinite. As such, there is a similar expression for approximating the silica concentration:

$$2\ Al^{3+} +\ 2\ SiO_2 + 6\ OH^- =\ Al_2Si_2O_5(OH)_4 + H_2O \qquad (20)$$

which gives the apparent equilibrium expressions:

$$[Al^{3+}]^2 * [OH^-]^6 * [SiO_2]^2 = K_{Kaolinite} \qquad (21a)$$

And this can be re-arranged to:

$$[SiO_2] =\ K_K * \frac{[H^+]^6}{[Al^{3+}]^2} \qquad (22a)$$

which leads to the "kaolinite" expression:

$$[SiO_2] =\ K_{Kaolinite} * \frac{[H^+]^3}{[Al^{3+}]} \qquad (22b)$$

Where $K_{Kaolinite}$ is the equilibrium coefficient being used. Note that the "equilibrium" equations assumed above, are not true equilibrium, and that kaolinite and gibbsite minerals are very slowly dissolving minerals under normal conditions. Both the "gibbsite" and "kaolinite" mentioned above are crude simplifications, possibly representing an amorphous precipitate combined with precipitation kinetics and ion exchange in the SkogSAFE model (The long term variant with variable mineralogy and surface areas, and that runs for 15,000 years in one simulation, see Alveteg et al., 1995, Rietz 1995, Warfvinge et al., 1996 for more information). These equations have been applied in the revised ForSAFE-2D model.

### 4.9. The full kinetic expression
The equations and approximations summarized above leads to the full revised mineral dissolution rate equations:

$$r = k_H * \frac{[H^+]^{n_H}}{f_H} \ + \ \frac{k_{H_2O}}{f_{H_2O}} \ + \ k_{CO_2} * P_{CO_2}^{n_{CO_2}} * \frac{1}{f_{CO_2}}$$
$$+ k_R * [R]^{n_R} * \frac{1}{f_R} \ + \ k_{OH} * \frac{[OH^-]^{n_{OH}}}{f_{OH}} \qquad (23)$$

where the retarding functions are given by:

$$f_{H^+} = \left(1 + \frac{[BC]}{C_{BC,H}}\right)^{x_H} * \left(1 + \frac{[Al^{3+}]}{C_{Al,H}}\right)^{y_H} * \left(\left(1 + K_{Si,H} * \left(\frac{[Si]}{C_{Si,H^+}}\right)^{z_H}\right)\right) \qquad (24)$$

$$f_{H_2O} = \left(1 + \frac{[BC]}{C_{BC,H_2O}}\right)^{x_{H_2O}} * \left(1 + \frac{[Al^{3+}]}{C_{Al,H_2O}}\right)^{y_{H_2O}} * \left(\left(1 + K_{Si,H_2O} * \left(\frac{[Si]}{C_{Si,H_2O}}\right)^{z_{H_2O}}\right)\right) \qquad (25)$$

$$f_{CO_2} = \left(1 + K_{CO_2} * \frac{P_{CO_2}}{P_{CO_2 Limit}}\right)^{n_{CO_2}} * \left(1 + \frac{[BC]}{C_{BC,CO_2}}\right)^{x_{CO_2}} * \left(1 + \frac{[Al^{3+}]}{C_{Al,CO_2}}\right)^{y_{CO_2}}$$
$$* \left(1 + K_{Si,CO_2} * \left(\frac{[Si]}{C_{Si,CO_2}}\right)^{z_{CO_2}}\right) \qquad (26)$$



$$f_R = \left(1 + \frac{[R]}{[R]_{Limit}}\right)^{n_R} * \left(1 + \frac{[BC]}{C_{BC,R}}\right)^{x_R} * \left(1 + \frac{[Al^{3+}]}{C_{Al,R}}\right)^{y_R} * \left((1 + K_{Si,R} * \left(\frac{[Si]}{C_{Si,R}}\right)^{z_R}\right) \quad (27)$$

$$f_{OH^-} = \left(1 + \frac{[BC]}{C_{BC,OH}}\right)^{x_{OH}} * \left(1 + \frac{[Al^{3+}]}{C_{Al,OH}}\right)^{y_{OH}} * \left((1 + K_{Si,OH} * \left(\frac{[Si]}{C_{Si,OH^-}}\right)^{z_{OH}}\right) \quad (28)$$

where:

$C_{BC,i}$ is the lower limiting base cation concentration in reaction i,

$C_{Al,i}$ is the lower limiting aluminium concentration in reaction i,

$C_{Si,i}$ is the lower limiting silica concentration in reaction i,

$P_{CO2limit}$ is the lower limiting carbon dioxide partial pressure in reaction i,

$[R]_{limit}$ is the lower limiting organic acid concentration in reaction i as concentration of DOC,

$x_i$ is the base cation brake reaction order for i,

$y_i$ is the aluminium brake reaction order for i

$z_i$ is the silica brake reaction order of i.

$K_{CO2}$ is the $CO_2$ brake coefficient and set to 20.

$K_{Si,i}$ is the silica brake constant for reaction i, set to 100.

Table 2. Alteration series from muscovite, biotite and feldspars to clays, corresponding to Figure 14.

| # | Mineral | Interlayer | Octahedral | Tetrahedral |
|---|---------|-----------|-----------|------------|
| | Muscovite pathway | | | |
| 1 | Muscovite | K | $Al_2$ | $Al_{1.0}Si_{3.0}O_{10}(OH)_2$ |
| 2 | Illite 1 | $K_{0.5}Mg_{0.01}Ca_{0.01}Al_{0.05}$ | $Al_{1.6}Fe_{0.25}Mg_{0.1}Ti_{0.04}$ | $Al_{0.6}Si_{3.4}O_{10}(OH)_2$ |
| 3 | Illite 2 | $K_{0.44}Mg_{0.01}Ca_{0.01}Al_{0.07}$ | $Al_{1.6}Fe_{0.25}Mg_{0.1}Ti_{0.04}$ | $Al_{0.6}Si_{3.4}O_{10}(OH)_2$ |
| 4 | Illite 3 | $K_{0.39}Mg_{0.013}Ca_{0.013}Al_{0.06}$ | $Al_{1.5}Fe_{0.32}Mg_{0.1}Ti_{0.08}$ | $Al_{0.6}Si_{3.4}O_{10}(OH)_2$ |
| 5 | Illitic vermiculite | $K_{0.35}Mg_{0.03}Ca_{0.03}Al_{0.06}$ | $Al_{1.63}Fe_{0.32}Mg_{0.08}Ti_{0.07}$ | $Al_{0.6}Si_{3.4}O_{10}(OH)_2$ |
| 6 | Kaolinite | | | $Al_2Si_2O_5(OH)_4$ |
| | Chlorite pathway | | | |
| 1 | Chlorite | $Ca_{0.5}Mg_{1.5}$ | $Al_{1.0}Fe_{0.5}\,Mg_{1.5}$ | $Al_{1.0}Si_{3.0}O_{10}(OH)_2$ |
| 2 | Vermiculite 1 | $K_{0.32}Mg_{0.07}Ca_{0.09}Al_{0.05}$ | $Al_{1.52}Fe_{0.35}Mg_{0.1}$ | $Al_{0.6}Si_{3.4}O_{10}(OH)_2$ |
| 3 | Vermiculite 2 | $K_{0.30}Mg_{0.05}Ca_{0.05}Al_{0.05}$ | $Al_{1.55}Fe_{0.32}Mg_{0.05}Ti_{0.06}$ | $Al_{0.6}Si_{3.4}O_{10}(OH)_2$ |
| 4 | Vermiculite 3 | $K_{0.25}Mg_{0.04}Ca_{0.04}Al_{0.08}$ | $Al_{1.55}Fe_{0.32}Mg_{0.05}Ti_{0.06}$ | $Al_{0.6}Si_{3.4}O_{10}(OH)_2$ |
| 5 | Al/OH interlayered vermiculite | $K_{0.11}Mg_{0.04}Ca_{0.04}Al_{0.1}$ | $Al_{1.52}Fe_{0.4}Mg_{0.05}Ti_{0.08}$ | $Al_{0.5}Si_{3.5}O_{10}(OH)_2$ |
| 6 | Kaolinite | | | $Al_2Si_2O_5(OH)_4$ |
| | Biotite pathway | | | |
| 1 | Biotite | $K_{1.0}Mg_{2.0}$ | $Al_{0.5}Fe_{0.5}Mg_{1.0}$ | $Al_{1.0}Si_{3.0}O_{10}(OH)_2$ |
| 2 | Vermiculite 1 | $K_{0.32}Mg_{0.07}Ca_{0.09}Al_{0.05}$ | $Al_{1.52}Fe_{0.35}Mg_{0.1}$ | $Al_{0.6}Si_{3.4}O_{10}(OH)_2$ |
| 3 | Vermiculite 2 | $K_{0.30}Mg_{0.05}Ca_{0.05}Al_{0.05}$ | $Al_{1.55}Fe_{0.32}Mg_{0.05}Ti_{0.06}$ | $Al_{0.6}Si_{3.4}O_{10}(OH)_2$ |
| 4 | Vermiculite 3 | $K_{0.25}Mg_{0.04}Ca_{0.04}Al_{0.08}$ | $Al_{1.55}Fe_{0.32}Mg_{0.05}Ti_{0.06}$ | $Al_{0.6}Si_{3.4}O_{10}(OH)_2$ |
| 5 | Al/OH interlayered vermiculite | $K_{0.1}Mg_{0.04}Ca_{0.04}Al_{0.1}$ | $Al_{1.52}Fe_{0.4}Mg_{0.05}Ti_{0.08}$ | $Al_{0.5}Si_{3.5}O_{10}(OH)_2$ |
| 6 | Kaolinite | | | $Al_{2.0}Si_2O_5(OH)_4$ |
| | Feldspar pathway | | | |
| 1 | Feldspar | K, Na, Ca | | $Al_1Si_3O_8$ |
| 2 | Sericite | $Na_{0.1}K_{0.75}$ | $Al_{1.9}Mg_{0.1}$ | $Al_{0.84}Si_{3.16}O_{10}(OH)_2$ |
| 3 | Sericitic vermiculite 1 | $K_{0.3}\,Mg_{0.02}Ca_{0.05}$ | $Al_{0.02}$ | $Al_{1.0}Si_3O_{10}(OH)_2$ |
| 4 | Sericitic vermiculite 2 | $K_{0.1}\,Mg_{0.05}Ca_{0.02}$ | $Al_{0.05}$ | $Al_{1.0}Si_3O_{10}(OH)_2$ |
| 5 | Al/OH interlayered vermiculite | $K_{0.1}Mg_{0.04}Ca_{0.04}Al_{0.1}$ | $Al_{1.52}Fe_{0.4}Mg_{0.05}Ti_{0.08}$ | $Al_{0.5}Si_{3.5}O_{10}(OH)_2$ |
| 6 | Kaolinite | | | $Al_{2.0}Si_2O_5(OH)_4$ |




### 4.9. Secondary phases in the soil

A significant fraction of the primary minerals dissolve incongruently to alteration minerals. Attention was also paid to the secondary minerals and clays. Both terms are inconsistently used in the literature, and thus we define them as follows: We have defined clay minerals by their composition (Kaolinite, gibbsite, quartz) and as listed in Table 3. This approach is thus not based on their particle size, but on the molecular crystalline structure. Secondary minerals formed in either two ways; a mineral that has been altered significantly in situ as is described in Table 2, for example when muscovite is altered through a series of illite and vermiculite phases and finally to kaolinite as the end product. Vermiculite, illite, montmorillonite are minerals of variable composition that are often called clays when they are not in crystalline form. However on the microscopic level, they have a crystalline structure. Thus, clay can be defined by particle size alone, or as a specific mineral. We have used the specific mineral name, independent of particle size. In the soil, amorphous phases are composed of aluminium, silicate and soil organic substances. These amorphous phases slowly change composition as the organic matter decomposes and a more solid structure emerges. The alteration series from muscovite, biotite and feldspars to clays, are illustrated schematically in Figure 14 and listed in Table 2. The concept behind Table 2 is that as these minerals go through incongruent dissolution (alteration), they become depleted in certain ions (like Ca, Mg, K or Na, and depending on pH, in aluminium (at low pH) or silica (at high pH), but the crystal structure remains constant. Thus the crystal lattice destruction rate remains, but the base cation content of this structure becomes poorer, yielding less cations and less acidity neutralization. We have simplified this process down to 4 pathways, the muscovite pathway, the chlorite pathway, the biotite pathway and the feldspar pathway. Muscovite changes through a series of alteration reactions to illite and finally to kaolinite. Chlorite alters to vermiculites and finally to kaolinite. Biotite goes through a series of alterations to vermiculite and kaolinite. Feldspars go through alterations, K-Feldspars through sericites and plagioclases to vermiculites (Holmqvist 2004, Holmqvist 2002, 2003). This sequence has been discussed in the SUFOR project and again in the QWARTS workshops and will be later implemented into ForSAFE-2D.

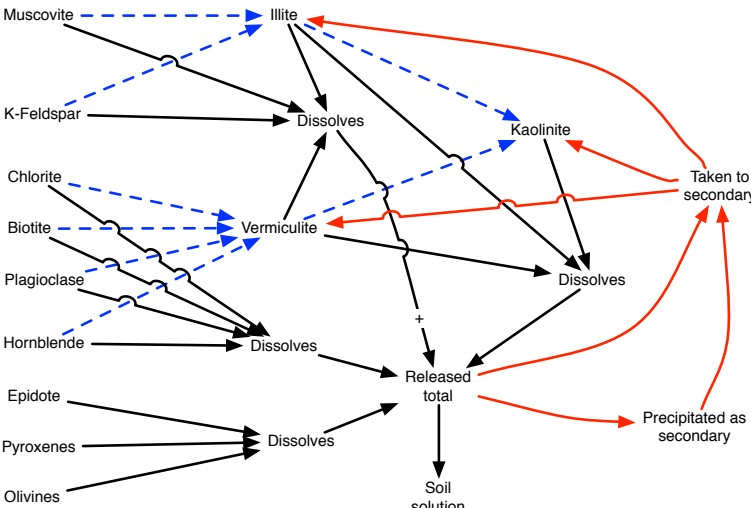

*Figure 14. The alteration sequence developed for primary mineral towards alteration minerals, of which some are clay minerals. All minerals that dissolve contribute to the precipitation of secondary minerals.*

### 4.10. The parameterization of the kinetic rate equations

The parameterization database for the PROFILE model (and ForSAFE) was updated to be consistent with previous databases (Sverdrup 1990, 1996, 2009, Sverdrup and Warfvinge 1988a,b, 1991, 1992a,b, 1993, 1995, Holmqvist 2002, 2003). The original PROFILE database had kinetic data for 59 different minerals, and about 25 different carbonates and some artificial silicates. In addition new data from our





own experiments (Sverdrup 1998, 1996, Sverdrup and Alveteg 1998, Holmqvist et al., 2002, 2003; Sverdrup and Holmqvist 2004) and from the literature[2] have been considered. Care of these new data sources we have about 90 different silicate or aluminium minerals and 6 generic carbonates listed. Of these minerals, the regression of ~20 have yet to be published. In due time, these will get their own proper publications, it is not the scope of this study to do them in detail. Such a documentation would be 1-2 years into the future from the present time. Rather some selected examples will be presented below. The estimation of rate parameters was performed using the complete rate equation 1 and Equations 21-26. As such for every rate from an experiment, the rate must be known, along with the concentrations of all reactants at the conditions that rate was observed including $[H^+]$, $pCO_2$, $[R]$, $[OH^-]$, as well as the reaction products in solution potentially contributing to retarding the dissolution reaction; $[Ca^{2+}]$, $[Mg^{2+}]$, $[K^+]$, $[Na^+]$, $[Al^{3+}]$, $[Al(OH)_4^-]$, $[H_4SiO_4]$ (Sverdrup 1990, Sverdrup and Warfvinge 1995). The experiments must have been performed over sufficient reaction conditions for the parameters in Equation 29 to be estimated. In some cases, the data from different experimental studies were combined, to determine rate parameters or a reaction orders. During the regression process, experimental studies with insufficient data or documentation were omitted, unless the gap could be bridged with reasonable assumptions. Data regression was performed by rearranging equation (22) to:

---

[2]Examples are the following list of articles and studies we have used, but not limited to: Ajemba and Onokwuli 2012, Alekseyev 2007, Alexeyev et al., 1997, Amram and Ganor 2005, Amrhein and Suare 1992, Anbeek 1992a,b, Anbeek et al., 1994, Aradottir et al., 2013, Bandstra et al., 1998, Beig and Lüttge 2006, Bengtsson and Sjöberg 2009, Berg and Banwart 1994, 2000, Bibi et al., 2010, Bickmore et al., 2006, Blake and Walther 1996, Blum and Stillings 1995, Blum and Lasaga 1988, 1991, Blum 1994, Brady and Walther 1992, Bray et al., 2015, Brandt et al., 2005, Brantley 2003, 2008a,b, Brantley and Stillings 1994, 1996, Brantley and Chen 1995, Brantley and Conrad 2008, Brady and Walther 1992, Braun et al., 2016, Bray 2015, Cama et al., 2000, Carrol and Knauss 2005, Carrol and Walther 1990, Carrol and Smith 2013, Casetou-Gustafsson et al., 2018, Casey et al., 1991, Casey and Sposito 1992, Casey and Westrich 1992, Chaïrat et al., 2007, Chen and Brantley 1997, 1998, 2000, Chin and Mills 1991, Critelli et al., 2015, 2014, Cotton 2008, Crundwell 2013, 2014, b,c,d, 2015a,b, 2017, Daval et al., 2010a,b, 2013, Devidal et al., 1997, Diedrich et al., 2014, Dixit and Carrol 2007, Dove and Crerar 1990, Dorozhkin 2012, Dresel 1989, Drever et al., 1994, 1996, Drewer and Clow 1995, Drewer and Zobrist 1992, Drever and Stillings 1997, Dorozin 2012, Duckworth and Martins 2003a,b, Fernandez-Bastero et al., 2008, Fischer and Liebscher 2014, Finlay et al., 2010, Fouda et al., 1996a,b, Frogner and Schweda 1998, Fumuto et al., 2001, Gahrke et al., 2005, Ganor et al., 2005, Gautier et al., 1994, Gislasson and Hans, 1987, Gislasson and Oelkers 2003, Gislasson et al., 1996, Godderis et al., 1996a,b, Glover et al., 2003, Godderis et al., 2006, Golubev et al., 2004, 2005, Guidry and Mackenzie 2003, Goyne et al., 2006, Gudbrandsson et al., 2011, 2014, Gustafsson and Puigdomenech 2003, Hamilton et al., 2000, 2001, Hangx and Spiers 2009, Harouiya et al., 2007, Harouiya and Oelkers 2004, Haug et al., 2010, Hausrath et al., 2009, Hayashi and Yamada 1990, Helgeson et al., 1984, Hellmann 2007, 2006, 2010, Hilley et al., 2010, Holmqvist and Sverdrup 2001, Holmqvist et al., 1999, 2002, 2003, 2004, Hodson 2006a,b, Hodson and Langan 1999, Hodson et al., 1996, 1997, Hänchen et al., 2006, Huertas et al., 1999, 2001, Jin et al., 2011, Johnsson et al., 1992, Johnson et al., 2014, Jonckbloedt 1998, Jönsson et al., 1995, Kalinowski 1997, Kalinowsli and Schweda 1995, Kalinowski et al., 1998, Knauss et al., 1993, Køhler et al., 2003, 2005, Kuwahara 206a,b, 2008, Labat and Viville 2006, Lagache 1965, Langan et al., 1996, Lartigue 1994, Lasaga 1995, 1998, Lowson et al., 2005, 2007, Lazaro et al., 2015, Lu et al., 2013, 2015, Ludwig et al., 2013, Maher 2010, Malmstrøm and Banwart 1997, Malmström et al., 1996, Maurice et al., 2002, Mazer and Walther 1994, McCourt and Hendershot 1992, Metz et al., 2005, Meyer 2014, Mongeon et al., 2007, Murakami et al., 1998, Murphy and Helgesson 1987, Murphy et al., 1992, 1996, Nagy 1995, Nagy and Lasaga 1992, Nagy et al., 1991, Navarre-Sitchler and Thyne 2007, Nesbitt et al., 1991, Nyström-Claesson and Andersson 1996, Numan and Weaver 1969, Oelkers 2001a,b,, Oelkers and Schott 1995a,b, 1998, 2001, Oelkers et al., 1994, 2008, Oelkers and Gislasson 2001, Olsen 2007, 2008, Olsson 2007, Opolot and Finke 2015, Oxburgh 1991, Oxburgh et al., 1994, Paces 1983, Palandri and Kharka 2004, Pokrowsky and Schott 2000a,b, 2002, Pokorowsky et al., 2004, Poulson 1997, Prajapati et al., 2014, Price et al., 2005, Pigiobbe et al., 2009, Ragnarsdottir 1993, Ragnarsdottir and Graham 1996, Raschmann and Fedorockova 2008, Rietz 1995, Rimstidt et al., 2012, Ross 1969, Rosso and Rimstidt 1999, Rozalen et al., 2014, Running and Gower 1991, Saldi et al., 2007, Sanemasa and Katura 1973, Schnoor 1990, Schofield et al., 2015, Schott et al., 2009, 2012, Smith et al., 2013, Smits and Wallander 2016, Smits et al., 2014, Soler et al., 2008, Stephens and Hering 2003, Stillings and Brantley 1995, Stillings et al., 1996, Stockmann et al., 2008, Stumm and Wollast 1990, Stumm and Wieland 1990, Sverdrup 1990, 1996a,b, 1998, 2009, Sverdrup and Bjerle 1982, Sverdrup and Alveteg 1998, Sverdrup and Holmqvist 2016, Sverdrup and Warfvinge 1992a,b, 1995, Sverdrup et al., 1986, 1987, 1995,a,b, 1998, 2002, 2006, 2008, 2010, Traven et al., 2005, Swoboda-Collberg and Drever 1993, Taylor et al., 1999, 2000, Taylor and Blum 1995, Taylor et al., 2017, Techer et al., 2007, Teir et al., 2007, Terry 1983a,b,c, Terry and Monhemius 1983, Thom et al., 2013, Valsami-Jones et al., 1998, Turpault and Trotignon 1994, Valsami-Jones et al., 1998, Voltini et al., 2012, Wang and Giammar 2012, Wang et al., 2017, Warfvinge and Sverdrup 1992a,b,c,d, 1993, 1995, Warfvinge et al., 1987, 1992, 1993, 1996, 2000, Weissbart and Rimstidt 2000, Welch and Ullman 1993, 1996, 2000, Westrich et al., 1993, White and Brantley 1995, 2003, White and Blum 1995, White et al., 1999, Whitfield et al., 2009, 2010, Wogelius and Walther 1991, 1992, Wolff-Boenisch et al., 2004a,b, 2011, Wood et al., 1999, Xie and Walter 1994, Yadaw and Chakrapani 2006, Yadaw et al., 2000, Yang and Steefel 2008, Yoo et al., 2009, Yu et al., 2016, 2017, Zabowski et al., 2007, Zhang and Bloom 1999a,b, Zhang et al., 1996, 2015, Zhang et al., 2013, Zhang and Lüttge 2017, 2009a,b, Zhu et al., 2010, Zassi 2009, Zavodsky et al., 1995, Zysset and Schindler 1996).



$$k_H * \frac{[H^+]^{n_H}}{f_H} = r_{Observed} - (\frac{k_{H_2O}}{f_{H_2O}} + k_{CO_2} * \frac{P_{CO_2}^{n_{CO_2}}}{1 + K_{CO_2} * P_{CO_2}^{n_{CO_2}}} * \frac{1}{f_{CO_2}}$$
$$+ k_R * \frac{[R]^{n_R}}{1 + K_R * [R]^{n_R}} * \frac{1}{f_R} + k_{OH} * \frac{[OH^-]^{n_{OH}}}{f_{OH}}) \qquad (29)$$

In the neutral pH range, such as pH 7 and lower, this equation can be simplified in most instances by removing the OH-reaction to get (Sverdrup 1990):

$$k_H * \frac{[H^+]^{n_H}}{f_H} = r_{Observed} - (\frac{k_{H_2O}}{f_{H_2O}} + k_{CO_2} * \frac{P_{CO_2}^{n_{CO_2}}}{1 + K_{CO_2} * P_{CO_2}^{n_{CO_2}}} * \frac{1}{f_{CO_2}}$$
$$+ k_R * \frac{[R]^{n_R}}{1 + K_R * [R]^{n_R}} * \frac{1}{f_R}) \qquad (30)$$

and the in the acid pH range, this may be reduced to:

$$k_H * \frac{[H^+]^{n_H}}{f_H} = r_{Observed} \qquad (31)$$

By entering the concentrations of $H^+$, base cations, aluminium and silica into these equations, we can determine the rate coefficient, $k_H$, and $f_{H+}$. When the experiment was performed in the absence of organic acids, as is often the case, Equation (29) reduces to:

$$k_H * \frac{[H^+]^{n_H}}{f_H} = r_{Observed} - (\frac{k_{H_2O}}{f_{H_2O}} + k_{CO_2} * \frac{P_{CO_2}^{n_{CO_2}}}{1 + K_{CO_2} * P_{CO_2}^{n_{CO_2}}} * \frac{1}{f_{CO_2}}) \qquad (32)$$

Some experiments were conducted at very low or with no dissolved $CO_2$ present and with organic ligands absent. In such cases, Equation (29) reduces to (Sverdrup 1990, Chin et al., 1991):

$$r_H = k_H * \frac{[H^+]^{n_H}}{f_H} = r_{Observed} - \frac{k_{H_2O}}{f_{H_2O}} \qquad (33)$$

In this latter case, two reactions influence mineral dissolution rates: 1) the $H^+$ reaction, and 2) the water reaction. The variation of rates as a function of pH at such conditions consists of a 'flat part' where rates are controlled by the water reaction (Figure 17). At these conditions, by entering the concentrations of retarding base cations, aluminium and silica, the rate coefficients can be determined. In the semi-neutral region (pH 6-8), the expression may be a flat line and the rate expression is reduced to:

$$r_{Observed} = \frac{k_{H_2O}}{f_{H_2O}} + k_{CO_2} * \frac{P_{CO_2}^{n_{CO_2}}}{1 + K_{CO_2} * P_{CO_2}^{n_{CO_2}}} * \frac{1}{f_{CO_2}} + k_R * \frac{[R]^{n_R}}{1 + K_R * [R]^{n_R}} * \frac{1}{f_R}) \qquad (34)$$

When neither organic ligands nor $CO_2$ is present, and in the pH range of 6-8, this is reduced to:

$$r_{Observed} = \frac{k_{H_2O}}{f_{H_2O}} \qquad (35)$$

With only organic acid ligands but no $CO_2$ present, and in the pH range of 6-8, the rate expression becomes:

$$r_{Observed} = \frac{k_{H_2O}}{f_{H_2O}} + k_R * \frac{[R]^{n_R}}{1 + K_R * [R]^{n_R}} * \frac{1}{f_R}) \qquad (36)$$





In the far alkaline region (pH 10-14), where we may assume that the OH- reaction will be dominant, the rate expression reduces to:

$$k_{OH} * \frac{[OH^-]^{n_{OH}}}{f_{OH}} = r_{Observed} \qquad (33)$$

By entering the concentrations of base cations, aluminium and silica, $f_{OH}$ can be determined and the rate coefficient, $k_{OH}$, and reaction order, $n_{OH}$ be determined. The reaction order $n_H$ and the coupled $n_{OH}$ for the $H^+$ and the $OH^-$ reaction is derived from plots of the rate versus the solution pH

Figure 15 shows diagrams used to quantify the retarding effect of aluminium on the dissolution rate of albite feldspar. The figures were adapted from Sverdrup (1990) and the work prepared for Sverdrup and Warfvinge (1995) and Sverdrup et al., (2009). Similar results for aluminium was found by Oelkers (2001), Oelkers and Gislasson (2001), Oelkers and Schott (2001, 1995a,b), Oelkers et al., (1999) for several minerals. The aluminium brake is very prominent in the range of log [Al] from -7 to -4.5. For further information, see Sverdrup (1990) and Sverdrup and Warfvinge (1995).

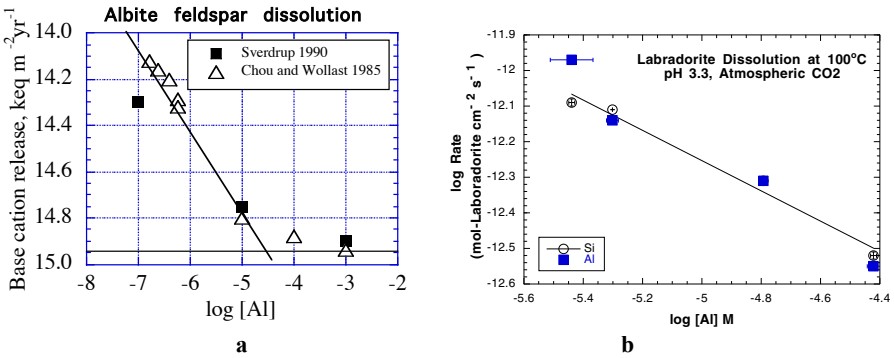

*Figure 15. Regression plots showing the retarding effect of aluminium on the dissolution rate of albite. The figures were adapted from Sverdrup (1990). The decrease of rates as a function of aqueous aluminium concentration (the aluminium brake) is very prominent in the range of log [Al] from -7 to -4.5. Aluminium concentrations are in kmol m$^{-3}$. The figures were adapted from (a) Sverdrup et al. (1990) and from (b) Carrol and Knauss (2001). For further information, see Sverdrup (1990) and Sverdrup and Warfvinge (1995).*

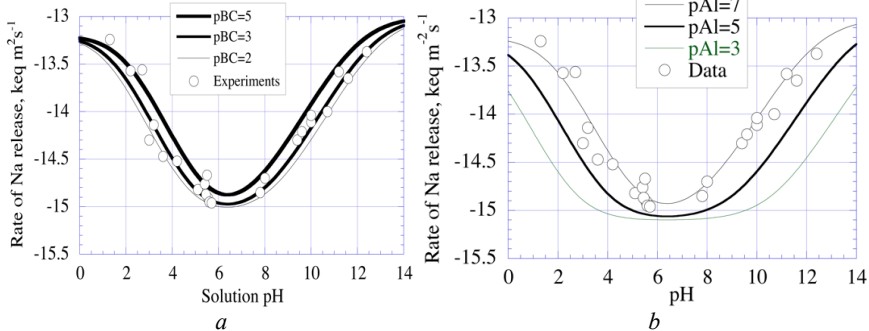

*Figure 16. The effect on the base cation (a) and the aluminium concentration (b) on the dissolution rate of albite. (Sverdrup 1990). The circles represent the data from experiments, the solid lines the model simulations.*



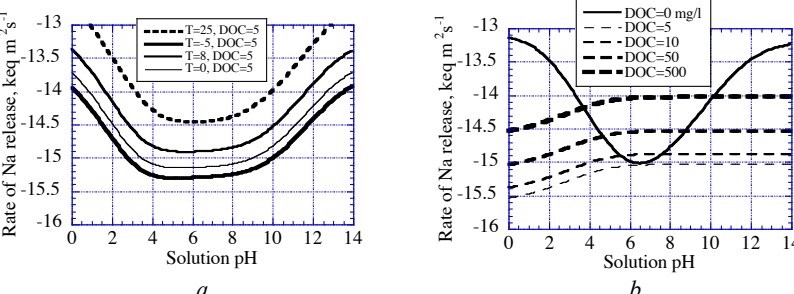

*Figure 17. The effect on the base cation (a) and the aluminium concentration (b) on the dissolution rate of albite. The solid line is the reaction rate without $CO_2$ or organic acid ligands.*

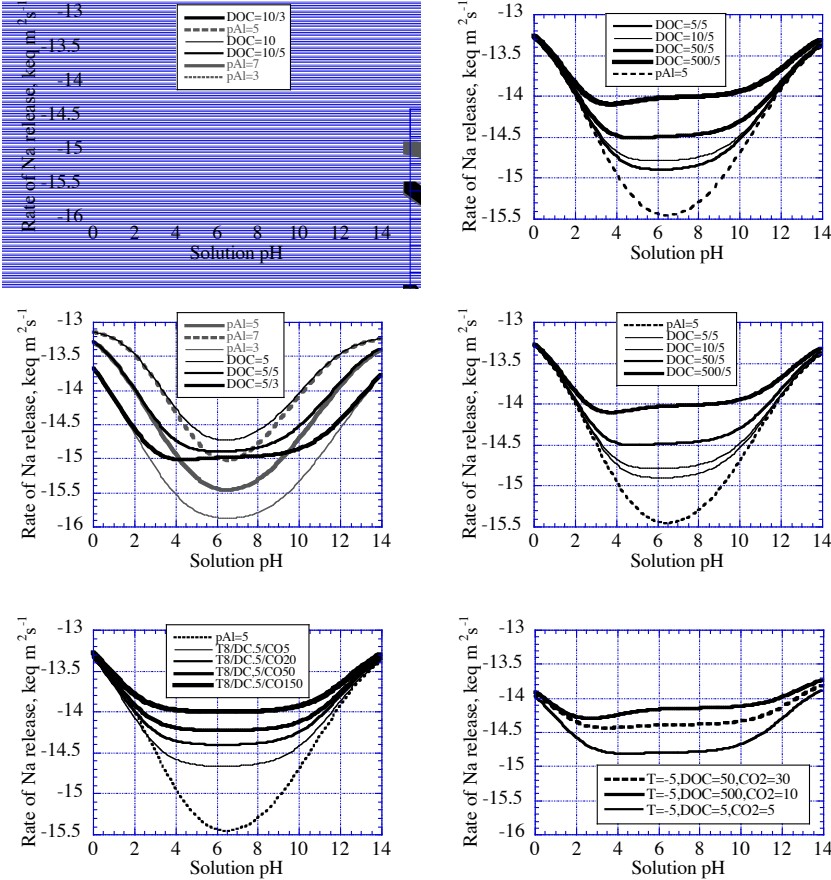

*Figure 18. The weathering rate model was used to plot different combinations of conditions, to investigate the different shapes the weathering rate dependency can change (See Figure 10 and 12 for how the principle works). The experimental data were overlaid in such diagrams, to help retreive kinetic parameters (e.g. rate coefficients and reaction orders). The last diagram, lower right, shows the combination of different combinations of organic acid ligand concentrations and $CO_2$ pressures in atmospheres.*





The reaction order for the organic acid reaction is derived from experiments where only the concentration of organic ligand, [R], has been varied. This was found to be $n_R=0.5$ on most experiments and this exponent value was universally adopted, suggesting a divalent ligand being the reactive agent (Sverdrup 1990, Sverdrup and Warfvinge 1995, Oelkers and Schott 1998).

The reaction order $n_{CO_2}$ for the reaction with $CO_2$ has been very difficult to constrain, as very few experiments that allow it to be determined are available (Daval et al., 2013, Berg and Banwart 2000, Golubev et al., 2005, Fernandez-Bastero et al., 2008, Hangx and Spiers 2009, Lagache 1965, Wogelius and Walther 1991, Wolff-Boenisch et al., 2011, Stephens and Hering 2004, Sverdrup 1990). The few experiments available to do not completely agree on the issue. Many experiments dealing with the effect of $CO_2$ on weathering do not have the required resolution to allow data regression,. For the minerals where the $CO_2$ has little or no effect, this is fine, but for some it is. It was found to be $n_{CO_2}=0.6$ and was universally adopted. Sometimes these parameterizations can be determined by making single factor plots, but more often, the whole model must be used to recreated the experiments, taking many factors into account simultaneously. Figure 16 shows the effect on the base cation (a) and the aluminium concentration (b) on the dissolution rate of albite. Various plots were used to help data interpretation. Figure 17-18 illustrates how the model was used to plot up different combinations of conditions, to investigate how distinct factors affect the weathering rates. The experimental data were overlaid in such diagrams (Figures 16-20) to help interpretation towards kinetic parameters (rate coefficients and reaction orders), for example the combination of different organic acid ligand concentrations and aluminium concentrations. The last diagram, on the lower right of Figure 18, shows the combination of different combinations of organic acid ligand concentrations and $CO_2$ pressures in atmospheres. Figure 19 shows the effect on rates of the base cation (a) and the aluminium concentration (b) on the dissolution rate for albite. The circles represent the data from experiments.

A further example of parameterization efforts is shown in Figure 19 for the case of hornblende dissolution rate data reported by from Holmqvist and Sverdrup (2004) and Holmqvist et al. (2002, 2003). Figure 19a and 19b shows these data as a function of pH. The figures were adapted from Holmqvist et al., 2003). Figure 19c shows the retarding effect of aluminium on the dissolution rate of hornblende, adapted from Holmqvist et al., (2003). Figure 19d shows a three-dimensional plot for the dissolution rate of hornblende, as a function of solution pH and aluminium concentration (Sverdrup, 1990).

In total, the dissolution rate of hornblende is defined by a response surface in at least 8 and perhaps 9 different chemical factors: pH, Ca+Mg, K, Na, Al, DOC, $CO_2$, Si and sometimes Fe, and in addition to mineral surface area, soil wetting degree and temperature. For example changes in the aluminium concentration, can change the weathering rate by several orders of magnitude. Additional examples are presented in Figs. 20-24.

Figure 20 shows a typical example of data generated for different minerals during the 1996-2002 field seasons using a continuous, flow through, fluidized bed, with constant concentration feed solutions. This is for epidote after Holmqvist et al, (2003). Figure 21 shows the experimentally measured dissolution rates of epidote as a function of pH according to a number of weathering experiments. The release of all relevant ions were monitored by frequent during the experiments. Figure 22a shows the activation energy for the dissolution of epidote. The dependence of the dissolution rate of epidote on the calcium concentration at pH 2 and pH 4 is shown in Figure 22b. Figure 23 and 24 shows data from Holmqvist and Sverdrup (2004) and Holmqvist et al., (2002, 2003) confirming that an arithmetic addition of the various rate contributions gives the best fit of the data, consistent with the principle shown in Figure 10. Figure 24 shows results from hornblende, the bottom diagrams (A, B) shows results from a natural illite mineral extracted from an agricultural soil sample taken at the agricultural research site at Lanna, Swedish Agricultural University, Uppsala, Sweden. Model lines were fitted to the data points to set the rate coefficients and reaction orders. Note that a complete set of kinetic parameters could not be directly generated for all minerals due to incomplete experimental data sets. Estimates for some of the rate coefficients in Table 3 were estimated based on mineral crystal structure analogies (Sverdrup 1990, Holmqvist 2003, Sverdrup and Stiernquist 2002, Crundwell 2014a,b, 2016), crystal bond energies (Sverdrup 1990, Velbel 1999, Crundwell 2014b, 2016) and comparison with analogue minerals. For many of the minerals, the dissolution kinetics patterns are very consistent. The dissolution rate curve shapes of feldspars, garnets, olivines, zoisites allow for this, but also muscovite to illite alteration series, K-feldspar to sericite alteration series.





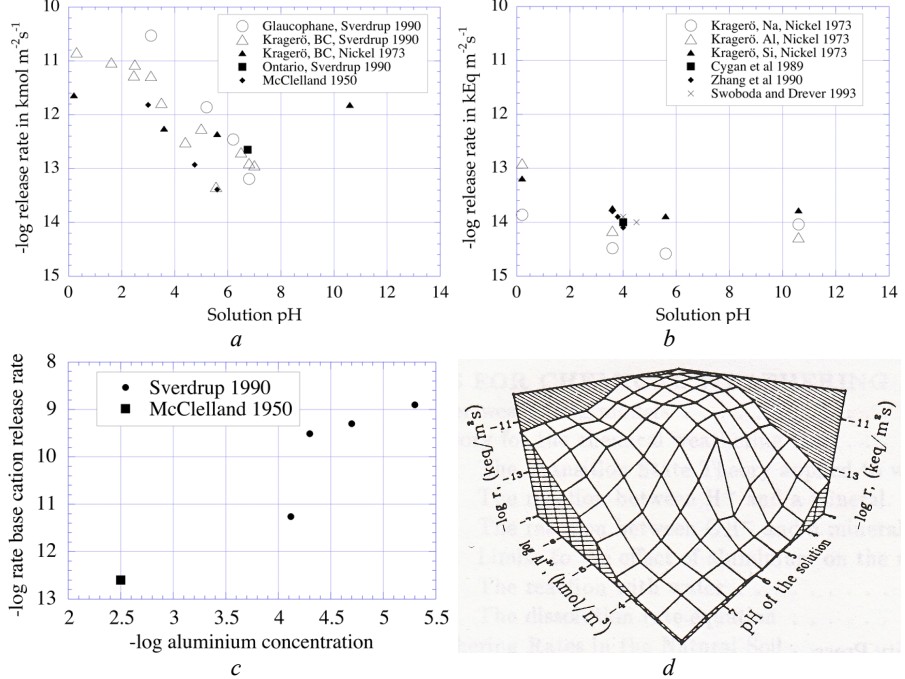

*Figure 19. Diagrams (a) shows the dissolution rate of minerals presented as base cation release rates as a function of pH and (b) shows the dissolution rate for hornblende as a function of solution pH, but under different experimental conditions. Diagram (c) The retarding effect of aluminium on the dissolution rate of hornblende. (Adapted from Holmqvist et al., 2003). Diagram (d) shows a three-dimensional plot for the dissolution rate of hornblende, as a function of solution pH and aluminium concentration (Sverdrup 1990).*

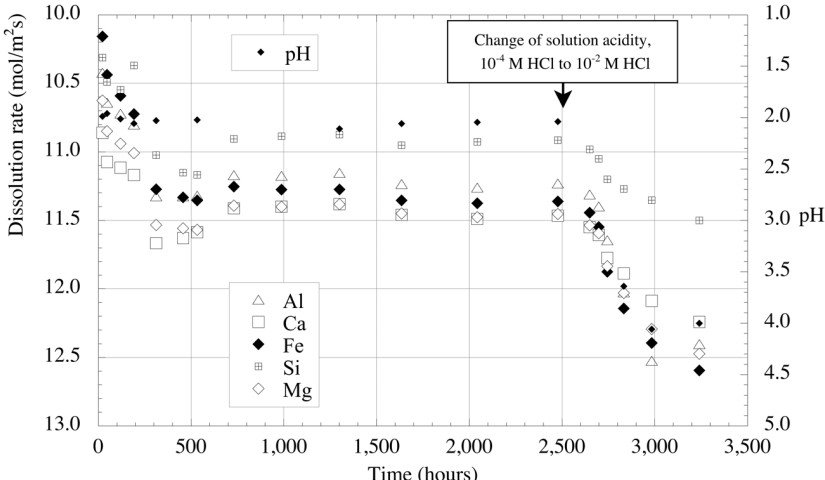

*Figure 20. Typical example of dissolution rate data generated for epidote during the season 1996-2002 using a continuous, flow through, fluidized bed, with constant concentration feed solutions (Holmqvist 2002, 2003). All relevant constituents of the mineral were monitored in the aqueous solution in the experiment.*





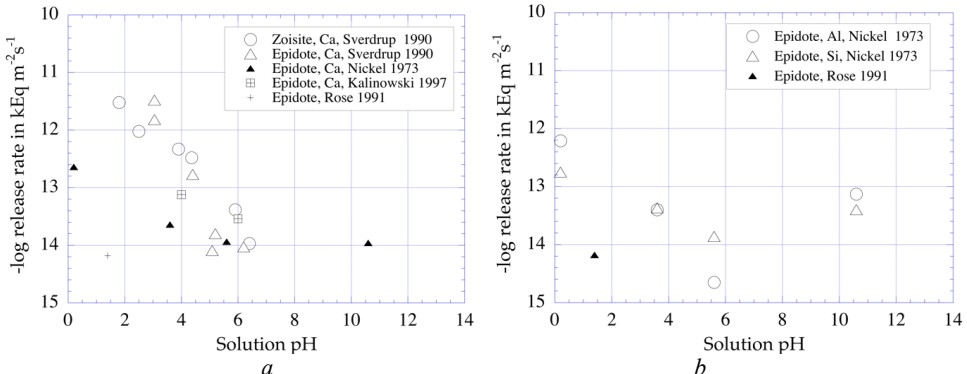

*Figure 21. Epidote dissolution rate versus pH according to experiments reported by Holmqvist and Sverdrup and other literature sources data.*

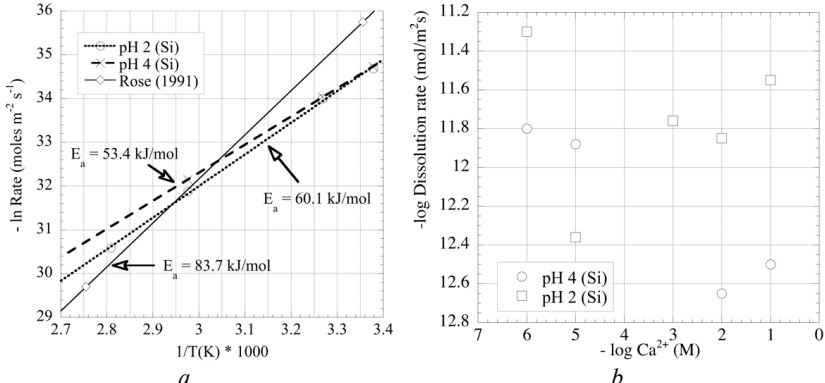

*Figure 22. a) Estimates of the energy of activation for the dissolution of epidote. (b) the dependence of the rate of epidote on the calcium concentration at pH 2 and pH 4 (From one series of experiments by the authors).*

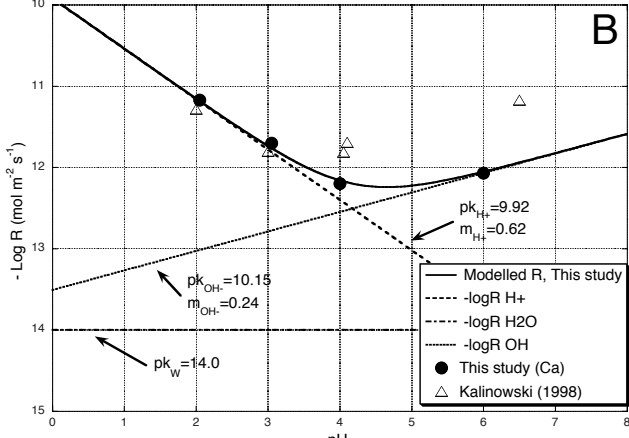

*Figure 23. Hornblende dissolution rate data from Holmqvist and Sverdrup (2004) and Holmqvist et al., (2002, 2003) suggests that an arithmetic addition gives the best fit.*



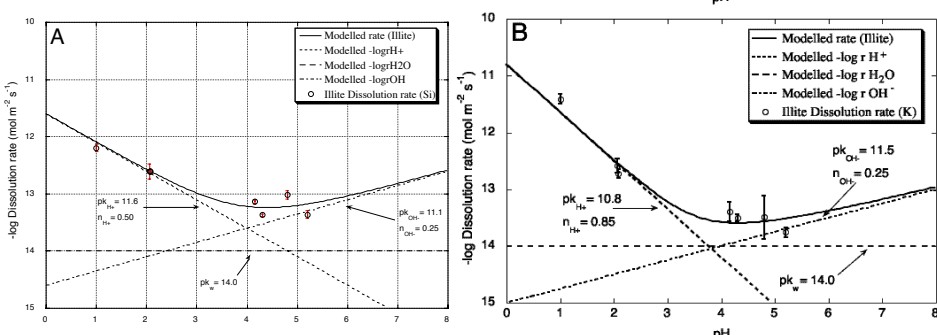

*Figure 24. Diagram A show regression results from hornblende, diagram (B) shows regression results from a natural illite mineral extracted from an agricultural soil sample taken at the agricultural research site at Lanna, Uppsala, Sweden. Data from Holmqvist and Sverdrup (2004) and Holmqvist et al., (2002, 2003)*

For example, for the feldspars, we have the data to parameterize the $H^+$ reaction for 5 different plagioclases, the mixed composition plagioclases from albite to anorthite. A plagioclase with a different composition will be interpolated between these as shown in Figure 24. We have the same situation for K-feldpars with increasing contents of Na and Ca, giving a systematic shift in parameter values. The pattern is very consistent as can be seen from the diagrams shown in Sverdrup (1990). However, for the $OH^-$ reaction we have less information. The $OH^-$ rate equation is theoretically linked to the $H^+$ reaction, but more sensitive to the concentration of the same base cation as in the mineral (Na, K, Ca). With the available data and the theoretical link, we can estimate the missing parameters for some of the feldspars. There is a similar situation for the $H_2O$ reaction. We have the experiments that allow it to be constrained for most of the feldspars, and the shifts between the feldspars are systematic and consistent.

For the reaction with organic acid ligands, the situation is more complex. Many of the dissolution experiments run with organic acids were present were poorly documented, and getting any accurate parameterization out of them is not possible. For some minerals like feldspars and olivine, some experimental results are available (Stillings et al., 1996 is one example for feldspar) that allow for kinetic parameter estimation. They found $n_R=0.75$ in the range pH 3-7. For other minerals, we have only single experiments, scattered among some few minerals. Few experiments are available, and for only a few types of minerals. These have delivered suggestions for expert judgement on what the parameter values probably would be. The situation is similar for the reaction between the mineral surface and $CO_2$. The reaction seems to be weak, and only play a role at elevated pressures. For example, Wang (2013), based on the experimental results of Hänchen et al., (2006) concluded there was no effect of the $CO_2$ reaction on olivine dissolution rates beyond the effect caused by $CO_2$ on pH.

Retrieved kinetic parameters are provided in Table 3. Parameters that are derived directly from of one or more experiments are given in **bold** font. The kinetic parameters that were estimated are shown in roman font. The minerals in this table are divided into 11 groups of basic crystalline structures. Some of the minerals inside each group have large commonalities with respect to how they dissolve, and this was of great help in parameter estimation table.

For feldspars, nesosilicates and phyllosilicates, the amount of experimental data available make the retrieved parameters robust. If three different compositions of basically the same type of mineral, A, B and C, are known to have relative rates A>B>C, and we have the kinetic parameters for A and C, then we can be fairly certain that the values for the kinetic parameters for B are constrained between A and C (see Figure 25). If they are close, then we would be able to set B fairly accurately, even with sparse experimental data for B. This has been the case for many minerals (In particular feldspars, nesosilicates, phyllosilicates), and is a way to get more parameterization out of a limited experimental data sets. For the pyroxenes and amphiboles, the experiments indicate that the minerals tend to behave with some variety depending on their composition, making the estimates less accurate. But, many pyroxenes are mixtures of definable end members and this was utilized to interpolate and estimate missing parameters.





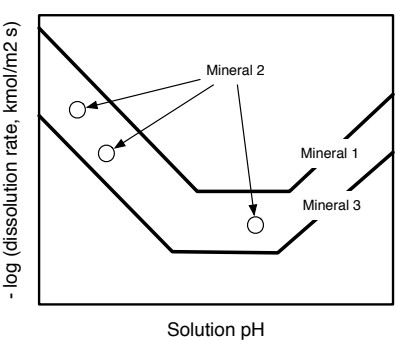 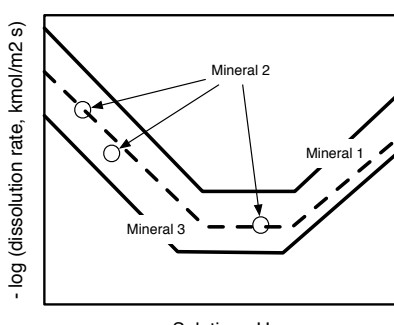

| a: Data points drawn in | b: Interpolate line |

*Figure 25. Some mineral groups have very similar dissolution rate behaviours. Such similarities can be used to interpolate between them (b) when we have intermediate minerals with only a few data points available (a).*

Nevertheless all parameters in Table 3 together with their kinetic expressions should be further validated as additional experimental data become available. The ultimate test of the kinetics equations and parameters how well they describe both laboratory experiments and field data where independent estimates of the weathering rate is available. Such tests have been generally successful (see the publications referred to earlier, and Erlandsson Lampa et al., 2016, 2019), suggesting that the combined methodology (experiments, analogues, interpolations, estimates based on theoretical rescaling, predictions made based on crystal bond energies) have captured the kinetics sufficiently well. More on this will be forthcoming as the publishing of further comparisons are made.

## 5. Results
### 5.1. Kinetics and parameterization
The tabulated kinetics coefficients are the major result of this report and they are provided in the Tables 1-4. In total the dissolution kinetics parameterization for 93 minerals are provided. The fundamental rate equation, as described above was adapted after Sverdrup and Warfvinge (1988, 1992, 1995) and Sverdrup (1990, 1998) and parameters are for a temperature of $8^{\circ}C$ and standard atmospheric pressure. The numbers in bold in Table 3 represent direct measurement, normal font parameters were estimated by interpolation from analogues. The following default approximations were adopted due to the lack of data; $C_{Al}$ for the $H^+$-reaction is taken to be equal to $^1/_3$ of the $C_{Al}$ for the $OH^-$-reaction. $C_{BC}$ for the $H^+$-reaction is taken to be $^1/_3$ of the $C_{BC}$ for the $OH^-$-reaction. The retarding reaction orders for base cations (x), aluminium (y) and silicate (z) have been extracted from separate datasets and experiments where it was possible to separate out the effect of silicate alone, having subtracted the effect of base cations and aluminium first. Default values were computed and scaled with Madelung crystal lattice site energy (See Sverdrup 1990 and Velbel 1999 for how a-priori weathering rate coefficient estimates are made from crystal properties). Irreversible dissolution implies that the mineral cannot be formed from solution under soil conditions, and that there is no saturation concentration or any back reaction. Pokrovsky and Schott (2000) and Rosso and Rimstidt (2000) reports a reaction order of $n_{H+}=0.5$ for forsterite, but others report $n_{H+}=1.0$ (Grandstaff 1986, Blum and Lasaga 1988, Siegel and Pfannkuch 1984, Sverdrup 1990). $n_{H+}=1.0$ seems to be a property of the nesosilicate group, but there is a possibility that presence of impurities such as pyroxenes or feldspars in the nesosilicate may give it a different crystal structure and thus a different $n_{H+}$. Others, Berg and Banwart (2000), report $n_{H+}$ in the range 0.5 to 1, depending on pH.

Table 4 shows the temperature dependencies of the dissolution rates. All variations of rates on temperature are computed using a modified Arrhenius equation (Sverdrup 1990, 1998, Sverdrup and Warfvinge 1988, 1992, 1995). Parameters for this equation generated from experimentally measured rates are shown in bold. Where experimental data were not available estimates were computed and scaled with Madelung crystal lattice site energy from garnet (Sverdrup 1990, Velbel 1999). Values in normal font were estimated from the lattice energies and the properties of the mineral surface. Table 5 shows the stoichiometry of the minerals considered in this study.





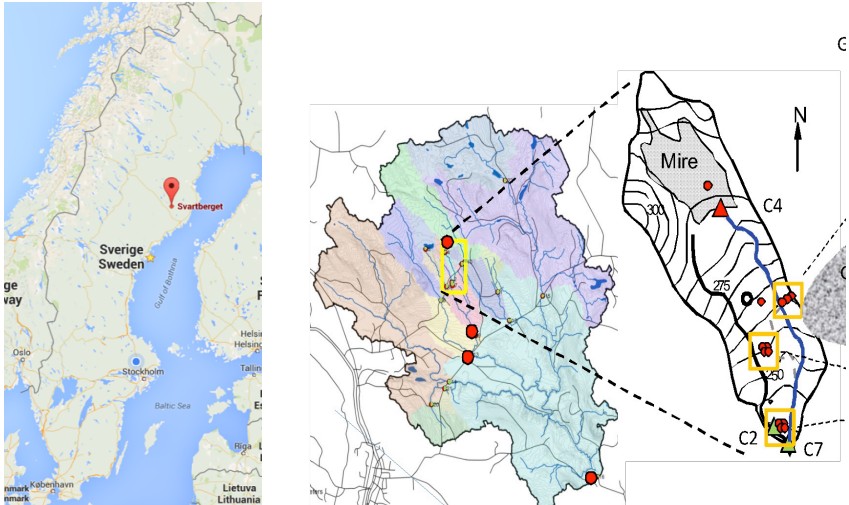

*Figure 26. Location of the research site in Northern Sweden. The colours delineate the different subcatchments of the Svartberget research area. The map on the left shows the catchment co nsidered in this comparison.*

### 5.2. Testing the model

The most recent comparison between model results and field observations follows in the article by Erlandsson-Lampa et al. (This issue). The research catchment where many of the model applications are focused is located in Northern Sweden (Figure 26). A few examples are shown in Figure 27 and 28. Figure 27 shows a comparison between calculated and observed base cation concentrations at the Svartberget research site. The model takes into account all the soil processes such as ion exchange, vegetation interactions, decomposition of organic matter, transport in the soil of the catchment in horizontal and vertical directions and weathering. The model results reproduces the observed concentration pattern (Zanchi et al., 2016). Figure 27a shows the modelled Bc[3] and Figure 27b shows the Si concentrations, plotted against $\log_{10}$ of water transit time (smooth lines). Overlaid are the observed base cation (Bc) and Si-concentrations from the soil profile, plotted against $\log_{10}$ of soil depth (solid lines with markers). The weathering model considers all soil processes including ion exchange, vegetation interactions, decomposition of organic matter, water transport in the catchment in both the horizontal and vertical directions (Belyazid et al., 2004, 2011a,b, 2010a,b, 2015, 2019, Erlandsson-Lampa et al., 2019, Sverdrup et al., 1995, 2002). The model reproduces the observed field observations as a function of depth (Zanchi et al., 2016). The close correspondence between the calculated dissolved metal concentrations and the field observation are notable considering that we employed a simple silicate dissolution rate model to determine the composition of the aqueous phase in the soil.

### 6.3. Discussion

The detailed comparisons between laboratory measured and field determined weathering rates generated using the kinetic models coupled to soil processes performed using PROFILE and ForSAFE stand out in stark contrast to the traditional geochemical models, which give results that are several orders of magnitude off (Erlandsson-Lampa et al., 2019). It was discovered that past efforts to describe field weathering rates using laboratory measured dissolution rates without consideration of its coupling to the major soil processes yielded inaccurate results (Model types represented by codes such as PHRQKIN and similar codes) – see Erlandsson Lampa et al. (2016) and Nyström-Claesson and Andersson, (1996). Such observations demonstrate a need for a new approach that takes into account the complete set of processes occurring in the soil.

---

[3]Bc is the base cations that the plants take up; Ca+Mg+K, BC is Na+K+Ca+Mg.



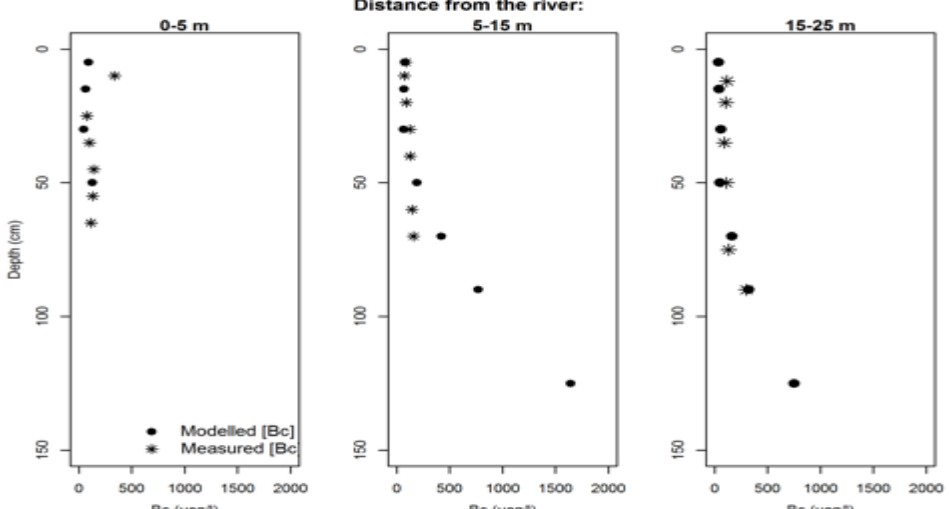

*Figure 27. Comparison of calculated with measured base cation concentrations at the Svartberget field site, (Zanchi et al., 2016). Note the base cation concentrations refer to the sum of the concentrations of Na, H, Ca, and Mg in units of microequivalents per litre.*

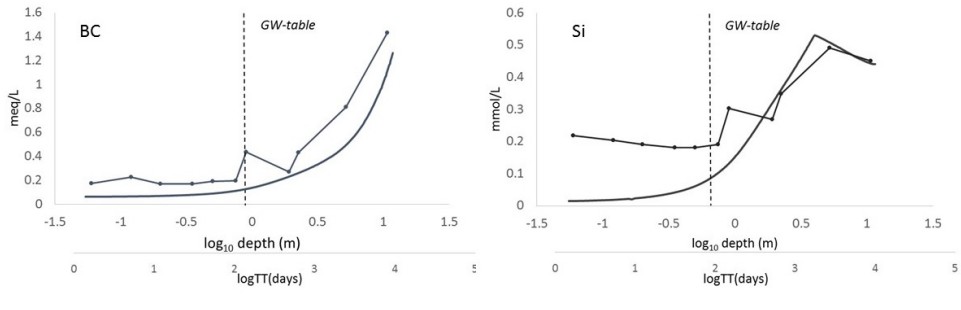

*Figure 28. Modelled base cation (a) and Si (b) concentrations from plotted against $\log_{10}$ of water transit time (smooth lines) at the Svartberget field site (See Erlandsson-Lampa et al., 2016, 2019 for a full description of the field test of the model). Overlain are the observed base cation and Si-concentrations from the soil profile, plotted against $\log_{10}$ of soil depth (straight lines with symbols).*

Note that the weathering brakes used in this approach act differently on the weathering rates that the equilibrium expressions used in earlier models (Aagaard and Helgeson 1982, Murphy et al., 1987, Alekseyev et al., 1997, 2004, 2007, Oelkers 2001, Oelkers et al., 1994, 2001, 2008). The preference for using the brakes rather than the traditional saturation expression based on an assumption of equilibrium between the surface and the liquid, is that the weathering process is irreversible. Thus, an equilibrium assumption is not permitted. The earlier models, lacked the representation of an ecosystem the soils. This is likely the major reason why the earlier approaches failed to estimate field weathering rates.

**7. Conclusions**

The complex nature of weathering in the field is nearly impossible to interpret without a comprehensive model for the whole process. A first step to such interpretations can be the quantitative description of the dissolution rates of then major rock forming minerals. Even the dissolution rates of an individual mineral can involve several simultaneous reactions. Thus, experimentally measured rates results can




only be accurately interpreted when a full system model is used . Under field conditions, the mineral dissolution is coupled to other soil processes, and thus a full ecosystem system model is needed for their interpretation. The apparent difference between field and laboratory dissolution rates arise from the coupling of these processes, and disappear once a full model is employed. Use of a fully coupled model shows these differences to be negligible (Keegan and Laskow-Lehey 2014).

Taking account the vast literature reporting experimentally measured mineral dissolution rates, it was possible to create a fully parameterized kinetic database for about 92 minerals. About 40% of the kinetic parameters were determined directly from experiment interpretations, and the rest was with inter-mineral interpolations and using of analogues.

The adjustment of aluminium 'brake functions' and the introduction of a silica "brake function" as described in this work were necessary to improve the description of weathering rates in the lower part of the soil, below 1 meter depth. The test at the Svartberget catchment suggests that this revised mineral dissolution model works adequately as can be seen from Figures 28-29.

## 8. Acknowledgement

This work rest on the work of Prof. Dr. Harald Sverdrup and Prof. Dr. Per Warfvinge that initiated the new model approaches a long time ago, and did the initial but long experiments that ran without stop 1984-1996. It also rests on the tireless work of Dr. Matthias Alveteg, Prof. Per Warfvinge, Dr. Cecilia Akselsson, Dr. Salim Belyazid and Daniel Kurz in making the model code operational and useful since 1988. Dr. Johan Holmqvist and Harald Sverdrup were instrumental in taking up the second long campaign in weathering experiments, generating more kinetic data with all the required considerations in place 1997-2004. Dr. Salim Belyazid is the present head code editor of the PROFILE and ForSAFE models.

This study was a part of the QWARTZ Project, coordinated by Prof. Kevin Bishop, Uppsala University, Sweden. Dr. Salim Beliazid, Natural Geography and Quaternary Geology, Stockholm University, Sweden, Dr. Martin Erlandsson Lampa, Institute of Hydrology, University of Uppsala, Sweden, Dr. Cecilia Akselsson, Earth Sciences, Lund University, Lund, Sweden, Daniel Kurz, EKG Geoscience, Bern, Switzerland, Dr. Max Posch, CCE, RIVM, Bilthoven, Netherlands, Dr Julian Aherne, Ecology, University of Trent, Canada, Dr. Jennifer Phelan, RTI Inc, Triangle Park, North Carolina, United States of America and Professor Harald Sverdrup, Industrial Engineering, University of Iceland, Reykjavik, Iceland (Earlier at Lund University) took part in the parameterization workshops, with the aim to have this updated kinetics database completed.

Professor Dr.. Eric Oelkers was external advisor to the project, which turned out to be a good choice. He participated very willingly, eagerly and with excellent advice to the research process and in writing this paper.

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





Table 3. Dissolution kinetics parameterization for the 103 minerals from 12 major mineral structural groups that can be used by the PROFILE and ForSAFE models for estimating the field soil weathering rate. expressed as the value at 8°C. Many of the minerals can be grouped into closely related crystallographic groups where many analogues are possible. C is the limiting concentration for retarders in the format $C*10^{-6}$ kmol/m$^3$. Data was also taken from earlier unpublished experimental data from the earlier weathering experiments by Sverdrup and Holmqvist's laboratory experimental archives (These are kept by Sverdrup in University of Iceland. Numbers in **bold** are based on an estimate based on an experiment. All other are in some was estimated from interpolation or analogues. The table was revised at workshop at Ystad Saltsjöbad, April 11-14, 2016. Reassessed during 2017 and 2018 by integrating more information from the scientific literature. Nothing fundamental changed after 2016, but some numbers are backed up better.

Fundamental chemical weathering reaction coefficients, reaction orders, and feedback effect threshold concentrations

| Mineral | | H+-reaction | | | | | | H2O-reaction | | | | | | | CO2-reaction[4] | | Organic acids | | | OH-reaction | | | | | | | |
|---|---|---|---|---|---|---|---|---|---|---|---|---|---|---|---|---|---|---|---|---|---|---|---|---|---|---|---|
| | | pkH | nH | yAl | CAl | xBC | CBC | pkH2O | yAl | CAl | xBC | CBC | zSi | CSi | pkCO2 | nCO2 | pkOrg | nOrg | COrg | pkOH- | wOH- | yAl | xBC | CAl | CBC | zSi | CSi |
| **1a. Feldspars; techtosilicates** | | | | | | | | | | | | | | | | | | | | | | | | | | | |
| 1.1 | K-Feldspar, generic | 14.7 | 0.5 | 0.4 | 0.4 | 0.4 | 0.5 | 17.5 | 0.14 | 4 | 0.15 | 0.5 | 3 | 900 | 16.95 | 0.6 | 15.0 | 0.5 | 5 | 15.2 | 0.3 | 0.1 | 0.5 | 12 | 5 | 1 | 900 |
| 1.2 | K-Feldspar I. | 14.8 | 0.5 | 0.4 | 0.4 | 0.4 | 0.5 | 17.8 | 0.14 | 4 | 0.15 | 0.5 | 3 | 900 | 17.05 | 0.6 | 15.1 | 0.5 | 5 | 15.4 | 0.3 | 0.1 | 0.5 | 12 | 5 | 1 | 900 |
| 1.3 | K-Feldspar II. | 14.7 | 0.5 | 0.4 | 0.4 | 0.4 | 0.5 | 17.4 | 0.15 | 4 | 0.15 | 0.5 | 4 | 900 | 16.85 | 0.6 | 13.9 | 0.5 | 5 | 15.3 | 0.3 | 0.1 | 0.5 | 12 | 5 | 1 | 900 |
| 1.4 | K-Feldspar III | 14.7 | 0.5 | 0.4 | 0.4 | 0.4 | 0.5 | 17.4 | 0.15 | 4 | 0.15 | 0.5 | 4 | 900 | 16.80 | 0.6 | 13.9 | 0.5 | 5 | 15.2 | 0.3 | 0.1 | 0.5 | 12 | 5 | 1 | 900 |
| 1.5 | Anorthoclase | 13.6 | 0.6 | 0.4 | 0.5 | 0.4 | 0.5 | 17.2 | 0.15 | 5 | 0.15 | 0.5 | 3 | 900 | 16.65 | 0.6 | 13.7 | 0.5 | 5 | 14.2 | 0.3 | 0.1 | 0.5 | 15 | 5 | 2 | 900 |
| 1.6 | Albite (Ab) | 14.6 | 0.5 | 0.4 | 0.4 | 0.4 | 0.5 | 16.8 | 0.15 | 4 | 0.15 | 1 | 3 | 900 | 16.05 | 0.6 | 14.7 | 0.5 | 5 | 15.4 | 0.3 | 0.1 | 0.5 | 12 | 4 | 3 | 900 |
| 1.7 | Oligoclase | 14.6 | 0.5 | 0.4 | 0.4 | 0.4 | 1 | 16.8 | 0.15 | 4 | 0.15 | 2 | 4 | 900 | 16.05 | 0.6 | 14.7 | 0.5 | 5 | 15.4 | 0.3 | 0.1 | 0.5 | 12 | 4 | 3 | 900 |
| 1.8 | Labradorite | 13.9 | 0.5 | 0.3 | 0.5 | 0.4 | 2 | 16.8 | 0.15 | 5 | 0.15 | 2 | 5 | 900 | 16.05 | 0.6 | 14.7 | 0.5 | 5 | 14.5 | 0.3 | 0.1 | 0.5 | 15 | 3 | 3 | 900 |
| 1.9 | Bytownite | 13.8 | 0.6 | 0.3 | 0.6 | 0.4 | 3 | 16.7 | 0.15 | 6 | 0.15 | 3 | 6 | 900 | 15.95 | 0.6 | 14.6 | 0.5 | 5 | 14.4 | 0.3 | 0.1 | 0.5 | 18 | 3 | 3 | 900 |
| 1.10 | Other plagioclase | 14.6 | 0.5 | 0.4 | 0.4 | 0.4 | 1 | 16.8 | 0.15 | 4 | 0.15 | 1 | 4 | 900 | 16.05 | 0.6 | 14.7 | 0.5 | 5 | 15.4 | 0.3 | 0.1 | 0.5 | 12 | 4 | 3 | 900 |
| **1b. Zeolites; techtosilicates** | | | | | | | | | | | | | | | | | | | | | | | | | | | |
| 1.11 | Heulandite | 11.9 | 0.73 | 0.2 | 30 | 0.2 | 20 | 16.8 | 0.15 | 4 | 0.15 | 20 | 3 | 900 | 16.05 | 0.6 | 14.7 | 0.5 | 5 | 14.8 | 0.3 | 0.1 | 0.5 | 12 | 4 | 2 | 900 |
| 1.12 | Analcime | 14.5 | 0.5 | 0.2 | 30 | 0.2 | 20 | 16.5 | 0.15 | 4 | 0.15 | 20 | 3 | 900 | 16.05 | 0.6 | 14.7 | 0.5 | 5 | 12.4 | 0.4 | 0.1 | 0.5 | 12 | 4 | 2 | 900 |
| 1.13 | Clinoptilolite | 14.5 | 0.3 | 0.3 | 30 | 0.2 | 20 | 16.5 | 0.15 | 4 | 0.15 | 20 | 3 | 900 | 16.05 | 0.6 | 14.7 | 0.5 | 5 | 14.8 | 0.3 | 0.1 | 0.5 | 12 | 4 | 2 | 900 |
| 1.14 | Stilbite | 14.5 | 0.3 | 0.2 | 30 | 0.2 | 20 | 16.2 | 0.15 | 4 | 0.15 | 20 | 3 | 900 | 16.05 | 0.6 | 14.7 | 0.5 | 5 | 14.7 | 0.3 | 0.1 | 0.5 | 12 | 4 | 2 | 900 |
| **2. Nesosilicates** | | | | | | | | | | | | | | | | | | | | | | | | | | | |
| 2.1 | Monticellite | 7.7 | 0.55[6] | 0.1 | 100 | 0.3 | 50 | >16.4 | 0 | 100 | 0.2 | 50 | 16 | 900 | 15.4 | 0.6 | 13.9 | 0.5 | 5 | 13.3 | 0.6 | 0.1 | 0.2 | 100 | 60 | 14 | 900 |
| 2.2 | Tephroite[5] | 9.3 | 0.56[6] | 0.1 | 100 | 0.3 | 50 | >17.0 | 0 | 100 | 0.2 | 50 | 16 | 900 | 15.4 | 0.6 | 13.3 | 0.5 | 5 | 13.3 | 0.6 | 0.1 | 0.2 | 100 | 60 | 14 | 900 |
| 2.3 | Nepheline[5] | 9.5 | 1.0 | 0.4 | 10 | 0.4 | 10 | 14.4 | 0.2 | 10 | 0.2 | 10 | 6 | 900 | 14.8 | 0.6 | 14.4 | 0.5 | 5 | 13.0 | 0.5 | 0.1 | 0.2 | 30 | 30 | 4 | 900 |
| 2.4 | Anorthite[6] (An) | 10.3 | 1.0 | 0.4 | 100 | 0.2 | 3 | 15.8 | 0.15 | 100 | 0.2 | 200 | 6 | 900 | 16.4 | 0.6 | >13.9[6] | 0.5 | 5 | 13.7[6] | 0.25 | 0.1 | 0.2 | 30 | 30 | 4 | 900 |
| 2.5 | Forsterite (Fo) | 10.2 | 1.0[6] | 0.1 | 1000 | 0.3 | 10 | 16.4 | 0 | 5000 | 0.2 | 5 | 16 | 900 | 15.4[7] | 0.6 | 14.7 | 0.5 | 5 | 13.3[6] | 0.6 | 0.1 | 0.2 | 100 | 60 | 14 | 900 |

[4] There seems to be some type of CO$_2$ saturation of the surface between 10 and 50 atm CO$_2$ for mica and chlorites, beyond where the rate is no more affected. Some other minerals have indications of similar behaviour, but it remains elusive in terms of parameterization. Some minerals appear to have no detectable reaction with CO$_2$, some are slightly inhibited.

[5] Nepheline is classified as a feldspatoid in the mineralogical literature. However, when dissolving, the pre-dissolution complexing process at the mineral water interface create an activated surface complex with a nesosilicate structure. Thus, a nesosilicate classification here.

[6] Anorthite is classified as a feldspar in the mineralogical literature. However, when dissolving pure anorthite, the pre-dissolution complexing process at the mineral water interface create an activated surface complex with a nesosilicate structure. This applied only to pure anorthite with less than 2% other feldspars in the solution. That is why it is listed among the nesosilicates. See Sverdrup (1990) for further details. This may be thecase with Monticellite and Tephroite as well.

[7] According to Golubev et al., (2005) is the CO$_2$ reaction either very weak or absent, mostly from observations at high 5. For diopside and forsterite over the whole pH range (Golubev et al., 2005).





**Nesosilicates (continued)**

| Mineral | pkH | nH | yAl | CAl | xBC | CBC | pkH2O | yAl | CAl | xBC | CBC | zSi | CSi | pkCO2 | nCO2 | pkOrg | nOrg | COrg | pkOH | wOH- | yAl | CAl | xBC | CBC | zSi | CSi |
|---|---|---|---|---|---|---|---|---|---|---|---|---|---|---|---|---|---|---|---|---|---|---|---|---|---|---|
| 2.6 Olivine (Fo60Fa40) | **12.0** | **1.0[8]** | 0.3 | 30 | 0.3 | 30 | **>18.0** | 0.1 | 30 | 0.2 | 5 | 16 | 900 | **15.9ᵇ** | 0.6 | **14.7** | 0.5 | 5 | n.d | 0.6 | 0.1 | 100 | 0.2 | 60 | 14 | 900 |
| 2.7 Fayalite (Fa) | **10.2** | **1.0[6]** | 0.1 | 1000 | 0.3 | 50 | 16.4 | 0 | 5000 | 0.2 | 5 | 16 | 900 | 15.4 | 0.6 | 13.9 | 0.5 | 5 | 13.3 | 0.6 | 0.1 | 100 | 0.2 | 60 | 14 | 900 |
| 2.8 Al44Py44Gr12 | | | | | | | | | | | | | | | | | | | | | | | | | | |
| 2.9 Al65Py35 | | | | | | | | | | | | | | | | | | | | | | | | | | |
| 2.10 Ad6Gr20 | **12.4** | **1.0** | 0.4 | 300 | 0.2 | 50 | **16.9** | 0.2 | 300 | 0.2 | 500 | 8 | 900 | 15.8 | 0.6 | 14.7 | 0.5 | 5 | **14.9** | **0.2** | 0.12 | 100 | 0.2 | 100 | 6 | 900 |
| 2.11 Al50Py40Gr10 | | | | | | | | | | | | | | | | | | | | | | | | | | |
| 2.12 Gr88Py6Ad6 | | | | | | | | | | | | | | | | | | | | | | | | | | |
| 2.13 Grossular, (Gr) | | | | | | | | | | | | | | | | | | | | | | | | | | |
| 2.14 Andradite (Ad) | | | | | | | | | | | | | | | | | | | | | | | | | | |
| 2.15 Pyrope (Py) | | | | | | | | | | | | | | | | | | | | | | | | | | |
| 2.16 Almandine (Al) | **12.4** | **1.0** | 0.4 | 200 | 0.2 | 40 | **16.9** | 0.2 | 200 | 0.2 | 300 | 8 | 900 | 15.8 | 0.6 | 14.7 | 0.5 | 5 | 14.9 | 0.2 | 0.12 | 60 | 0.2 | 60 | 6 | 900 |
| 2.17 Uvarovite (Uv) | | | | | | | | | | | | | | | | | | | | | | | | | | |
| 2.18 Spessarite (Sp) | | | | | | | | | | | | | | | | | | | | | | | | | | |
| 2.19 Staurolite | **14.7** | **1.0** | 0.4 | 200 | 0.2 | 20 | 17.4 | 0.2 | 200 | 0.3 | 5 | 16 | 900 | 15.2 | 0.6 | 14.4 | 0.5 | 5 | **17.1** | **0.3** | 0.12 | 60 | 0.2 | 60 | 14 | 900 |
| 2.20 Disthene | **15.5** | **1.0** | 0.33 | 10 | 0 | 500 | 17.0 | 0.33 | 10 | 0 | 500 | 4 | 900 | 16.5 | 0.5 | 15.6 | 0.5 | 5 | 15.8 | 0.4 | 0.1 | 400 | 0.3 | 60 | 3 | 900 |
| 2.21 Kyanite | | | | | | | | | | | | | | | | | | | | | | | | | | |

**3. Pyroxenes[9] or single band inosilicates.**

| Mineral | pkH | nH | yAl | CAl | xBC | CBC | pkH2O | yAl | CAl | xBC | CBC | zSi | CSi | pkCO2 | nCO2 | pkOrg | nOrg | COrg | pkOH | wOH- | yAl | CAl | xBC | CBC | zSi | CSi |
|---|---|---|---|---|---|---|---|---|---|---|---|---|---|---|---|---|---|---|---|---|---|---|---|---|---|---|
| 3.1 Alite | **9.6** | **0.67** | 0.2 | 1000 | 0.3 | 200 | 7.85 | 0.1 | 400 | 0.3 | 5 | 16 | 900 | n.d | n.d | n.d | n.d | n.d | n.d | n.d | n.d | n.d | n.d | n.d | n.d | n.d |
| 3.2 Wollastonite | **9.6** | 0.7 | 0 | 5000 | 0.3 | 100 | 15.1 | 0 | 5000 | 0.3 | 5 | 16 | 900 | **15.2** | **0.6** | 13.5 | 0.5 | 5 | **11.6** | **0.6** | 0 | 5000 | 0.5 | 5 | 8 | 900 |
| 3.3 Spodumene | **9.6** | 0.7 | 0.2 | 400 | 0.3 | 200 | 17.2 | 0.1 | 400 | 0.3 | 5 | 16 | 900 | 15.8 | 0.6 | 14.2 | 0.5 | 5 | 14.6 | 0.6 | 0.1 | 400 | 0.5 | 5 | 8 | 900 |
| 3.4 Diopside | **11.1** | **0.67** | 0.2 | 400 | 0.35 | 150 | 14.9 | 0.1 | 400 | 0.3 | 5 | 16 | 900 | **>14.8ᵇ** | 0.6 | 16.4 | 0.5 | 5 | **13.2** | **0.6** | 0 | 400 | 0.5 | 5 | 8 | 900 |
| 3.5 Jadeite | **11.2** | 0.7 | 0.2 | 400 | 0.35 | 150 | 14.5 | 0.1 | 400 | 0.3 | 5 | 16 | 900 | 14.4 | 0.6 | 14.0 | 0.5 | 5 | 12.9 | 0.6 | 0 | 400 | 0.5 | 5 | 8 | 900 |
| 3.6 Leucite | **11.1** | 0.4 | 0.2 | 400 | 0.35 | 150 | 14.5 | 0.1 | 400 | 0.3 | 5 | 16 | 900 | 14.4 | 0.6 | 14.0 | 0.5 | 5 | 12.9 | 0.6 | 0.1 | 400 | 0.5 | 5 | 8 | 900 |
| 3.7 Augite I | **12.3** | 0.7 | 0.2 | 500 | 0.3 | 200 | 17.5 | 0.1 | 500 | 0.3 | 5 | 16 | 900 | 15.8 | 0.6 | 14.4 | 0.5 | 5 | 14.8 | 0.6 | 0.1 | 500 | 0.5 | 5 | 8 | 900 |
| 3.8 Augite II | **12.3** | 0.7 | 0.2 | 500 | 0.2 | 200 | 17.5 | 0.1 | 500 | 0.3 | 5 | 16 | 900 | 15.8 | 0.6 | 14.4 | 0.5 | 5 | 14.8 | 0.6 | 0.1 | 500 | 0.5 | 5 | 8 | 900 |
| 3.9 Hedenbergite | **12.8** | 0.7 | 0.25 | 500 | 0.3 | 200 | 17.5 | 0.16 | 500 | 0.3 | 5 | 16 | 900 | 15.8 | 0.6 | 14.4 | 0.5 | 5 | 14.8 | 0.6 | 0.1 | 400 | 0.5 | 5 | 8 | 900 |
| 3.10 Augite II | **13.8** | 0.7 | 0.2 | 400 | 0.2 | 200 | 17.5 | 0.1 | 400 | 0.3 | 5 | 16 | 900 | 15.8ᵇ | 0.6 | 14.4 | 0.5 | 5 | 14.8 | 0.6 | 0.1 | 400 | 0.5 | 5 | 8 | 900 |
| 3.11 Enstatite | **13.0** | 0.7 | 0.2 | 400 | 0.2 | 100 | 17.6 | 0.1 | 400 | 0.3 | 5 | 16 | 900 | 15.8 | 0.6 | 14.5 | 0.5 | 5 | **15.0** | **0.6** | 0.1 | 400 | 0.5 | 5 | 8 | 900 |
| 3.12 Hypersthene | **13.2** | 0.7 | 0.2 | 400 | 0.2 | 100 | 17.7 | 0.1 | 400 | 0.3 | 5 | 16 | 900 | 15.8 | 0.6 | | 0.5 | 5 | 15.0 | 0.6 | 0.1 | 400 | 0.5 | 5 | 8 | |
| 3.13 Ferrosilite | 14.0 | 0.7 | 0.2 | 400 | 0.3 | 200 | | | 400 | 0.3 | 5 | 16 | 900 | 15.8ᵇ | 0.6 | 14.5 | 0.5 | 5 | 14.8 | 0.6 | 0.1 | 400 | 0.5 | 5 | 8 | 900 |
| 3.14 Bronzite | **14.4** | 0.7 | 0.2 | 400 | 0.2 | 200 | 17.5 | 0.1 | 400 | 0.3 | 5 | 16 | 900 | 15.8 | 0.6 | 14.4 | 0.5 | 5 | 14.8 | 0.6 | 0.1 | 500 | 0.5 | 5 | 8 | 900 |
| 3.15 Pidgeonite | **13.8** | | 0.2 | 400 | 0.3 | 200 | 17.5 | 0.1 | 400 | 0.3 | 5 | 16 | 900 | 15.8ᵇ | 0.6 | 14.4 | 0.5 | 5 | 14.8 | 0.6 | 0.1 | 400 | 0.5 | 5 | 8 | 900 |
| 3.16 Other pyroxenes | 14.0 | 0.7 | 0.2 | 500 | 0.3 | 200 | 17.5 | 0.1 | 500 | 0.3 | 5 | 16 | 900 | 15.8 | 0.6 | 14.4 | 0.5 | 5 | 14.8 | 0.6 | 0.1 | 500 | 0.5 | 5 | 8 | 900 |

**4. Amphiboles or double band inosilicates**

| Mineral | pkH | nH | yAl | CAl | xBC | CBC | pkH2O | yAl | CAl | xBC | CBC | zSi | CSi | pkCO2 | nCO2 | pkOrg | nOrg | COrg | pkOH | wOH- | yAl | CAl | xBC | CBC | zSi | CSi |
|---|---|---|---|---|---|---|---|---|---|---|---|---|---|---|---|---|---|---|---|---|---|---|---|---|---|---|
| 4.1 Glaucophane | **13.5** | 0.7 | 0.3 | 5 | 0.3 | 5 | **16.7** | 0.6 | 15 | 0.3 | 200 | 16 | 900 | 16.1 | 0.6 | 14.7 | 0.5 | 5 | >16.7 | 0.3 | 0.15 | 400 | 0.5 | 60 | 8 | 900 |
| 4.2 Pargasite | 13.8 | 0.7 | 0.3 | 5 | 0.3 | 5 | 16.6 | 0.6 | 15 | 0.3 | 200 | 16 | 900 | 16.1 | 0.6 | 14.7 | 0.5 | 5 | >16.7 | 0.3 | 0.15 | 400 | 0.5 | 60 | 8 | 900 |
| 4.3 Hornblende I | **13.4** | 0.7 | 0.4 | 5 | 0.3 | 5 | **16.3** | 0.6 | 15 | 0.3 | 200 | 16 | 900 | 15.9ᵇ | 0.6 | 14.4 | 0.5 | 5 | **17.5** | **0.1** | 0.15 | 400 | 0.5 | 60 | 8 | 900 |
| 4.4 Hornblende II | **14.8** | 0.6 | 0.3 | 5 | 0.3 | **5** | 16.5 | **0.6** | 15 | 0.3 | **200** | 16 | 900 | 16.1ᵇ | 0.6 | **14.5** | **0.5** | 5 | 18.2 | 0.1 | 0.15 | 400 | 0.5 | 60 | 8 | 900 |

[8] A number of studies report this exponent to be 0.5. It was observed that all nesosilicates have reaction order n=1 in our own experiments, and in about half of all in the literature.

[9] He=Hedenbergite, En=Enstatite, Wo=Wollastonite, Di=Diopside, Au=Augite, Ja=Jadeite, Le=Leucite, Bz= Bronzite




### 5. Phyllosilicates or sheet silicates

| Mineral | | H+-reaction pkH | nH | yAl | CAl | xBC | CBC | H2O-reaction pkH2O | yAl | CAl | xBC | CBC | zSi | CSi | CO2-reaction pkCO2 | nCO2 | Organic acids pkOrg | nOrg | COrg | OH-reaction pkOH | wOH | yAl | CAl | xBC | CBC | zSi | CSi |
|---|---|---|---|---|---|---|---|---|---|---|---|---|---|---|---|---|---|---|---|---|---|---|---|---|---|---|---|
| 4.5 | Tremolite | 15.2 | 0.2 | 0.2 | 5 | 0.3 | 5 | 16.8 | 0.6 | 15 | 0.3 | 200 | 16 | 900 | 16.2 | 0.6 | 14.8 | 0.4 | 5 | 16.1 | 0.3 | 0.15 | 400 | 0.5 | 60 | 8 | 900 |
| 4.6 | Riebeckite | 14.9 | 0.7 | 0.2 | 5 | 0.3 | 5 | 18.4 | 0.6 | 15 | 0.3 | 200 | 16 | 900 | 16.2 | 0.6 | 14.8 | 0.5 | 5 | 16.1 | 0.3 | 0.15 | 400 | 0.5 | 60 | 8 | 900 |
| 4.7 | Anthophyllite | 13.8 | 0.25 | 0.2 | 5 | 0.3 | 5 | 18.4 | 0.6 | 15 | 0.3 | 200 | 16 | 900 | 16.2 | 0.6 | 14.9 | 0.1 | 5 | 16.4 | 0.1 | 0.2 | 400 | 0.5 | 60 | 8 | 900 |
| 4.8 | Other amphiboles | 14.8 | 0.6 | 0.3 | 5 | 0.3 | 5 | 16.5 | 0.6 | 15 | 0.3 | 200 | 16 | 900 | 16.1 | 0.6 | 14.5 | 0.5 | 5 | 18.2 | 0.1 | 0.15 | 400 | 0.5 | 60 | 8 | 900 |
| 5.1 | Glauconite | 11.8 | 0.7 | 0.4 | 4 | 0.2 | 500 | 17.0 | 0.2 | 50 | 0.1 | 200 | 16 | 900 | 14.5 | 0.5 | 14.5 | 0.5 | 5 | 15.5 | 0.4 | 0.15 | 400 | 0.3 | 60 | 14 | 200 |
| 5.2 | Serpentinite, Antigotite Chrysotile | 12.7 | 0.8 | 0.2 | 50 | 0.2 | 200 | 17.5 | 0.1 | 50 | 0.1 | 200 | 16 | 900 | 14.8 | 0.5 | >14.1 | 0.5 | 5 | 17.8 | 0.6 | 0.15 | 400 | 0.3 | 60 | 14 | 200 |
| 5.3 | Talc | 13.3 | 0.7 | 0.2 | 50 | 0.2 | 200 | 16.7 | 0.1 | 50 | 0.1 | 200 | 16 | 900 | 14.5 | 0.5 | 14.5 | 0.5 | 5 | 15.5 | 0.4 | 0.15 | 400 | 0.3 | 60 | 14 | 200 |
| 5.4 | Nontronite | 14.8 | 0.3 | 0.2 | 30 | 0.2 | 20 | 16.5 | 0.15 | 4 | 0.15 | 250 | 3 | 900 | 16.05 | 0.6 | 14.7 | 0.5 | 5 | 15.4 | 0.3 | 0.1 | 12 | 0.3 | 4 | 2 | 900 |
| 5.5 | Phlogopite | 14.8 | 0.6 | 0.3 | 10 | 0.2 | 50 | 16.7 | 0.2 | 10 | 0.2 | 500 | 6 | 900 | 15.8 | 0.5 | 15.8 | 0.5 | 5 | 15.8 | 0.5 | 0.15 | 400 | 0.3 | 60 | 5 | 900 |
| 5.6 | Biotite[10] | 14.8 | 0.6 | 0.3 | 10 | 0.2 | 50 | 16.7 | 0.2 | 10 | 0.1 | 50 | 6 | 900 | 15.8 | 0.5 | 15.8 | 0.5 | 5 | 15.8[5] | 0.5 | 0.15 | 400 | 0.3 | 60 | 3 | 50 |
| 5.7 | Mg-Vermiculite[14] | 14.8 | 0.6 | 0.4 | 4 | 0.2 | 5 | 17.2 | 0.1 | 4 | 0.1 | 500 | 4 | 900 | 16.2 | 0.5 | 15.2 | 0.5 | 5 | 15.8 | 0.5 | 0.15 | 400 | 0.3 | 60 | 3 | 50 |
| 5.8 | Mg-Vermiculite 2[4] | 14.8 | 0.6 | 0.4 | 4 | 0.2 | 5 | 17.2 | 0.1 | 4 | 0.1 | 500 | 4 | 900 | 16.2 | 0.5 | 15.2 | 0.5 | 5 | 18.8 | 0.5 | 0.15 | 400 | 0.3 | 60 | 3 | 50 |
| 5.9 | Mg-Vermiculite 3[4] | 14.8 | 0.6 | 0.4 | 4 | 0.2 | 5 | 17.2 | 0.1 | 4 | 0.1 | 200 | 3 | 900 | 16.2 | 0.5 | 15.2 | 0.5 | 5 | 18.8 | 0.5 | 0.15 | 400 | 0.3 | 60 | 3 | 50 |
| 5.10 | Fe-vermiculite | 15.2 | 0.6 | 0.4 | 4 | 0.2 | 50 | 17.6 | 0.1 | 4 | 0.2 | 200 | 3 | 900 | 16.5 | 0.5 | 15.2 | 0.5 | 5 | 18.8 | 0.5 | 0.15 | 400 | 0.3 | 60 | 3 | 50 |
| 5.11 | Illitic vermiculite | 15.0 | 0.6 | 0.4 | 4 | 0.1 | 5 | 17.3 | 0.1 | 4 | 0.1 | 500 | 4 | 900 | 16.5 | 0.5 | 15.5 | 0.5 | 5 | 17.0 | 0.5 | 0.15 | 400 | 0.3 | 60 | 3 | 50 |
| 5.12 | Vermiculite Al-OH interlayer mineral | 15.2 | 0.5 | 0.4 | 4 | 0.1 | 5 | 17.5 | 0.2 | 4 | 0.1 | 500 | 6 | 900 | 16.5 | 0.5 | 15.6 | 0.5 | 5 | 17.2 | 0.4 | 0.15 | 400 | 0.3 | 60 | 5 | 100 |
| 5.13 | Fe-Chlorite | 14.8 | 0.7 | 0.2 | 50 | 0.2 | 5 | 17.0 | 0.1 | 50 | 0.1 | 200 | 4 | 900 | 16.2 | 0.5 | 15.0 | 0.5 | 5 | 18.3 | 0.4 | 0.15 | 400 | 0.3 | 60 | 3 | 50 |
| 5.14 | Chlorite | 14.8 | 0.5 | 0.2 | 50 | 0.2 | 5 | 17.0 | 0.1 | 50 | 0.1 | 200 | 4 | 900 | 16.2 | 0.5 | 12.6 | 0.5 | 5 | 18.0 | 0.4 | 0.15 | 400 | 0.3 | 60 | 3 | 50 |
| 5.15 | Mg-Chlorite | 14.3 | 0.7 | 0.2 | 50 | 0.2 | 200 | 16.7 | 0.1 | 50 | 0.1 | 200 | 4 | 900 | 15.8 | 0.5 | 14.5 | 0.5 | 5 | 18.0 | 0.4 | 0.15 | 400 | 0.3 | 60 | 3 | 50 |
| 5.16 | Smectites[11] | 14.9 | 0.5 | 0.4 | 4 | 0.2 | 500 | 17.6 | 0.2 | 4 | 0.1 | 50 | 4 | 900 | 16.5 | 0.5 | 15.6 | 0.5 | 5 | 17.5 | 0.5 | 0.1 | 400 | 0.3 | 60 | 3 | 50 |
| 5.17 | Muscovite[3] | 15.2 | 0.5 | 0.4 | 4 | 0.1 | 5 | 17.5 | 0.2 | 4 | 0.1 | 500 | 12 | 900 | 16.5 | 0.5 | 15.3 | 0.5 | 5 | 17.2 | 0.4 | 0.15 | 400 | 0.3 | 60 | 10 | 100 |
| 5.18 | Mixed muscovites | 15.1 | 0.5 | 0.4 | 4 | 0.1 | 5 | 17.5 | 0.2 | 4 | 0.1 | 500 | 12 | 900 | 16.5 | 0.5 | 15.3 | 0.5 | 5 | 17.2 | 0.4 | 0.15 | 400 | 0.3 | 60 | 10 | 100 |
| 5.19 | Illite 1[12] | 15.0 | 0.5 | 0.4 | 4 | 0.1 | 5 | 17.5 | 0.1 | 4 | 0.1 | 500 | 3 | 900 | 16.5 | 0.5 | 15.4 | 0.5 | 5 | 17.2 | 0.4 | 0.15 | 400 | 0.3 | 60 | 3 | 100 |
| 5.20 | Illite 2[3] | 15.2 | 0.5 | 0.4 | 4 | 0.1 | 5 | 17.5 | 0.2 | 4 | 0.1 | 500 | 3 | 900 | 16.5 | 0.5 | 15.6 | 0.5 | 5 | 17.2 | 0.4 | 0.15 | 400 | 0.3 | 60 | 3 | 100 |
| 5.21 | Illite 3[3] | 15.2 | 0.5 | 0.4 | 4 | 0.1 | 5 | 17.5 | 0.2 | 4 | 0.1 | 500 | 3 | 900 | 16.5 | 0.5 | 15.8 | 0.5 | 5 | 17.2 | 0.4 | 0.15 | 400 | 0.3 | 60 | 3 | 100 |
| 5.22 | Bentonite | 15.1 | 0.5 | 0.4 | 4 | 0.2 | 500 | 17.6 | 0.2 | 4 | 0.1 | 50 | 4 | 900 | 16.5 | 0.5 | 15.6 | 0.5 | 5 | 17.5 | 0.5 | 0.1 | 400 | 0.3 | 60 | 3 | 50 |
| 5.23 | Montmorillonite | 15.1 | 0.5 | 0.4 | 4 | 0.2 | 500 | 17.6 | 0.2 | 4 | 0.1 | 50 | 4 | 900 | 16.5 | 0.5 | 15.6 | 0.5 | 5 | 17.5 | 0.5 | 0.1 | 400 | 0.3 | 60 | 3 | 50 |
| 5.24 | Sericite | 15.2 | 0.5 | 0.4 | 4 | 0.1 | 5 | 17.5 | 0.2 | 4 | 0.1 | 500 | 3 | 900 | 16.5 | 0.5 | 15.6 | 0.5 | 5 | 17.2 | 0.4 | 0.15 | 400 | 0.3 | 60 | 2 | 100 |

### 6. Cyclosilicates

| Mineral | | H+-reaction pkH | nH | yAl | CAl | xBC | CBC | H2O-reaction pkH2O | yAl | CAl | xBC | CBC | zSi | CSi | CO2-reaction pkCO2 | nCO2 | Organic acids pkOrg | nOrg | COrg | OH-reaction pkOH | wOH | yAl | CAl | xBC | CBC | zSi | CSi |
|---|---|---|---|---|---|---|---|---|---|---|---|---|---|---|---|---|---|---|---|---|---|---|---|---|---|---|---|
| 6.1 | Tourmaline | 13.2 | 1.0 | 0.3 | 200 | 0.2 | 200 | 15.4 | 0.2 | 200 | 0.3 | 100 | 8 | 900 | 14.8 | 0.6 | 14.4 | 0.5 | 5 | >17.0 | 0.5 | 0.15 | 400 | 0.3 | 60 | 8 | 30 |
| 6.2 | Cordierite | 15.4 | 1.0 | 0.3 | 200 | 0.2 | 200 | 16.5 | 0.2 | 200 | 0.3 | 100 | 8 | 900 | 15.9 | 0.6 | 15.5 | 0.5 | 5 | 17.4 | 0.5 | 0.15 | 400 | 0.3 | 60 | 8 | 30 |

### 7. Sorosilicates

| Mineral | | H+-reaction pkH | nH | yAl | CAl | xBC | CBC | H2O-reaction pkH2O | yAl | CAl | xBC | CBC | zSi | CSi | CO2-reaction pkCO2 | nCO2 | Organic acids pkOrg | nOrg | COrg | OH-reaction pkOH | wOH | yAl | CAl | xBC | CBC | zSi | CSi |
|---|---|---|---|---|---|---|---|---|---|---|---|---|---|---|---|---|---|---|---|---|---|---|---|---|---|---|---|

[10] All biotite and vermiculites have the same lattice breakdown rate (Sverdrup and Holmqvist 2004), the release rate results from the combination of lattice breakdown kinetics and the mineral stoichiometry

[11] All smectites, including montmorillonites and bentonites have the same lattice breakdown rate (Sverdrup and Holmqvist 2004), the release rate results from the combination of lattice breakdown kinetics and the mineral stoichiometry

[12] All muscovite and illites have the same lattice breakdown rate (Sverdrup and Holmqvist 2004), the release rate results from the combination of lattice breakdown kinetics and the mineral stoichiometry.





**Table 4.** Temperature dependencies, measured are in bold. Default values were computed and scaled with Madelung crystal lattice site energy from Garnet (Sverdrup 1990). Normal font means we have estimated it from the lattice energies and the properties of the mineral surface. Based on the modified Arrhenius equation (Sverdrup 1990, 1998, Sverdrup and Warfvinge 1988, 1992, 1995).

| Mineral | H+-reaction | | | | | | H2O-reaction | | | | | | | CO2-reaction | | Organic acids | | | OH-reaction | | | | | | | |
|---|---|---|---|---|---|---|---|---|---|---|---|---|---|---|---|---|---|---|---|---|---|---|---|---|---|---|
| | $pk_H$ | $n_H$ | $y_{Al}$ | $C_{Al}$ | $x_{BC}$ | $C_{BC}$ | $pk_{H2O}$ | $y_{Al}$ | $C_{Al}$ | $x_{BC}$ | $C_{BC}$ | $z_{Si}$ | $C_{Si}$ | $pk_{CO2}$ | $n_{CO2}$ | $pk_{Org}$ | $n_{Org}$ | $C_{Org}$ | $pk_{OH}$ | $w_{OH-}$ | $y_{Al}$ | $C_{Al}$ | $x_{BC}$ | $C_{BC}$ | $z_{Si}$ | $C_{Si}$ |
| 7.1 Epidote (Ep) | **14.0** | **0.8** | **0.3** | **50** | **0.2** | **5** | **17.7** | **0.2** | **50** | **0.2** | **20** | **32** | **900** | 16.2 | 0.5 | 14.4 | 0.5 | 5 | **18.4** | **0.2** | **0.15** | **400** | **0.3** | **60** | **32** | **200** |
| 7.2 Zoisite (Zo) | **15.2** | **0.5** | 0.2 | 50 | 0.2 | 5 | 17.4 | 0.2 | 200 | 0.2 | 20 | 32 | 900 | 16.3 | 0.5 | 14.7 | 0.5 | 5 | 17.2 | 0.3 | 0.15 | 400 | 0.3 | 60 | 32 | 200 |
| 7.3 Other zoisites | 15.2 | 0.5 | 0.2 | 50 | 0.2 | 5 | 17.4 | 0.2 | 200 | 0.2 | 20 | 32 | 900 | 16.3 | 0.5 | 14.7 | 0.5 | 5 | 17.2 | 0.3 | 0.15 | 400 | 0.3 | 60 | 32 | 200 |

**8. Aluminosilicates and quartz**

| Mineral | $pk_H$ | $n_H$ | $y_{Al}$ | $C_{Al}$ | $x_{BC}$ | $C_{BC}$ | $pk_{H2O}$ | $y_{Al}$ | $C_{Al}$ | $x_{BC}$ | $C_{BC}$ | $z_{Si}$ | $C_{Si}$ | $pk_{CO2}$ | $n_{CO2}$ | $pk_{Org}$ | $n_{Org}$ | $C_{Org}$ | $pk_{OH}$ | $w_{OH-}$ | $y_{Al}$ | $C_{Al}$ | $x_{BC}$ | $C_{BC}$ | $z_{Si}$ | $C_{Si}$ |
|---|---|---|---|---|---|---|---|---|---|---|---|---|---|---|---|---|---|---|---|---|---|---|---|---|---|---|
| 8.1 Kaolinite | **15.1** | **0.7** | 0.4 | 4 | 0.4 | 5 | 17.6 | 0.2 | 5 | 0.4 | 50 | 2 | 900 | 16.5 | 0.5 | **19.5** | **0.5** | 5 | >15.1 | 0.6 | 0.15 | 400 | 0.3 | 60 | 1 | 900 |
| 8.2 Gibbsite | **13.9** | **1.0** | 0.5 | 5 | 0 | 500 | 16.4 | 0.2 | 5 | 0.4 | 0 | n.a. | n.a. | >18.0 | 0.5 | 16.3 | 0.5 | 5 | >13.4 | 1.0 | 0 | 5 | 0 | 5000 | n.a. | n.a. |
| 8.3 Quartz | **18.4** | **0.3** | 0.3 | 5 | 0 | 500 | >17.8 | 0 | 5 | 0 | 5000 | 4 | 900 | >18.0 | 0.5 | **16.3** | **0.5** | 5 | **14.1** | 0.3 | 0.4 | 200 | 0 | 5000 | 1 | 900 |

**9. Volcanic glasses**

| Mineral | $pk_H$ | $n_H$ | $y_{Al}$ | $C_{Al}$ | $x_{BC}$ | $C_{BC}$ | $pk_{H2O}$ | $y_{Al}$ | $C_{Al}$ | $x_{BC}$ | $C_{BC}$ | $z_{Si}$ | $C_{Si}$ | $pk_{CO2}$ | $n_{CO2}$ | $pk_{Org}$ | $n_{Org}$ | $C_{Org}$ | $pk_{OH}$ | $w_{OH-}$ | $y_{Al}$ | $C_{Al}$ | $x_{BC}$ | $C_{BC}$ | $z_{Si}$ | $C_{Si}$ |
|---|---|---|---|---|---|---|---|---|---|---|---|---|---|---|---|---|---|---|---|---|---|---|---|---|---|---|
| 9.1 Base cation poor volcanic glass | **15.2** | **0.5** | 0.4 | 5 | 0.1 | 300 | 18.2 | 0.1 | 5 | 0 | 50 | 2 | 900 | 17.9* | 0.5 | **15.7** | **0.5** | 5 | **15.7** | **0.25** | 0.25 | 5 | 0.3 | 60 | 2 | 900 |
| 9.2 Base cation rich volcanic glass | **15.2** | **0.5** | 0.4 | 5 | 0.1 | 300 | 18.2 | 0.1 | 5 | 0 | 50 | 2 | 900 | 17.9* | 0.5 | 19.5 | 0.5 | 5 | **15.8** | **0.25** | 0.25 | 5 | 0.3 | 60 | 2 | 900 |
| 9.3 Other glasses | 15.2 | 0.5 | 0.4 | 5 | 0.1 | 300 | 18.2 | 0.1 | 5 | 0 | 50 | 2 | 900 | 17.9 | 0.5 | 19.5 | 0.5 | 5 | **15.8** | **0.25** | 0.25 | 5 | 0.3 | 60 | 2 | 900 |

**10. Carbonates**

| Mineral | $pk_H$ | $n_H$ | $y_{Al}$ | $C_{Al}$ | $x_{BC}$ | $C_{BC}$ | $pk_{H2O}$ | $y_{Al}$ | $C_{Al}$ | $x_{BC}$ | $C_{BC}$ | $z_{Si}$ | $C_{Si}$ | $pk_{CO2}$ | $n_{CO2}$ | $pk_{Org}$ | $n_{Org}$ | $C_{Org}$ | $pk_{OH}$ | $w_{OH-}$ | $y_{Al}$ | $C_{Al}$ | $x_{BC}$ | $C_{BC}$ | $z_{Si}$ | $C_{Si}$ |
|---|---|---|---|---|---|---|---|---|---|---|---|---|---|---|---|---|---|---|---|---|---|---|---|---|---|---|
| 10.1 Calcite[13] | **13.6** | **1.0** | 0 | 5000 | **0.4** | **5** | **14.2** | 0 | 5000 | **0.2** | 1000 | 16 | 900 | **13.2** | **0.6** | **13.2** | **0.5** | **5** | **0** | **0** | 0 | 5000 | 0 | **5000** | 16 | 900 |
| 10.2 Aragonite | **13.6** | **1.0** | 0 | 5000 | 0.4 | 5 | **14.6** | 0 | 5000 | 0.2 | 1000 | 16 | 900 | **13.4** | 0.6 | **13.4** | 0.5 | 5 | 0 | 0 | 0 | 5000 | 0 | 5000 | 16 | 900 |
| 10.3 Dolomite | **11.1** | **0.5** | 0 | 3000 | **0.4** | **5** | **17.5** | 0 | 3000 | **0.2** | **10** | 4 | 900 | **14.8** | **0.6** | **14.4** | **0.5** | 5 | **0** | **0** | 0 | 5000 | 0 | 5000 | 4 | 900 |
| 10.4 Magnesite | **13.1** | **0.5** | 0 | 3000 | 0.4 | 5 | **17.6** | 0 | 3000 | 0.2 | **5** | 3 | 900 | 14.8 | 0.6 | 14.4 | 0.5 | 5 | 0 | 0 | 0 | 5000 | 0 | 5000 | 3 | 900 |
| 10.5 Siderite[14] | 15.4 | 0.74 | 0 | 3000 | 0.4 | 5 | 18.8 | 0 | 3000 | 0.2 | 10 | 8 | 900 | 14.8 | 0.6 | 14.4 | 0.5 | 5 | 0 | 0 | 0 | 5000 | 0 | 5000 | 4 | 900 |
| 10.6 Rhodochrosite[11] | 15.6 | 0.67 | 0 | 3000 | 0.4 | 5 | 18.6 | 0 | 3000 | 0.2 | 10 | 8 | 900 | 14.6 | 0.6 | 14.2 | 0.5 | 5 | 0 | 0 | 0 | 5000 | 0 | 5000 | 4 | 900 |

**11. Phosphates**

| Mineral | $pk_H$ | $n_H$ | $y_{Al}$ | $C_{Al}$ | $x_{BC}$ | $C_{BC}$ | $pk_{H2O}$ | $y_{Al}$ | $C_{Al}$ | $x_{BC}$ | $C_{BC}$ | $z_{Si}$ | $C_{Si}$ | $pk_{CO2}$ | $n_{CO2}$ | $pk_{Org}$ | $n_{Org}$ | $C_{Org}$ | $pk_{OH}$ | $w_{OH-}$ | $y_{Al}$ | $C_{Al}$ | $x_{BC}$ | $C_{BC}$ | $z_{Si}$ | $C_{Si}$ |
|---|---|---|---|---|---|---|---|---|---|---|---|---|---|---|---|---|---|---|---|---|---|---|---|---|---|---|
| 11.1 Apatite[15] | **12.8** | **0.67** | 0 | - | **0.4** | **100** | **16.1** | 0.2 | 20 | **0.4** | **50** | n.a. | n.a. | **15.8** | **0.6** | **19.5** | **0.5** | 5 | **12.8** | **0.6** | **0.15** | **400** | **0.3** | **60** | n.a. | n.a. |
| 11.2 Fluoroapatite | 12.8 | 0.7 | 0 | - | 0.4 | 100 | 15.9 | 0.2 | 20 | 0.4 | 50 | n.a. | n.a. | 15.8 | 0.5 | 19.5 | 0.5 | 5 | 12.8 | 0.5 | 0.15 | 400 | 0.3 | 60 | n.a. | n.a. |
| 11.3 Other soil phosphorus solids | 12.8 | 0.7 | 0 | - | 0.4 | 100 | 15.8 | 0.2 | 20 | 0.4 | 50 | n.a. | n.a. | 15.8 | 0.5 | 19.5 | 0.5 | 5 | **12.8** | **0.5** | 0.15 | 400 | 0.3 | 60 | n.a. | n.a. |

| Mineral | Fundamental chemical reactions | | | |
|---|---|---|---|---|
| | $H_2O$ | $CO_2$ | Organic acids | $OH^-$ |
| $H^+$ | | | | |

Comments

[13] This is a general calcite. Accurate kinetic data are available for 8 different Swedish and 6 different American commercially available calcites, and 4 different Swedish, English, Finnish and Estonian dolomites (See Sverdrup and Bjerle 1983).

[14] Siderite and rhodocrosite have strong inhibition of the water reaction by dissolved oxygen in the solution.

[15] Apatite dissolution is retarded at all pH by oxalate concentrations and the presence of aluminium and iron. Silica seems to interfere less with the rate of dissolution.



| ID | Name | | | | | | | Reaction |
|---|---|---|---|---|---|---|---|---|
| **1. Feldspars** | | | | | | | | |
| 1.1-1.2 | K-Feldspar I; Orthoclase, Sanidine | 3500 | 1940 | 3200 | 1200 | 1700 | 3200 | Irreversible dissolution |
| 1.3 | K-Feldspar II; Microcline | 3470 | 1820 | 3200 | 1200 | 1700 | 3200 | Irreversible dissolution |
| 1.4 | K-Feldspar III; Orthoclase | 4090 | 2000 | 3500 | 1200 | 1700 | 3500 | Irreversible dissolution |
| 1.5 | Anorthoclase | 3500 | 2000 | 3200 | 1200 | 1680 | 3200 | Irreversible dissolution |
| 1.6 | Plagioclase; Albite | 3350 | 2500 | 3100 | 1200 | 1700 | 3100 | Irreversible dissolution |
| 1.7 | Plagioclase; Oligoclase | 4200 | 2330 | 3600 | 1200 | 1700 | 3600 | Irreversible dissolution |
| 1.8 | Plagioclase; Labradorite | 4200 | 2500 | 3500 | 2200 | 1700 | 3500 | Irreversible dissolution |
| 1.9-1.10 | Plagioclase; Bytownite and near anorthite | 3500 | 2500 | 3100 | 1200 | 1700 | 3100 | Irreversible dissolution |
| 1.11 | All other feldspars | 3685 | 2085 | 3100 | 1200 | 1690 | 3100 | Irreversible dissolution |
| **1b. Zeolites** | | | | | | | | |
| 1.12 | Heulandite | 3500 | 2550 | 3450 | 1200 | 1700 | 3450 | Irreversible dissolution |
| 1.13 | Analcime | 3500 | 2500 | 3400 | 1200 | 1700 | 3400 | Reversible reaction |
| 1.14 | Clinoptilolite | 3500 | 2550 | 3600 | 1200 | 1700 | 3600 | Irreversible dissolution |
| 1.15 | Stilbite | 3500 | 2500 | 3400 | 1200 | 1700 | 3400 | Irreversible dissolution |
| **2. Nesosilicates** | | | | | | | | |
| 2.1 | Monticellite | 3480 | 4200 | 2200 | 1600 | 1700 | 2200 | Irreversible dissolution |
| 2.2 | Tephroite | 2551 | 4400 | 1450 | 1534 | 1700 | 1450 | Irreversible dissolution |
| 2.4 | Anorthite (An) | 1820 | 5670 | 1700 | 1800 | 1700 | 1700 | Irreversible dissolution |
| 2.5 | Forsterite (Fo) | 3350 | 4510 | 2100 | 1800 | 1700 | 2100 | Irreversible dissolution |
| 2.6 | Olivine | 2580 | 4510 | 2100 | 1800 | 1700 | 2100 | Irreversible dissolution |
| 2.7 | Fayalite | 2550 | 4400 | 2200 | 1800 | 1700 | 2200 | Irreversible dissolution |
| 2.13 | Nepheline | 3630 | 3130 | 2180 | 1800 | 1700 | 2180 | Irreversible dissolution |
| 2.8-2.18 | Garnet mixes, all garnets | 2500 | 3500 | 2000 | 1800 | 1700 | 2000 | Irreversible dissolution |
| 2.19 | Staurolite | 3100 | 3200 | 3100 | 1800 | 1700 | 3100 | Irreversible dissolution |
| 2.20-2.21 | Disthene, Kyanite | 3918 | 2400 | 2200 | 1800 | 1700 | 2200 | Irreversible dissolution |
| 2.22 | All other nesosilicates | 2676 | 4436 | 2180 | 1800 | 1700 | 2180 | Irreversible dissolution |
| **4. Pyroxenes** | | | | | | | | |
| 3.2 | Wollastonite | 3100 | 3600 | 2100 | 2000 | 1700 | 2100 | Irreversible dissolution |
| 3.4 | Diopside | 2610 | 3400 | 2000 | 2000 | 1700 | 2000 | Irreversible dissolution |
| 3.9 | Hedenbergite | 2311 | 3500 | 2000 | 2000 | 1700 | 2000 | Irreversible dissolution |
| 3.7-3.8, 3.10 | Augite | 2700 | 4100 | 2000 | 2000 | 1700 | 2000 | Irreversible dissolution |
| 3.11 | Enstatite | 2550 | 5950 | 2000 | 2000 | 1700 | 2000 | Irreversible dissolution |
| 3.16 | All other pyroxenes | 2700 | 4100 | 2000 | 2000 | 1700 | 2000 | Irreversible dissolution |
| **4. Amphiboles** | | | | | | | | |
| 4.1 | Glaucophane | 4300 | 3800 | 3500 | 2000 | 1700 | 3500 | Irreversible dissolution |
| 4.2 | Hornblende I | 4300 | 3800 | 3500 | 2000 | 1700 | 3500 | Irreversible dissolution |
| 4.3 | Hornblende II | 4300 | 4000 | 3500 | 2200 | 1800 | 3500 | Irreversible dissolution |
| 4.4 | Tremolite | 4500 | 3390 | 3600 | 2000 | 1700 | 3600 | Irreversible dissolution |
| 4.5 | Antophyllite | 3800 | 3300 | 4500 | 2200 | 1700 | 4500 | Irreversible dissolution |
| 4.6 | All other amphiboles | 4300 | 3390 | 3500 | 2000 | 1700 | 3500 | Irreversible dissolution |
| **5. Phyllosilicates** | | | | | | | | |
| 5.1 | Glauconite | 4300 | 1950 | 3500 | 2000 | 1700 | 3500 | Irreversible dissolution |
| 5.2 | Serpentinite, | 4282 | 3600 | 3500 | 2000 | 1700 | 3500 | Irreversible dissolution |



| ID | Mineral | | | | | | Comment |
|---|---|---|---|---|---|---|---|
| | Chrysotile, Antigorite | | | | | | |
| 5.3 | Talc | 4200 | 3700 | 1700 | 2000 | 3500 | Irreversible dissolution |
| 5.4 | Nontronite | 4500 | 3500 | 1700 | 1200 | 3400 | Irreversible dissolution |
| 5.6 | Biotite | 4500 | 3840 | 1700 | 2000 | 3500 | Irreversible dissolution |
| 5.5 | Phlogopite | 4500 | 3840 | 1700 | 2000 | 3500 | Irreversible dissolution |
| 5.7 | Vermiculite 1 | 4500 | 3840 | 1700 | 2000 | 3500 | Alteration mineral, irreversible dissolution |
| 5.8 | Vermiculite 2 | 4500 | 3840 | 1700 | 2000 | 3500 | Alteration mineral, irreversible dissolution |
| 5.9 | Vermiculite 3 | 4500 | 3840 | 1700 | 2000 | 3500 | Irreversible dissolution |
| 5.10 | Fe-Chlorite | 4500 | 3800 | 1700 | 2000 | 3500 | Irreversible dissolution |
| 5.14 | Fe-Mg-Chlorite | 4520 | 3500 | 1700 | 1800 | 3500 | Irreversible dissolution |
| 5.17 | Mg-Chlorite | 4500 | 1400 | 1700 | 1700 | 3500 | Irreversible dissolution |
| 5.19 | Muscovite | 3038 | 3800 | 1700 | 2000 | 4656 | Irreversible dissolution |
| 5.21 | Illite 1 | 4500 | 3800 | 1700 | 2000 | 3500 | Alteration mineral, irreversible dissolution |
| 5.22 | Illite 2 | 4500 | 3800 | 1700 | 2000 | 3500 | Alteration mineral, irreversible dissolution |
| 5.23 | Illite 3 | 4500 | 3800 | 1700 | 2000 | 3500 | Irreversible dissolution |
| 5.24 | Montmorillonite | 4300 | 3840 | 1700 | 2000 | 3500 | Alteration mineral, irreversible dissolution |
| 5.27 | All other phyllosilicates | 4410 | 3770 | 1700 | 2000 | 3500 | Irreversible dissolution |
| **7. Cyclosilicates** | | | | | | | |
| 6.1 | Tourmaline | 3600 | 3100 | 1700 | 1800 | 2500 | Irreversible dissolution |
| 6.2 | Cordierite | 2600 | 5900 | 1700 | 2000 | 2000 | Irreversible dissolution |
| 6.3 | All other cyclosilicates | 3100 | 4500 | 1700 | 1900 | 2250 | Irreversible dissolution |
| **8. Sorosilicates** | | | | | | | |
| 7.1 | Epidote | 5330 | 3800 | 1700 | 2000 | 2300 | Irreversible dissolution |
| 7.2 | Zoisite | 4400 | 3900 | 1800 | 2200 | 3300 | Irreversible dissolution |
| 73 | All other sorosilicates | 4375 | 3850 | 1750 | 2100 | 3300 | Irreversible dissolution |
| **10. Oxides and simple aluminosilicates** | | | | | | | |
| 8.1 | Kaolinite | 5310 | 3580 | 1700 | 2000 | 4100 | Irreversible dissolution, gibbsite possible outcome |
| 8.2 | Gibbsite | 3400 | 3600 | 1700 | 2000 | 3170 | Alteration mineral, irreversible dissolution |
| 8.3 | Quartz | 3890 | n.a. | 2200 | 2000 | 3320 | Reversible reactions, back reaction, dissolution is kinetically limited |
| **11. Volcanic glasses** | | | | | | | |
| 9.1 | Volcanic glass, base cation poor | 3890 | 3010 | 2400 | 2800 | 2700 | Irreversible dissolution |
| 9.2 | Volcanic glass, base cation rich | 4500 | 3310 | 2500 | 2800 | 3400 | Irreversible dissolution |
| 9.3 | All other volcanic glasses | 4200 | 3110 | 2450 | 2800 | 3050 | Irreversible dissolution |
| **10 Carbonates** | | | | | | | |
| 10.1 | Calcite and limestones | 444 | 1180 | 2180 | 2200 | - | Reversible reaction. Back reaction important |
| 10.2 | Aragonite | 530 | 1210 | 2200 | 2400 | - | Reversible reaction. Back reaction important |
| 10.3 | Dolomite | 1880 | 2700 | 1800 | 2200 | - | Irreversible dissolution. Back reaction to calcite and magnesite |
| 10.5 | Siderite | 3300 | 3500 | 1700 | 2000 | 2500 | Irreversible dissolution |
| 10.6 | Rhodochrosite | 3300 | 3500 | 1700 | 2000 | 2500 | Irreversible dissolution |
| **11 Phosphates** | | | | | | | |
| 11.1 | Apatite | 3500 | 4000 | 1700 | 1200 | 2500 | Irreversible dissolution, precipitates with oxalate and aluminium important |
| 11.2 | Fluoroapatite | 1110 | 4790 | 1700 | 1200 | 2500 | Irreversible dissolution, precipitates with oxalate and aluminium important |
| 11.3 | Immobilized inorganic phosphorus, all other phosphorus | 2350 | 4000 | 1700 | 1200 | 2200 | Possibly reversible reaction |





Table 5. Stoichiometry of the minerals applied in Tables 3 and 4.

### 1a. Feldspars

| | Mineral | Formula |
|---|---|---|
| 1.1 | K-Feldspar | $KAlSi_3O_8$  =  Or |
| 1.2 | K-Feldspar I: Orthoclase, K-Feldspar I; Sanidine, 100-90% | $Or_{97}An_3$ |
| 1.3 | K-Feldspar II: 90%, Microcline | $Or_{97}Ab_2An_1$ |
| 1.4 | K-Feldspar II: 80%, Orthoclase | $Or_{80}Ab_{20}$ |
| 1.5 | Anorthoclase | $Or_{20}Ab_{62}An_{17}$ |
| 1.6 | Albite | $NaAlSi_3O_8$  =  Ab |
| 1.7 | Plagioclase: Oligoclase | $Ab_{85}An_{15}$ |
| 1.8 | Plagioclase; Labradorite | $Ab_{46}An_{54}$ |
| 1.9 | Plagioclase; Bytownite | $Ab_{22}An_{78}$ |
| 1.10 | Plagioclase; feldparic Anorthite | $Ab_6An_{94}$ |

### 1b. Zeolites with techtosilicate structure

| | Mineral | Formula |
|---|---|---|
| 1.12 | Helulandite | $(Ca,Na)_{0.45}Al_{0.89}Si_{3.1}O_8 \cdot 2.7\ H_2O$ |
| 1.13 | Analcime | $NaAlSi_2O_6 \cdot H_2O$ |
| 1.14 | Clinoptilolite | $(Na,K,Ca)_{2-3}Al_3(Al,Si)_2Si_{13}O_{36} \cdot 12H_2O$ |
| 1.15 | Stilbite | $Na_{0.09}Ca_{0.66}AlSi_3O_8 \cdot 3.1\ H_2O$ |

### 2. Nesosilicates

| | Mineral | Formula |
|---|---|---|
| 2.1 | Monticellite | $CaMgSiO_4$ |
| 2.2 | Tephroite | $Mn_2SiO_4$ |
| 2.3 | Nepheline | $(Na_{0.75}K_{0.25})AlSiO_4$ |
| 2.4 | Anorthite | $CaAl_2Si_2O_8$  =  An |
| 2.5 | Forsterite | $Mg_2SiO_4$ |
| 2.6 | San Carlos, Arizona Forsterite | $Mg_{1.81}Fe_{0.19}SiO_4$ |
| | Salem, Tamil Nadu Indian olivine | $Mg_{1.84}Fe_{0.16}SiO_4$ |
| | Norwegian Olivine ($Fo_{65}Fa_{35}$) | $Mg_{1.5}Fe_{0.35}Al_{0.02}Si_{1.04}O_4$ |
| 2.7 | Fayalite | $Fe_2SiO_4$ |
| 2.8-2.12 | Generic garnet, continuous series | $Al_{44}Py_{44}Gr_{12}, Al_{65}Py_{35}, Ad_{80}Gr_{20}, Al_{50}Py_{40}Gr_{10}, Gr_{88}Py_6Ad_6$ |
| 2.13 | Grossular | $Ca_3Al_2(SiO_4)_3$ |
| 2.14 | Almandine =Al | $Fe_3Al_2(SiO_4)_3$ |
| 2.15 | Spessartine = Sp | $Mn_3Al_2(SiO_4)_3$ |
| 2.16 | Andradite = Ad | $Ca_3Fe_2(SiO_4)_3$ |
| 2.17 | Uvarovite = Uv | $Ca_3Cr_2(SiO_4)_3$ |
| 2.18 | Pyrope = Py | $Mg_3Al_2(SiO_4)_3$ |
| 2.19 | Staurolite | $Mg_{0.2}Fe_{1.2}Al_{7.4}Si_{4.3}O_{22}(OH)_2$ |
| 2.20 | Disthene | $Al_2SiO_5$ |
| 2.21 | Kyanite | $Al_2SiO_5$ |

2.223. Pyroxenes (End members are diopside, hedenbergite, enstatite, ferrosilite)

| | Mineral | Formula |
|---|---|---|
| 3.1 | Alite (T-slag, K-slag) | $Ca_3SiO_5$ or $(CaO)_3SiO_2$ |




| 3.2 | Wollastonite ($Ca_{22}Si_{22}O_6$) | $Ca_{1.7}Mg_{0.11}Si_{2.2}O_6$ |
| 3.3 | Spodumene ($LiAlSi_2O_6$) | $LiAl_{0.86}Fe_{0.3}Si_2O_6$ |
| 3.4 | Diopside ($CaMgSi_2O_6$) | $Ca_{1.04}Mg_{1.0}Al_{0.02}Fe_{0.01}Si_{2.03}O_6$, $Ca_{0.8}Mg_{0.8}Fe_{0.2}Al_{0.2}Si_2O_6$ |
| 3.5 | Jadeite ($NaAlSi_2O_6$) | $Na_{1.0}Ca_{0.2}Fe_{0.3}AlSi_2O_6$ |
| 3.6 | Leucite ($KAlSi_2O_6$) | $Na_{0.05}K_{1.09}Al_{1.15}Si_{2.3}O_6$ |
| 3.7 | Augite I | $He_{55}En_{45}$ |
| 3.8 | Augite II | $En_{51}Wo_{39}He_{10}$ |
| 3.9 | Hedebergite ($CaFeSi_2O_6$) | $Ca_{0.4}Mg_{0.7}Fe_{0.09}Al_{0.15}Si_{1.86}O_6$ |
| 3.10 | Augite III | $Ca_{0.86}Mg_{1.0}Fe_{0.02}Si_2O_6$ |
| 3.11 | Enstatite ($Mg_2Si_2O_6$) | $Mg_{1.7}Fe_{0.3}Si_2O_6$ |
| 3.12 | Hypersthene | $MgFeSi_2O_6$ ($En_{50}Fs_{50}$) |
| 3.13 | Ferrosilite | $Fe_2Si_2O_6$ |
| 3.14 | Bronzite (mixed) | $Mg_{1.54}Fe_{0.42}Ca_{0.2}Si_{1.9}O_6$ ($En_{70}He_{10}Fs_{20}$) |
| 3.15 | Pidgeonite | $Mg_{50}Ca_{15}Fe_{35}Si_2O_6$ |
| 3.16 | Mixed pyroxenes | $Ca_{0.8}Mg_{0.9}Fe_{0.3}Al_{0.04}Si_2O_6$ ($Di_xEn_yFs_zHe_w$) |
| | **4. Amphiboles** | |
| 4.1 | Glaucophane | $Na_2MgFe_2Al_2Si_8O_{22}(OH)_2$ |
| 4.2 | Pargasite | $NaCa_2(MgAl)(Si_6Al_2)O_{22}(OH)_2$ |
| 4.3 | Hornblende I (Norwegian) | $Ca_{2.1}Mg_{4.5}Na_{0.08}Al_{2.1}Si_7O_{22}(OH)_2(PO_4)_{0.01}$ |
| 4.4 | Hornblende II (Canadian) | $Ca_{2.0}Mg_{4.0}Na_{0.16}Al_{0.4}Si_{8.3}O_{22}(OH)_2$ |
| 4.5 | Tremolite | $Ca_2Mg_5Si_8O_{22}(OH)_2$ |
| 4.6 | Riebeckite | $Na_2Fe^{+2}_3Fe^{3+}_2Si_8O_{20}(OH)_2$ |
| 4.7 | Anthophyllite | $Mg_{5.7}FeAl_{0.1}Si_{7.8}O_{22}(OH)_2$ |
| 4.8 | Other amphiboles | Various compositions |
| | **5. Phyllosilicates** | |
| 5.1 | Glauconite | $(K,Na)(Fe^{3+},Al,Mg)_2(Si,Al)_4O_{10}(OH)_2$ |
| 5.2 | Serpentine, Antigorite, Chrysotile | $Mg_{2.1}Fe_{0.4}Al_{0.15}Si_{2.8}O_{10}(OH)_4$, $\underline{(Mg,\_Fe)_3Si_2O_5(OH)_4}$ |
| 5.3 | Talc | $Mg_{2.8}Fe_{0.18}Si_4O_{10}(OH)_3$ |
| 5.4 | Nontronite | $Ca_{0.5}(Si_7Al_{1.8}Fe_2)(Fe_{3.5}Al_4Mg_{1.1})O_{20}(OH)_4$ |
| 5.5 | Phlogopite | $K_{1.0}Mg_3Al_{1.0}Si_3O_{10}(OH)_2$ |
| 5.6 | Biotite | $K_{0.9}Mg_{1.9}Fe_{1.1}Al_{1.0}Na_{0.1}Si_3O_{10}(OH)_2$ |
| 5.7 | Mg-Vermiculite I | $K_{0.5}Mg_{1.5}Fe_{1.1}Al_{1.7}Na_{0.05}Si_3O_{10}(OH)_2$ |
| 5.8 | Mg-Vermiculite II | $K_{0.3}Mg_{1.7}Fe_{1.1}Al_{1.5}Si_3O_{10}(OH)_2$ |
| 5.9 | Mg-Vermiculite III | $K_{0.1}Mg_{0.5}Fe_{1.1}Al_2Si_3O_{10}(OH)_2$ |
| 5.10 | Fe-Vermiculite | $(Mg,Fe^{+2},Fe^{+3})_3[(Al,Si)_4O_{10}](OH)_2 \cdot 4H_2O$ |
| 5.11 | Illitic vermiculite | $K_{0.35}Mg_{0.11}Ca_{0.03}Al_{2.13}Fe_{0.32}Ti_{0.07}Si_{3.4}O_{10}(OH)_2$ |
| 5.12 | Vermiculite Al-OH interlayer mineral | $(Mg,\_Al,\_Fe^{2+})_3(Si,Al)_4\_O_{10}\_(OH)_2 \cdot nH_2O$ |
| 5.13 | Fe-Chlorite V, Chamosite | $Fe_5Al_2Si_3O_{10}(OH)_8$ |
| 5.14 | Chlorite IV (mixed) | $Mg_{9.7}Fe_{2.7}Al_{2.3}Si_6O_{10}(OH)_8$ |
| 5.15 | Chlorite III (mixed) | $Mg_2Fe_3Al_2Si_3O_{10}(OH)_8$ |



| | | |
|---|---|---|
| 5.16 | Chlorite II (mixed) | $Mg_{4.9}Fe_{0.6}Al_{1.4}Si_3O_{10}(OH)_8$ |
| 5.17 | Mg-Chlorite I, Clinochlore | $Mg_5Al_2Si_3O_{10}(OH)_8$ |
| 5.18 | Smectite | $Ca_{0.2}Mg_{1.0}Na_{0.13}Al_{1.0}Si_4O_{10}(OH)_2$ |
| 5.19 | Muscovite | $KAl_3Si_3O_{10}OH_2$ |
| 5.20 | Muscovite (mixed) | $K_{0.9}Na_{0.02}Mg_{0.3}Fe_{0.4}Al_{2.7}Si_{3.5}O_{10}(OH)_2$ |
| 5.21 | Illite I | $K_{0.6}Mg_{0.28}Fe_{0.3}Al_{2.6}Si_{3.3}O_{10}(OH)_2$ |
| 5.22 | Illite II | $K_{0.7}Mg_{0.26}Fe_{0.1}Al_{2.5}Si_{3.1}O_{10}(OH)_2$ |
| 5.23 | Illite III | $K_{0.6}Mg_{0.25}Al_{2.3}Si_3O_{10}(OH)_2$ |
| 5.24 | Montmorillonite | $Ca_{0.2}Mg_{1.0}Na_{0.13}Al_{1.0}Si_4O_{10}(OH)_2$ |
| 5.25 | Bentonite | See illite |
| 5.26 | Sericite | $KAl_3Si_3O_{10}(OH)_2$ |
| | **6. Cyclosilicate** | |
| 6.1 | Tourmaline | $Ca_{1.0}Fe_3MgAl_5Si_6O_{18}(BO_3)_3(OH)_4(PO_4)_{0.01}$ |
| 6.2 | Cordierite | $Ca_{3.5}Fe_{0.07}K_{0.09}Al_{3.3}Si_{4.6}O_{18}$ |
| | **7. Sorosilicates** | |
| 7.1 | Epidote | $Ca_{1.5}K_{0.46}Fe_{0.74}Al_{1.5}Si_{3.4}O_{12}(OH)$ |
| 7.2 | Zoisite (Clino-) | $Ca_2Fe_{0.13}Al_{1.5}Si_{3.2}O_{12}(OH)$ |
| | **8. Clay minerals** | |
| 8.1 | Kaolinite | $Al_2Si_2O_5(OH)_4$ |
| 8.2 | Gibbsite | $Al(OH)_3$ |
| 8.3 | Quartz | $SiO_2$ |
| | **9. Glasses** | |
| 9.1 | Volcanic glass, base cation poor | $Ca_{0.2}Mg_{0.2}K_{0.4}Na_{0.4}Al_{0.8}Si_3O_8$ |
| 9.2 | Volcanic glass, base cation rich | $Ca_{0.62}Mg_{0.53}K_{0.27}Na_{0.27}Al_{0.66}Si_{2.68}O_8$ |
| | **10. Carbonates** | |
| 10.1a | Calcite (Ca) | $(CaCO_3)_{99.9}(Ca_5(PO_4)_3(OH))_{0.1}$ |
| 10.1b | Köping limestone | $Ca_{97}Do_2Ma_1Ap_{0.1}$ |
| 10.1c | Red Öland limestone | $Ca_{97}Do_1Sd_2Ap_{0.1}$ |
| 10.1d | Ignaberga limestone | $Ca_{50}Ar_{45}Do_1Sd_2Ap_{0.5}$ |
| 10.2 | Aragonite (Ar) | $(CaCO_3)_{99.9}(Ca_5(PO_4)_3(OH))_{0.1}$ |
| 10.3 | Dolomite (Do) | $(CaMg(CO_3)_2)_{99.9}(Ca_5(PO_4)_3(OH))_{0.1}$ |
| 10.4 | Magnesite (Ma) | $MgCO_3$ |
| 10.6 | Rhodochrosite | $MnCO_3$ |
| 10.5 | Siderite (Sd) | $FeCO_3$ |
| | **11. Phosphorus minerals** | |
| 11.1 | Apatite (Ap) | $Ca_5(PO_4)_3(OH)$ |
| 11.2 | Fluoroapatite | $Ca_5(PO_4)_3(OH_{0.7}F_{0.3})$ |
| 11.3 | Immobilized inorganic phosphorus | Unknown, assume as semi-apatite $(Ca_3AlFe_{0.5}Si_5(PO_4)(F_{0.1}OH_{0.4}(CO_3)_{0.5})$ |