# Peer review of "Reviews and synthesis: Weathering of silicate minerals in soils and watersheds: Parameterization of the weathering kinetics module in the PROFILE and ForSAFE models"

_Biogeosciences, 2019_

## Referee Comment (RC1) · Anonymous Referee #1 · 31 Mar 2019

This paper is a by and large clear description of the current state of the PROFILE model (and its derivatives) together with a summary of how the model has evolved over the course of over 20 years.

It links to other papers in this issue that use the revised version of the model.

As such it is fine for publication and is a useful explanation of how the PROFILE model works. It provides necessary background for other papers in which the model is applied.

[Figure]

Although cited in the references, the paper Introduction and section on "Earlier development work and background" makes no reference to the papers coming out of Langan's group in the 1990s that were critical of a number of aspects of the model. This paper is written by the model authors so it is perhaps understandable why they don't refer to papers that have highlighted aspects of the model that need improving but I think this should be done. For example Langan et al. (1997) reviewed the reworking of data from dissolution studies used to generate the kinetic parameters in the profile model (rate constants and reaction orders) and the default mineral compositions used in the model and found both to be inaccurate. As far as I can tell (first paragraph of Section 4 "Theory") this has now been addressed, though due to time constraints I haven't checked any reworking of the data in the many dissolution papers cited in the current manuscript.

Fig. 1 appears to show positive and negative feedback loops – it might be clearer to call these by those names.

Despite the lengthy reference list it appears to be incomplete – e.g. I couldn't find the Sverdrup and Stiernquist 2002 reference listed which is a shame as it appears to be a paper that describe the modelling philosophy of the Sverdrup group

I would argue that the PROFILE model isn't widely used – it is extensively used by Sverdrup and his group – see Steady-state weathering rate models" section

As stated in the paper, when assessing model performance the model authors are "comparing uncertain model estimates with equally or more uncertain field estimates at best". This is a fair comment. As the current manuscript describes the kinetic workings of the model and acknowledges uncertainty, it would be useful to include a comment on that major source of uncertainty facing all weathering calculations – mineral surface area. The core equations in PROFILE, e.g. equation (1a) require a measure of surface area. Surface area normalised dissolution rates are multiplied by the available mineral surface area on a mineral by mineral basis. The model does a good job at

producing weathering rate estimates that approximate those measured in the field but the surface area term remains a challenge. Many applications of the model assume that the distribution of surface area between minerals is equivalent to the distribution of their masses as determined by XRD. From a practical perspective it is hard to see what else can be done but it is always worth acknowledging this issue. Another issue is the total mineral surface area term. In this manuscript it is simply referred to, in previous papers it is suggested that this can be calculated from a consideration of the clay, silt and sand fractions present in samples. I'd encourage the paper authors to suggest how model users should measure this term (geometric, BET or their texture formula) and to comment on the uncertainty this introduces. If they still advocate their texture based equation for calculating surface area I would strongly encourage them to publish the data from which the relationship is derived – they have never done this and the only independent test of this equation by Langan's group showed a poor match between surface areas predicted by that equation and real measurements. The values were all of the same magnitude and so, given the uncertainty in the model, this isn't a particular issue for the model producing rate estimate similar to those measured in the field but it remains a cause for concern that perhaps the surface area term functions as a scaling factor to ensure comparability between modelled and measured rates rather than deriving rates from first principles.

Figures 12 and 13 – it needs to be clearer what the numerical values in the legend refer to

In Section 4.10 "The parameterization of the kinetic rate equations" it is stated "Of these minerals, the regressions of ∼20 have yet to be published. In due time, these will get their own proper publications....". I do understand the sentiment here and the authors have my sympathies but it makes me slightly nervous since it would be good to publish the data and processing of data used in a model before publishing the model as we can't predict whether these regressions really will be published or not. Also, when the original regressions for reaction orders and rate constants were checked by

Langan's group there were errors in them. If the current model uses orders and rates that haven't been published and can't be checked it calls into question confidence in the model – with the caveat that the model does seem to predict mineral weathering rates in the right ball park compared to other models and so is presumably functioning OK. The authors make this point at the end of Section 4 "The ultimate test. . .how well they describe both laboratory experiments and field data. . ..such tests have been generally successful. . .more on this will be forthcoming as the publishing of further comparisons are made" – as above it would be nice to see these comparisons published now though I acknowledge that the paper would be too long if the comparisons were included in it.

There are minor language issues through out.

---

## Referee Comment (RC2) · Anonymous Referee #2 · 23 Apr 2019

The contribution by Sverdrup and co-authors is an extensive description of the PRO-FILE and SAFE models, extending to the more recent SAFE-type models, and focusing on the development of a new parameterization of the dissolution kinetics. The article is rather long (31 pages), with the main part of it devoted to the model description. This is a an unusual format, but it appears that a joint paper has been submitted at the same time to the same volume, showing an in-depth comparison between output of the latest 2D-version of the SAFE model and field measurements (Erlandsson-Lampa et al., 2019).
I think that this exhaustive description of the Sverdrup's models is useful, either for scientists using them for simulations of the field weathering rates, or for scientists developing similar numerical tools. The article is well written and clear. I think two points should be addressed before it can be published.

One thing that confuses me is the way Al3+ and SiO2 are calculated, and the consequences of this methodology on the model output. For Al3+, it is assumed to be a function of the pH only, by forcing the model to be at equilibrium with "something amorphous of unknown composition", potentially looking like "gibbsite" (page 17). This method allows the calculation of the full speciation of Al in solution, without requiring the solving of a mass-balance equation for Al. From my own experience, I know that this assumption makes the model running faster and in a more stable mode, replacing one "critical" differential equation by an algebraic equation. This is a particularity of the SAFE model. I have no specific problem with this. But then, SiO2 is calculated in the same way, by assuming equilibrium with a kind of "kaolinite" mineral. This is a strong assumption, since the natural waters are often largely supersaturated with respect to kaolinite, due to the very low precipitation rate of this mineral. The central question is to what extent this second assumption impacts the model results? I suspects it is not insignificant, given that Al and Si are the major components of the silicate minerals.

My second point is related the "brake" functions. Several other modeling studies (Maher et al., 2009, GCA; Goddéris et al., 2010, GCA) have shown that the development of weathering profiles can be reproduced numerically, assuming that dissolution rates slow down when the percolating waters approach thermodynamic saturation. As noted by Maher et al (2009), this is the only way to account for the difference in the propagation rates of the reaction fronts for two minerals despite their very similar kinetic rate laws. Here Sverdrup and co-authors argue that the saturation of the soil solution with respect to primary minerals does not limit the dissolution rates, because the dissolution process is not reversible. Consequently, the Sverdrup's team has developed several brake functions, which limit dissolution rates and are functions of the Al3+, of the Si

concentration, and of several additional parameters. It looks like introducing empirical inhibiting factors to make up for not considering the effect of chemical affinity, but is it really more efficient than the more classical method? The only way to answer this question is to test a variety of models for the same study site. Given that the previous study also include inhibiting factors, I find that the section 6.3 is somewhat sententious, and I would open the discussion to possible future intercomparisons. Last but not least, I'm wondering whether the need for a new parameterization of the kinetic reactions is not dependent on the way $Al3+$ and $SiO2$ are calculated (previous point). These are not decoupled processes. Forcing the water composition at equilibrium with respect to kaolinite may result in an underestimation of the $Al3+$ and $SiO2$ concentrations in the fluid, explaining (at least partly) why "brake" functions must be defined to slow down dissolution rates in the SAFE model family.

Once these points addressed or discussed, I think this contribution will deserve publication.

---

## Referee Comment (RC3) · Anonymous Referee #3 · 26 Apr 2019

This ms provides an overview of various aspects of the kinetic weathering model in PROFILE and ForSAFE, focussing on kinetic parameterization. The stated purpose is to describe the updated mineral kinetics database used in these models, and revisions to the brakes on weathering reactions in the model. The brakes are discussed at various junctures throughout the document, while the kinetics data are presented largely in a sizeable table appended to the main body.

In addition to the improvements in the brakes, there is value in the conceptual diagrams that provide the reader with a more complete understanding of how the weathering rate

model works (eg. Fig 8,9,14). These are perhaps the clearest characterization of the model to date. Perhaps this could be revisited as an objective of this contribution, as much of what the paper contains is outside of the stated scope. Additionally, another addition to the model is the inclusion of reaction kinetics for OH. It would seem this would enhance the capacity to model weathering rates in agricultural soils, more so than improving rates in much of the acid sensitive areas that have been a focus point of model use to date. It seems this should be discussed, as otherwise this inclusion would have limited impact?

It seems clear that this ms originates as a report that has not had the necessary work done to transform it to a clear and coherent ms. The variety of statements about e.g. historical funding sources, contributions at different times, etc. belong in the acknowledgements and not as informal prose scattered throughout different sections of the paper. Likewise the ms is poorly put together, with mis-numbered sections, figures presented out of order, strange numbering of equations etc. The reference list is both incomplete and not reflective of the citations in the text. There are grammatical issues throughout. In the absence of line numbers, I won't point out these and other line-specific problems. Table formatting and figure captions, which can be inconsistent to the figures themselves, should also be addressed (e.g. fig 17) as should axes for which units are not always provided. Not all figures cited in text are shown.

Additional things to address: It is not adequately explained why the model fails at greater depths. This is referred to as an issue of depth, but since depth is not a parameter than is tied to weathering reactions it seems much more likely that this may be an issue of water table presence. If the water table is the issue, then certainly brakes are a necessary step to improving the rates, but only if the soils are not freely draining. A more comprehensive overview of this is needed, including how the model can address the issue of weathering reactions that occur transiently as soils are wetted after infiltrations, versus deeper soils that may be wet due to the water table but of a very different chemical composition. How are the necessary hydrological dynamics represented?

Most of the first 8 pages, and section 4 and 3 are not needed. This contribution would carry much more potential impact if it were focussed on the improvements to the model rather than rehashing a long-history that is tangential to the stated purpose. The plethora of figures included from previous work add little value.

It is remarkable that comparing uncertain model estimates of weathering with 'equally or more uncertain field estimates' yields error of +/-5%. What is missing from the section on page 5 is that error is often much larger than this, and this is shown in some of the citations in the ms. The sensitivity of the model has been documented in different ways over the last >2 decades. Problematically model sensitivity is strongly linked to parameters that are 'unparameterizable' or parameterized in highly uncertain ways (e.g. Hodson et al. 1997 WASP). It is troubling that this work has been overlooked while focussing on select examples where there was relatively good performance of the model. This raises important questions as to whether the model is producing realistic outcomes for the right reasons.

While the effort here has no doubt resulted in improvements to the model, it is difficult to assess the potential impact of this work as no thorough demonstration of improvements in capacity are illustrated. Considering the companion paper by Erlandsson Lampa will be helpful to this, but it remains unclear what the role of the 'results' and 'discussion' sections are in their current formulation in this ms, as they don't link back to the stated purpose.

There is potential to confuse the reader. The weathering sub-model requires no calibration, but it can only be carried out after calibration of the main ForSAFE model?

Section 4.2 has been presented numerous times elsewhere. What is the rationale for its inclusion here?

Where do data in Table 1 originate?

It is unclear how equation 23 arises from eq 12, but then reverts to eq. 29.

Equation terms are not always defined.

Page 20 and 21. How is something updated to be consistent with previous (rather than new)? It is difficult to discern the message of the remaining part of the paragraph.

Figures 16. Are the lines fitted to the observations, or generated with the model? Figure 19. Are these plots comparing individual base cations, with other cations with base cations as a group? The legend does not provide clarity

Figure 23. Why Figure B? Was this previously published and borrowed here?

The major result of the paper, the provision of an updated kinetic dataset is finally introduced on page 28, and receives modest attention relative to the numerous bits that came before it.

Section 5.2 and testing of the model comes here as a surprise. It was not described as an objective, and appears to be covered comprehensively in the companion paper. Its role here is unclear.

---

## Author Comment (AC1) · 1 Jun 2019

Comment to Reviewer #1.

This article records some of the evolvement of the PROFILE model and the comparable part in ForSAFE, for the last 32 years since its conception in 1987. Since the Langan group did their assessment 20 years ago, much work has been done to change the model. Johan Holmquist works with this in his PhD thesis, for both forest soils (Asa Research Park) and for agricultural soils (Öyebyn Experimental Farm, SLU Experimental

sites at Lanna, Uppsala). Cecilia Akselsson worked on this for her PhD thesis, when taking on how to parameterize both soil mineral surfaces and soil mineral wetting. Both represented significant challenges in terms of how measurement in a point in the landscape is transformed to a value representing a land unit or a landscape. It was not the scope of the article to describe this parameterization process, it has been described in these PhD thesises and in subsequent articles in detail. An empirical relationship between different field estimations and a careful estimation of BET surface (Which in itself required a careful preparation) the estimation of the geometric surface area. This was initially developed for Sweden, the target of our first weathering map. Later this was developed for Switzerland (Reports by D. Kurz for the Swiss Ministry of Environment), by the de Vries research group for (Alterra), Netherlands and Maryland in the United States (Grey reports for Maryland Department of Natural Resources, a scientific publication in 1994 by Sverdrup et al.). This was treated in a dedicated section in the Mapping Manual for Critical Loads by Sverdrup, de Vries and Henriksen, issued by the Nordic Council in 1990. Further, under controlled conditions, the adequacy of using the BET measure under laboratory conditions on pure minerals was shown by Anna Nyström-Claesson at the Nuclear Physics Department in Chalmers University, Göteborg. The weathering rate was estimated with great accuracy under very well controlled conditions. The Langan group tried to develop for Great Britain something similar to the translation of Swedish soil classification translation to soil mineral surfaces. They showed that the Swedish empirical relationship would not give adequate surface area estimates for some British soils, which is not surprising. But, this issue is external to the simulation model. I do remember that the Langan group found some errors in the kinetics 20 years ago. It was update then, a long time ago.

---

## Author Comment (AC2) · 1 Jun 2019

Comment to Reviewer #2

The way Al3+ is calculated has been much discussed over the years. It is generally recognized that this is an empirical equation, and that it can be tweaked to operate reasonably well (Sverdrup, de Vries, Henriksen 1990, Mapping manual for critical loads, Nordic Council of Ministers). It is generally agreed among both soil scientists and modelers that the mineral Gibbsite does not have anything to do with it, it is merely a name

that is used for an empirical equation. For silicate "kaolinite" expression approach used here, it is the same. It is not the mineral kaolinite that is involved, this is a name for an empirical equation. This was discussed by the members at the 2016 Ystad Workshop, where a simple approach and a complicated approach were evaluated. The 2016 Ystad Workshop agreement was to try the simple expression first, and only go to the full mass balance and dissolution and precipitation equations if the simple approach did not give satisfactory results. We have stuck to the workshop conclusions, thus that is why it is like it is. The tests of the model on field data shows that it does work, and this is the judge for whether it is a good approach or not.

―――――――――――――――――

---

## Author Comment (AC3) · 1 Jun 2019

Suzanne Anderson
Department of Geography
University of Colorado
Boulder, CO
UNITED STATES

30 May 2019

Dear Suzanne—

Attached you'll find the revised version of our manuscript 'Reviews and synthesis: Weathering of silicate minerals in soils and watersheds: Parameterization of the weathering kinetics module in the PROFILE and ForSAFE models' by Eric Oelkers, Martin Erlandsson Lampa, Salim Belyazid, Daniel Kurz, Cecilia Akselsson and myself prepared for publication in Biogeochemistry.

We have taken conderation of all of the insightful comments priovided by the reviewers. A detailed list of our response to each of these comments is provided on your website. Notably, we have revised the text for improved clarity, we rechecked all calculations.

From the care of our revisions, we feel this manuscript is improved now ready for publication. We hope you agree.

Thank you in advance for your continued assistance.

Sincerely yours,

Harald Sverdrup
Reykjavik, ICELAND

**Response to reviewer's comments.**

Reviewer 1: This reviewer suggests publication following minor revision and provided the following suggestions.

1. This reviewer suggests that we add some references to the work of Langan and co-workers to the 'Earlier development' section of the original manuscript. ***We agree and have added a reference to Langan et al. (1996) to our revised manuscript.***
2. This reviewer asks us to clarify the positive and negative feedback loops in Fig1 of the original manuscript. ***We agree and have added text to the caption of Figure 1 to clarify this to the reader.***
3. This reviewer suggests that we add the reference to Sverdrup and Stiernquist (2002) to the manuscript. ***We agree and have added this reference to the revised manuscript.***
4. This reviewer asks that we include some details on the major sources of uncertainty facing both the profile and all other weathering calculations, and notably the quantification of surface area. ***This has been added. Much of the uncertainty in the weathering estimates come as a consequence of how the input data was obtained and is thus external to the actual weathering rate model.***
5. This reviewer asks that we clarify the numerical values in Figs 12 and 13. ***In response to this comment we have revised the text describing these figures in the revised manuscript.***
6. This reviewer asks that we do our best to publish the regression information of those minerals yet to be published in section 4.10 of the original manuscript. ***These are quite many and have no room in this manuscript. This will be taken up in detail in future publications.***

Reviewer 2: This reviewer recommends publication after several minor corrections:

1. This reviewer asks for clarification of how $Al^{+3}$ and $SiO_2$ are calculated in the kinetic model. ***We agree and have added text to the revised manuscript to clarify this to the reader.***
2. This reviewer asks that we clarify somewhat the use of break functions in our kinetic model. ***We agree and have added text to the revised manuscript to clarify this to the reader***

Reviewer 3: This reviewer recommends publication after several clarifications.

1. This reviewer asks that we clarify up front that one of the goals of this manuscript is to review and clarify the PROFILE approach. ***We agree and have added two sentences to the beginning of the introduction to clarify these goals to the reader.***
2. This reviewer suggests that we discuss in more detail the break functions and how including a break function for OH might affect model results. ***We***

*agree and have added text to the revised manuscript to clarify this to the reader*

3. This reviewer asks that we move all statements on the amount funding sources into the acknowledgement section. ***We agree and have removed the statements on funding from the original manuscript.***

4. This reviewer suggests we proofread on the manuscript to remove informal statements, some mis-numbered sections, and some grammatical errors. ***This we have done several rounds of proofreading and made a fair number of minor corrections to this revised manuscript.***

5. This reviewer asks us to clarify why the model begins to fail at greater depths, and if these are related to hydrologic issues. ***We agree and have added text to the revised manuscript to clarify these issues to the reader***

6. This reviewer suggests condensing the first 8 pages of the text for improved impact. ***We agree and have condensed somewhat this section as suggested by this reviewer.***

7. This reviewer asks us to clarify the model uncertainty near page 5 of the original manuscript. ***We agree and have added substantial text to the revised manuscript to clarify these uncertainties to the reader.***

8. This reviewer suggest we rewrite the text somewhat to better connect the results and discussion with the stated purpose of the manuscript. ***We agree and have added text to the revised manuscript to clarify this connection to the reader.***

9. This reviewer asks that we clarify what we mean by the weathering sub-model does not require calibration. ***The model does not have adjustable parameters, its input is based on initial field observations. We have added text to the revised manuscript to clarify this to the reader.***

10. This reviewer questions if we could remove section 4.3 of the original manuscript. ***We disagree and have chosen to retain this short section in the revised manuscript.***

11. This reviewer asks for the source of Table 1. ***The original of this table was the Ystad Workshop. We added text to the revised manuscript to clarify this to the reader.***

12. This reviewer asks us to clarify the origin of equation 23 and its relation to equation 29. ***Eq 23 is a simplified version of Eq 29. We have added text to the revised manuscript to clarify this to the reader.***

13. This reviewer asks that we recheck to assure that all the terms of all equations are defined. ***We have gone through the text to verify all symbols have been defined.***

14. This reviewer asks us to revise the text on pages 20 and 21 of the original manuscript for improved clarity. ***We agree and have revised this section for improved clarity.***

15. This reviewer asks if the results on Figure 16 are fit to the observations or model calculations. ***The model was run with the parameters found from experiments and then data from the literature was drawn in. We believe this is clear in the revised manuscript.***

16. This reviewer asks that we improve the caption of Fig. 19 to improve its clarity. ***We agree and have revised this caption for improved clarity.***

17. This reviewer asks why there is a part B in Figure 23 of the original manuscript. ***We agree that this is an oversight and will correct this error in the page proofs.***

18. This reviewer suggests we better emphasize the updated database introduced in page 28 of the original manuscript as it is one of the major results of this work. ***We agree and have added text to clarify this to the reader.***

19. This reviewer asks that we better integrate section 5.2 into the revised manuscript. ***We agree and have revised this section somewhat to better integrate it into the rest of the manuscript.***

---

## Author Comment (AC4) · 1 Jun 2019

**Reviews and synthesis: Weathering of silicate minerals in soils and watersheds: Parameterization of the weathering kinetics module in the PROFILE and ForSAFE models**

Harald Ulrik Sverdrup[1*], Eric H. Oelkers[2,3] Martin Erlandsson Lampa[4],
Salim Belyazid[4], Daniel Kurz[5], Cecilia Akselsson[6],

1-Industrial Engineering, University of Iceland, Reykjavik, Iceland, 2-Earth Sciences, University College London, Gower Street, WC1E 6BT, London, UK, 3-CNRS, UMR 5563, Toulouse, France, 4-Institute of Hydrology, University of Uppsala, Uppsala, Sweden, 5-Physical Geography, Stockholm University, Stockholm, Sweden, 5-EKG Geoscience, Bern, Switzerland, 6-Earth Sciences, University of Lund, Lund, Sweden. *corresponding author (hus@hi.is)

**Abstract**

The PROFILE model, now incorporated in the ForSAFE model can accurately reproduce the chemical and mineralogical evolution of the soil unsaturated zone. However, in deeper soil layers and in groundwater systems, it overestimates weathering rates. This overestimation has been corrected by improving the kinetic expression describing mineral dissolution by adding or upgrading 'breaking functions'. The base cation and aluminium brakes have been strengthened, and an additional silicate brake has been developed, improving the ability to describe mineral-water reactions in deeper soils. These brakes are developed from a molecular-level model of the dissolution mechanisms. Equations, parameters and constants describing mineral dissolution kinetics have now been obtained for 102 different minerals from 12 major structural groups, comprising all types of minerals encountered in most soils. The PROFILE and ForSAFE weathering sub-model was extended to cover two-dimensional catchments, both in the vertical and the horizontal direction, including the hydrology. Comparisons between this improved model and field observations are available in Erlandsson Lampa et al. (2019, This special issue). The results showed that the incorporation of a braking effect of silica concentrations was necessary and helps obtain more accurate descriptions of soil evolution rates at greater depths and within the saturated zone.

**1. Introduction**

This manuscript is aims to review and clarify the chemical weathering approach adopted by the PROFILE and ForSAFE models. In particular, this contribution describes continuing efforts to upgrade the kinetic databases of these models for improved model calculations.

[revised manuscript text omitted]

The accuracy of the model, depends on the parameterization of its input data. Under very controlled circumstances, the model predictions are actually very precise (±5% or better). The main uncertainties when the model is applied to forest plots, watersheds or larger landscape squares. Then the main inaccuracy of the estimates originate in the generalization of soil parameter (such as mineral

surface area) estimates in larger areas, as well as how the outputs (valid for the area used) are interpreted. But these inaccuracies are external to the model (Akselsson et al., 2006, 2005, 2004).

The source of statistical uncertainty in regional weathering estimates were carefully worked out by Walse et al. (1996), and later by Barkman et al. (1996). Holmquist et al. (2020) further elaborate on this using large databases and assessing the geostatistical properties of the landscape. When the model system is used over very many points in a landscape, the uncertainties will to some degree cancel (Holmquist et al., 2002).

[Figure]

*Figure 3. Map of base cation release rates from chemical weathering of soil minerals in the upper 0.5 m of the soil in Sweden determined using the PROFILE model. The model accurately reproduces weathering rates in the upper soil layers, and provides useful estimates for soils of up to 1 meter in thickness. The map was created by Dr. Cecilia Akselsson at Lund University for Swedish forest sustainability assessments and critical loads for acid depositions (Akselsson et al., 2006, 2005, 2004, 2016, Sverdrup et al., 2017).*

[Figure]

*Figure 4. The diagram shows the weathering rate distributed among minerals, the diagram to the right shows the total rate, plotted as the sum of base cations released to the aqueous phase as a function of depth down a soil profile. The diagram to the left shows how selected minerals contribute to this overall rate. The site is catchment F1 at the Gårdsjön Research site, Sweden (Adapted from Sverdrup and Warfvinge 1992, 1995).*

Figure 4 shows an example from the Gårdsjön research site in Sweden (Sverdrup et al., 1992, 1993, 1998). The diagram shows the weathering rate distributed among minerals, and the total rate as a function of depth down a soil profile. The example shows the weathering rate at catchment F1 at the Gårdsjön Research site, Sweden (Sverdrup et al., 1992, 1993, 1996). The research site at Gårdsjön, near Göteborg, Sweden has played a key role in the development of our biogeochemical ecosystem models. The research site is one of Sweden's most important field research sites for soils, soil chemistry, material fluxes, geology, mineralogy, ecology, forestry and environmental pollution research, with nearly all aspects excellently documented and recorded for the last 40 years (Hultberg et al., 2007). Here the soil biogeochemistry models were tested, adapted and used for assessments, using the large amounts of data from this intensively monitored research site. Differences in calculated and observed weathering rates became evident when calculating weathering rates for deeper layers. Notably, the model overestimates the weathering rate at depths below 1-1.5 meters. The reason for the overestimation, is that the brake functions applied earlier in the kinetic model are not strong enough deeper in the soil when the silica concentrations are elevated. At that time no silicate break functions were present in the model.

Figure 5 shows an example of a soil weathering simulation of the weathering rate at Niwot Ridge, Rocky Mountain National Park, Colorado for four different environmental pollution scenarios with background acid deposition, current policy, no pollution control and elevated temperature from climate change. The weathering rate is reduced under the climate change scenario. The weathering rate is somewhat increased by the increase in temperature, but more reduced by reduced rainfall leading to drier soils at the site. The ForSAFE model was used with a daily time step, estimating a weathering rate every day. The time-step is numerically determined by the stiffness of the differential equations in the system. The time-step is set automatically by the model numeric routine and thus is variable and is optimized during integration. Under conditions where short-term changes happen, the time-step may be on the scale of hours.

[Figure]

*Figure 5. Example of a soil weathering rate calculation for Niwot Ridge, Rock Mountain National Park, Colorado for four different environmental pollution scenarios and their effect on the ecosystems (trees and biodiversity): 1) background acid deposition from sulphur and nitrogen, 2) current policy, 3) no pollution control and 4) elevated temperature. The weathering rate, here reported in terms of the equivalents of the sum of base cation released (See section 4.1), 
[revised manuscript text omitted]
 fewer active surface complexes (Sverdrup 1990, Sverdrup and Warfvinge 1995). $Al^{3+}$ is the concentration of positively charged aluminium ions in the solution (Sverdrup 1990 – see also section 4.8). The subscript BC,OH represents a term related to base cations (BC) in the OH⁻ reaction, Note this slowing of the rates with increasing fluid concentration is not due to the approach to a mineral-water equilibrium state. The dissolution of many primary silicate minerals is not reversible under normal soil conditions as the fluids do not attain close to equilibrium conditions. The dissolution of most primary minerals is irreversible under normal soil conditions, and thus there is no equilibrium between the mineral surface and the soil solutions. Instead, there will be a steady state between the reaction at the surface and the removal of ions by solute transport and precipitation into secondary phases. This may look like an equilibrium, but does not behave like one. A few minerals are exceptions such as calcite, a few other carbonates, hydroxides and quartz. Even with these the attainment of equilibrium is kinetically limited. For calcite in soils we have observed this to take several days or weeks (Warfinge et al., 1987). All other minerals (feldspars, pyroxenes, amphiboles, etc..) do not precipitate from solution, some amorphous aluminosilicate clay precursors only precipitate very slowly.

**4.5. The updated kinetics equation**
The original 4 mineral dissolution reactions have been enlarged to include OH⁻-reaction in the present study. The complete equation 
[revised manuscript text omitted]

Lampa et al. (2016) and Nyström-Claesson and Andersson, (1996). Such observations demonstrate a need to take into account the complete set of processes occurring in the soil.

[Figure]

*Figure 27. Comparison of calculated with measured base cation concentrations at the Svartberget field site, (Zanchi et al., 2016). Note the base cation concentrations (*[Bc]*) refer to the sum of the concentrations of Na, H, Ca, and Mg in units of microequivalents per litre.*

[Figure]

|                      |                      |
|:--------------------:|:--------------------:|
| a: Base cations      | b: Silica            |

*Figure 28. Modelled base cation (a) and Si (b) concentrations plotted against $\log_{10}$ of water transit time (smooth lines) at the Svartberget field site (See Erlandsson-Lampa et al., 2016, 2019 for a full description of the field test of the model). Overlain are the observed base cation and Si-concentrations from the soil profile, plotted against $\log_{10}$ of soil depth (straight lines with symbols).*

Note that the mineral dissolution 'brake functions' used in this approach act differently on the weathering rates that the equilibrium expressions used in earlier models (Aagaard and Helgeson 1982, Murphy et al., 1987, Alekseyev et al., 1997, 2004, 2007, Oelkers, 2001, Oelkers et al., 1994, 2001, 2008). The preference for using the brakes rather than the traditional rate expression based on a slowing of rates as equilibrium is approached between the surface and the liquid is that equilibrium is not approached for many primary silicate minerals and thus the weathering process is irreversible.

**7. Conclusions**

The complex nature of weathering in the field is nearly impossible to interpret without a comprehensive model for the whole process. A first step to such interpretations can be the quantitative description of the dissolution rates of the major rock forming minerals. Even the dissolution rates of an

individual mineral can involve several simultaneous reactions. Thus, experimentally measured rates results can only be accurately interpreted when a full system model is used. Under field conditions, mineral dissolution is coupled to other soil processes, and thus a full ecosystem system model is needed for their interpretation. The apparent difference between field and laboratory dissolution rates arise from the coupling of these processes, and disappear once a full model is employed. Use of a fully coupled model shows these differences to be negligible (Keegan and Laskow-Lehey 2014).

Taking account the vast literature reporting experimentally measured mineral dissolution rates, it was possible to create a fully parameterized kinetic database for about 100 minerals. About 40% of the kinetic parameters were determined directly from experiment interpretations, and the rest were determined from inter-mineral interpolations and using of analogues.

The adjustment of the aluminium 'brake function' and the introduction of a silica "brake function" as described in this work were necessary to improve the description of weathering rates in the lower part of the soil, below 1 meter depth. The test at the Svartberget catchment suggests that this revised mineral dissolution model works adequately as can be seen from Figures 27-28.

**8. Acknowledgements**

This work is based upon that of Prof. Dr. Harald Sverdrup and Prof. Dr. Per Warfvinge who initiated the new model approaches in the 1980's., Since then major contributions have been made by Dr. Matthias Alveteg (Uncertainty, programming the code), Prof. Per Warfvinge (Programming the code), Dr. Cecilia Akselsson (Regional parameterizations, geostatistics), Dr. Salim Belyazid (Programming the code, applying the model) and Daniel Kurz (Adaption of the mineral stoichiometry, adaptation to Switzerland). Dr. Johan Holmqvist and Harald Sverdrup were instrumental in taking up a second long campaign in weathering experiments, generating more kinetic data during 1997-2004. Dr. Johan Holmqvist carefully worked out the geostatistics of landscape sampling and robustness of regional parameterizations and creating geostatistically sound regional weathering rate maps), 
[revised manuscript text omitted]